# VideoNSA: Native Sparse Attention Scales Video Understanding

Enxin Song[1], Wenhao Chai[2], Shusheng Yang[3], Ethan Armand[1], Xiaojun Shan[1], Haiyang Xu[1], Jianwen Xie[4], and Zhuowen Tu[1]

[1]University of California, San Diego    [2]Princeton University    [3]New York University
[4]Lambda, Inc

## Abstract

Video understanding in multimodal language models remains limited by context length: models often miss key transition frames and struggle to maintain coherence across long time scales. To address this, we adapt Native Sparse Attention (NSA) to video-language models. **Our method, VideoNSA, adapts Qwen2.5-VL through end-to-end training on a 216K video instruction dataset. We employ a hardware-aware hybrid approach to attention, preserving dense attention for text, while employing NSA for video.** Compared to token-compression and training-free sparse baselines, VideoNSA achieves competitive performance on long-video understanding, temporal reasoning, and spatial benchmarks. Further ablation analysis reveals four key findings: (1) reliable scaling to 128K tokens; (2) an optimal global–local attention allocation at a fixed budget; (3) task-dependent branch usage patterns; and (4) the learnable combined sparse attention help induce dynamic attention sinks.

**Project page:** https://enxinsong.com/VideoNSA-web/
**Code:** https://github.com/Espere-1119-Song/VideoNSA
**Model:** https://huggingface.co/Enxin/VideoNSA

## 1 Introduction

Key moments of a video can occur at any time, exemplified by soccer where game deciding moments typically span seconds of a 90 minute game. Within those game deciding moments split second actions define the outcome: an assist, a missed tackle, the movement of the keeper. Multimodal large language models (MLLMs)(Team, 2025; Team et al., 2025b;a; Qiu et al., 2025; Yuan et al., 2025a; Gui et al., 2025; Chung et al., 2025; Li et al., 2025b) have achieved substantial progress in vision-language perception and reasoning, but still cannot match humans ability to extract and reason about salient moments in videos. While humans naturally sample color visuals around 60hz, (Kalloniatis & Luu, 2007) across large contexts, existing VLMs often sample a single frame per second. Intuitively, increasing the context for these models by sampling more frames improves accuracy (Cai et al., 2024; Wu et al., 2024a), particularly for long videos and complex reasoning tasks. However, this approach pays for improvement with additional tokens, increasing computational complexity and pushing against fundamental limits of model context.

To address these challenges, many approaches (Wang et al., 2024; Li et al., 2024b; Jin et al., 2024; Wang et al., 2025a; Yang et al., 2024) adopt token compression to reduce redundancy and increase informative context. However, when applied to complex reasoning tasks, these compression-based models perform worse compared to full-token methods (Song et al., 2025a). Moreover, compression strategies often limit generalization through reduced perception and reasoning capacity (Wen et al., 2025). In contrast, sparse attention mechanisms preserve tokens, but focus the models capabilities on relevant dependencies between tokens. Numerous sparse attention methods have already been employed in large language models (LLMs), but most are inadequate for video complexity (detailed in Appendix A). Therefore, we present VideoNSA, which adopts Native Sparse Attention (Yuan et al., 2025c), a learnable hardware-aware sparse attention mechanism proven to be effective in long-

context modeling. VideoNSA is the first learnable and hardware-aware sparse attention framework tailored for video understanding, effectively scaling to ultra-long vision-text context. We apply the learnable sparse attention to video token sequences, while preserving grouped-query attention for text tokens. Following this pattern, our experiments show that using only 3.6% of the attention budget on 128K context length while improving performance on various tasks

We further conduct massive experiments and analyses of VideoNSA , revealing several important findings: (1) VideoNSA extrapolates effectively to contexts beyond its training length, and the optimal balance between temporal density and spatial resolution is highly task dependent. (2) VideoNSA is also sensitive by attention scaling, with results remaining strongest near the training configuration. (3) The gating distribution evolves dynamically across layers, and the selection and sliding-window branches gradually lose importance in deeper layers. (4) The compression branch emerges as the main computational bottleneck. (5)Moreover, the learned sparse attention weights remain beneficial even under dense attention settings. (6) Learnable sparse attention induces distinctive attention sink behaviors across branches, with very few sinks in the selection branch and periodic sink formation in the compression branch.

In particular, our paper makes the following contributions:

- We propose VideoNSA, a hardware-aware native sparse attention mechanism, and systematically investigate its effectiveness for video understanding, scaling up to a 128K vision context length.
- We introduce hybrid sparse attention in VideoNSA, enabling flexible allocation of information and attention budgets to achieve optimal performance across diverse task.
- We dynamically combine global and local attention through three complementary branches, which effectively reduce attention sinks in long vision contexts.

## 2 VIDEONSA

### 2.1 PRELIMINARIES

**Native sparse attention.**    Existing training-free sparse attention methods are rarely hardware aligned, and typically don't increase training efficiency. Native Sparse Attention (Yuan et al., 2025c) (NSA) avoids computing attention between all key-value pairs $(\mathbf{K}_t, \mathbf{V}_t)$, instead, for each query $\mathbf{q}_t$, NSA dynamically constructs an information-dense KV cache subset. NSA combines three complementary cache branches with a learnable gate $g_t^c$ adaptively weighting each branch yielding $\mathbf{o}_t$:

$$\mathbf{o}_t = \sum_{c \in \{\mathrm{cmp,\ slc,\ win}\}} g_t^c \cdot \mathrm{Attn}\big(q_t, \tilde{\mathbf{K}}_t^c, \tilde{\mathbf{V}}_t^c\big). \tag{1}$$

Token Compression (CMP) branch aggregates sequential blocks of keys into more coarse-grained, single block-level representations $\tilde{\mathbf{K}}_t^{\mathrm{cmp}}$ via a learnable MLP $\varphi$:

$$\tilde{\mathbf{K}}_t^{\mathrm{cmp}} = \{\varphi(\mathbf{K}_{[id+1:id+m]}) \mid 0 \le i < \lfloor \frac{t-m}{d} \rfloor\}, \tag{2}$$

where $m$ is the block length, $d$ is the stride.

Token Selection (SLC) branch preserves the most salient key-value blocks by computing importance scores $p_t^{\mathrm{slc'}}$ and selecting the indices of the top-n blocks:

$$I_t = \{i \mid \mathrm{rank}(p_t^{\mathrm{slc'}}[i]) \le n\}. \tag{3}$$

The final set of selected keys is formed by concatenating these top-ranked blocks:

$$\tilde{\mathbf{K}}_t^{\mathrm{slc}} = \mathrm{Cat}(\{\mathbf{K}_{[im'+1:(i+1)m']} \mid i \in I_t\}), \tag{4}$$

where $I_t$ is the set of selected indices, $n$ is the number of blocks to retain.

Sliding Window (SWA) branch simply applies the standard sliding window attention, which retains the fixed $w$ most recent key-value pairs:

$$\tilde{\mathbf{K}}_t^{\mathrm{swa}} = \mathbf{K}_{t-w+1:t}, \quad \tilde{\mathbf{V}}_t^{\mathrm{swa}} = \mathbf{V}_{t-w+1:t}. \tag{5}$$

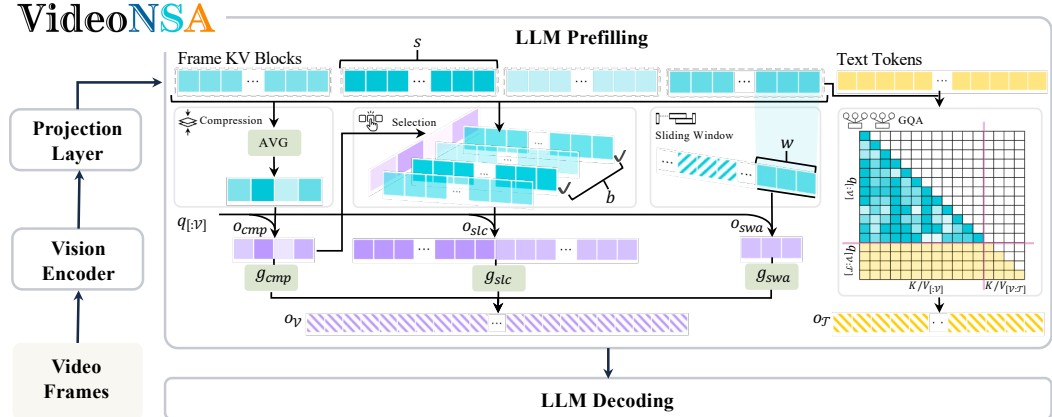

Figure 1: Overview of VideoNSA. Video frames are encoded into frame-level KV blocks. VideoNSA utilizes three sparse attention branches during prefilling stage: **compression branch** reduces redundancy via token averaging, **selection branch** identifies top-k important tokens, and **sliding window branch** enforces local temporal coverage. The outputs are combined through dynamic gating before integration with text tokens for LLM decoding.

**Grouped query attention.** In Multi-Head Attention (MHA), each query head has dedicated key–value (KV) projections, which makes the KV cache scale with the number of heads and increases inference cost. Grouped-Query Attention (GQA) (Ainslie et al., 2023) mitigates this by letting multiple query heads share fewer KV heads. For each input $\{x_i\}_{i=1}^{L}$, GQA partitions the $h$ query heads into $g$ groups ($1 \leq g \leq h$). At a given timestep $t$, the output $o_t^{(s)}$ for the $s$-th query head with group index $m(s) = \lceil sg/h \rceil$ is computed by applying attention to the shared keys and values as:

$$o_t^{(s)} = \text{Attention}(q_t^{(s)}, K_{\leq t}^{(m(s))}, V_{\leq t}^{(m(s))}) = \text{softmax}\left(\frac{(q_t^{(s)})^\top K_{\leq t}^{(m(s))}}{\sqrt{d_k}}\right) V_{\leq t}^{(m(s))}, \qquad (6)$$

where $q_i^{(s)} = x_i W_q^{(s)}$, $k_i^{(m(s))} = x_i W_k^{(m(s))}$, $v_i^{(m(s))} = x_i W_v^{(m(s))}$. The outputs $o_t$ from all heads are concatenated by $o_t = [o_t^{(1)}, o_t^{(2)}, \ldots, o_t^{(h)}]$. VideoNSA utilizes Qwen2.5-VL-7B (Bai et al., 2025) as the backbone, with Qwen2.5-7B (Qwen et al., 2025) as the LLM decoder, which employs GQA for efficient KV cache utilization using 28 query heads and 4 shared key/value heads.

## 2.2 ARCHITECTURE

Existing token compression methods (Yang et al., 2025d; Zhang et al., 2025c; Hyun et al., 2025; Zhang et al., 2025h) suffer from irreversible information loss on complex tasks and don't address computational and latency bottlenecks in LLM video understanding. From the perspective of attention as a message passing in a Graph Neural Network (Joshi, 2025; Pappone, 2025), it's clear this bottleneck is fundamental. Standard attention propagates information between nodes (tokens) through edges (attention weights), with each token being updated by aggregating features from its neighbors, weighted by attention scores. Training-free sparse attention often imposes a static adjacency matrix whose fixed subgraph connectivity restricts information flow. Conversely, NSA (Yuan et al., 2025c) provides data-dependent sparsity that preserves edges necessary for a particular task.

We build VideoNSA upon Qwen2.5-VL-7B (Qwen et al., 2025), which incorporates a vision encoder and adopts Qwen2.5-7B (Bai et al., 2025) as the LLM. As illustrated in Figure 1, VideoNSA introduces a hybrid attention mechanism in the LLM across different modalities. At each layer $l$, we split the input tokens $\mathbf{X}^{(l-1)}$ into vision tokens $\mathbf{X}_{\mathcal{V}}^{(l-1)}$ and text tokens $\mathbf{X}_{\mathcal{T}}^{(l-1)}$ according to their position IDs. For vision tokens, VideoNSA applies NSA (Yuan et al., 2025c) with a dedicated gate $g_t^c$ on each head. We set the block size $s$ equal to the token number per frame, and obtain the block-level representation by averaging all tokens within the block. The vision attention output $o_{\mathcal{V}}$ is dynamically

weighted by the compression, selection, and sliding window branches as:

$$\mathbf{o}_{\mathcal{V}}^{(l)} = \sum_{c \in \{\text{cmp,slc,win}\}} g_t^c \, \text{Attn}\big(q_t, \tilde{\mathbf{K}}_t^c, \tilde{\mathbf{V}}_t^c\big),$$

where $g_t^c$ is implemented as a two-layer MLP with a sigmoid activation.

The text attention output $\mathbf{o}_{\mathcal{T}}^{(l)}$ is computed using standard GQA (Ainslie et al., 2023) to preserve instruction following capabilities. We obtain the final output $\mathbf{o}^{(l)}$ of the layer $l$ by concatenating:

$$\mathbf{o}^{(l)} = [\,\mathbf{o}_{\mathcal{V}}^{(l)}; \, \mathbf{o}_{\mathcal{T}}^{(l)}\,].$$

## 2.3 TRAINING RECIPE

We conduct end-to-end training to adapt vision features for data-dependent sparse connectivity in the language model. The training dataset of VideoNSA is constructed from LLaVA-Video-178K (Zhang et al., 2024d) by filtering for question answer pairs at 4 fps and retaining videos with 350–550 frames, for a subset of 216K pairs. To emphasize sparse attention for temporal redundancy, we constrain the maximum pixels per frame to 50,176, and the maximum context length per training instance to 36K tokens. In VideoNSA, block size $s$ is set to 64, block $b$ is set to 32, and sliding window size $w$ is set to 256. We trained using SWIFT (Zhao et al., 2024), adapting the NSA (Yuan et al., 2025c) implementation from FLA (Yang & Zhang, 2024) and (Pai et al., 2025b). The complete training process requires 4600 H100 GPU hours. More training details including hyper-parameters selection can be found in Appendix B.

## 3 EXPERIMENTS

### 3.1 EFFECTIVENESS ON VIDEO UNDERSTANDING

**Baselines** Our primary baseline is Qwen2.5-VL-7B (Qwen et al., 2025) with dense FlashAttention (Dao, 2023). We compare VideoNSA against several strong baselines, including the quantization model AWQ (Team, 2024), training-free token compression models (Yang et al., 2025c; Zhang et al., 2025b; Chen et al., 2024a), and training-free sparse attention methods (Jiang et al., 2024; Xu et al., 2025a; Lai et al., 2025; Li et al., 2024c). All methods employ their official configuration without additional training and using Qwen2.5-VL-7B (Qwen et al., 2025) as a base. For token compression baselines, we use the token kept ratio and sampling fps from the original papers that yield the best accuracy, while for sparse attention baselines, we use the same configuration as VideoNSA. In addition, we fine-tune Qwen2.5-VL-7B (Qwen et al., 2025) using the same training dataset as VideoNSA to serve as a competitive baseline. We also include models with different backbones for a broad comparison.

We evaluate VideoNSA across three domains including **long video understanding**, **temporal reasoning**, and **spatial understanding** using LMMs-Eval (Zhang et al., 2024a) and VLMEvalKit (Duan et al., 2024). Table 1 indicates that sparse attention methods consistently outperform token compression approaches. We empirically evaluate the effectiveness of VideoNSA based on several popular long video understanding benchmarks, including **LongVideoBench** (Wu et al., 2024a), **MLVU** (Zhou et al., 2024), **TimeScope** (Zohar et al., 2025) and **LongTimeScope** (Zohar et al., 2025). VideoNSA achieves competitive results, narrowing the gap with state-of-the-art methods. We observe that VideoNSA shows clear advantages on tasks involving order-sensitive temporal reasoning and ultra-long video settings (**10 hours** in LongTimeScope (Zohar et al., 2025)). To evaluate the visual temporal reasoning capbility of VideoNSA, we evaluate VideoNSA on **Tomato** (Shangguan et al., 2024), a benchmark spanning six reasoning types and three video scenarios. VideoNSA attains the highest accuracy on Tomato (Shangguan et al., 2024), substantially outperforming compression-based methods, underscoring their limitations in fine-grained temporal inference. **VSIBench** (Yang et al., 2025a) focuses on spatial reasoning allowing us to test whether efficient models can preserve local fidelity while achieving efficiency. VideoNSA matches the strongest sparse attention baselines and significantly surpasses token compression methods in spatial understanding, confirming that it preserves spatial fidelity. All detailed evaluation settings and subset results can be found in Appendix C, Appendix D, Appendix E, and Appendix F.

Table 1: Results on long video understanding, temporal reasoning and spatial understanding tasks. LVB, LTS for LongVideobench (Wu et al., 2024a) and LongTimeScope (Zohar et al., 2025).

| Model | Long-form Video | | | | Temporal | Spatial |
|---|---|---|---|---|---|---|
| | LVB | $MLVU_{test}$ | TimeScope | LTS | Tomato | VSIBench |
| LLaVA-OneVision-7B (Li et al., 2024a) | 56.3 | – | – | – | 25.5 | 32.4 |
| LLaVA-Video-7B (Zhang et al., 2024c) | 58.2 | – | 74.1 | 34.0 | – | 35.6 |
| VideoLLaMA3-8B (Zhang et al., 2025a) | 59.8 | 47.7 | 69.5 | – | – | – |
| InternVL2.5-8B (Chen et al., 2024c) | 60.0 | – | 55.8 | – | – | – |
| Video-XL-2 (Qin et al., 2025b) | **61.0** | **52.2** | – | – | – | – |
| Qwen2.5-VL-7B (Qwen et al., 2025) | 58.7 | 51.2 | 81.0 | 40.7 | 22.6 | 29.7 |
| Qwen2.5-VL-7B-AWQ (Team, 2024) | 59.0 | 46.0 | – | – | – | 35.0 |
| Qwen2.5-VL-7B-SFT | 57.8 | 51.2 | 76.8 | 40.2 | 21.7 | 30.5 |
| *Token Compression Methods* | | | | | | |
| + FastV (Chen et al., 2024a) | 57.3 | 41.8 | 46.5 | 35.6 | 21.6 | 32.0 |
| + VScan (Zhang et al., 2025b) | 58.7 | 48.1 | 80.3 | 31.1 | 19.1 | 34.4 |
| + VisionZip (Yang et al., 2025c) | 52.4 | 33.1 | 43.5 | 40.4 | 23.6 | 32.1 |
| *Sparse Attention Methods* | | | | | | |
| + Tri-Shape (Li et al., 2024c) | 59.5 | 49.2 | 82.7 | 28.4 | 22.1 | 34.9 |
| + MInference (Jiang et al., 2024) | 59.2 | 49.2 | 82.7 | **44.4** | 23.0 | 36.5 |
| + FlexPrefill (Lai et al., 2025) | 58.4 | 46.0 | 83.0 | 39.1 | 23.7 | 34.0 |
| + XAttention (Xu et al., 2025a) | 59.1 | 50.2 | 83.1 | 41.1 | 21.4 | **36.6** |
| **VideoNSA** | 60.0 | 51.8 | **83.7** | 44.4 | **26.5** | 36.1 |

Table 2: Ablation study on branch selection across different tasks. LVB, LTS for LongVideobench (Wu et al., 2024a) and LongTimeScope (Zohar et al., 2025).

| Branch | | | Long Video Understanding | | | | Temporal Reasoning | Spatial Understanding |
|---|---|---|---|---|---|---|---|---|
| CMP | SLC | SWD | LVB | $MLVU_{test}$ | TimeScope | LTS | Tomato | VSIBench |
| ✓ | | | 48.1 | 43.9 | 41.5 | 25.1 | 23.3 | 29.2 |
| | ✓ | | 48.4 | 47.7 | 63.7 | 37.1 | 24.0 | 27.6 |
| | | ✓ | 49.1 | 40.2 | 59.3 | 29.8 | 24.0 | 29.8 |
| ✓ | ✓ | | 49.4 | 42.7 | 57.3 | 32.4 | 23.5 | 29.4 |
| ✓ | | ✓ | 49.3 | 42.4 | 65.2 | 34.4 | 23.0 | 29.1 |
| | ✓ | ✓ | 48.8 | 43.4 | 57.3 | 31.6 | 24.5 | 30.3 |
| ✓ | ✓ | ✓ | **60.0** | **51.8** | **83.7** | **44.4** | **26.5** | **36.1** |

## 3.2 ABLATION STUDY

To further analyze the components of VideoNSA, we visualize attention pattern in each branch in Appendix H and assess the effectiveness of different branches. Table 2 shows that single-branch models suffer significant degradation, and even two-branch combinations remain inferior to the full VideoNSA, highlighting the necessity of integrating all three branches with dynamic gating. Detailed results of different branch combination can be found in Appendix I.

## 4 SCALING ANALYSIS AND FINDINGS

**Finding 1.** Do learned sparse attention weights remain beneficial in dense attention settings?

Table 3: Ablation study on transferring sparse attention weights to dense attention across tasks.

| Model | Long Video Understanding | | | | Temporal Reasoning | Spatial Understanding |
|---|---|---|---|---|---|---|
| | LongVideoBench | $MLVU_{Test}$ | TimeScope | LongTimeScope | Tomato | VSIBench |
| Qwen2.5-VL-7B | 58.7 | 51.2 | 81.0 | 40.7 | 22.6 | 29.7 |
| Dense-SFT | 57.8 (-1.5%) | 51.2 (+0.0%) | 76.8 (-5.2%) | 40.2 (-1.2%) | 21.7 (-4.0%) | 30.6 (+2.1%) |
| Dense-NSA | 56.1 (-4.4%) | 51.6 (+0.8%) | **83.0 (+2.5%)** | 40.9 (+0.5%) | 23.4 (+3.5%) | 33.1 (+10.7%) |
| VideoNSA | **59.4 (+1.1%)** | **51.8 (+1.2%)** | 82.7 (+2.1%) | **44.4 (+9.1%)** | **26.2 (+15.9%)** | **36.1 (+20.3%)** |

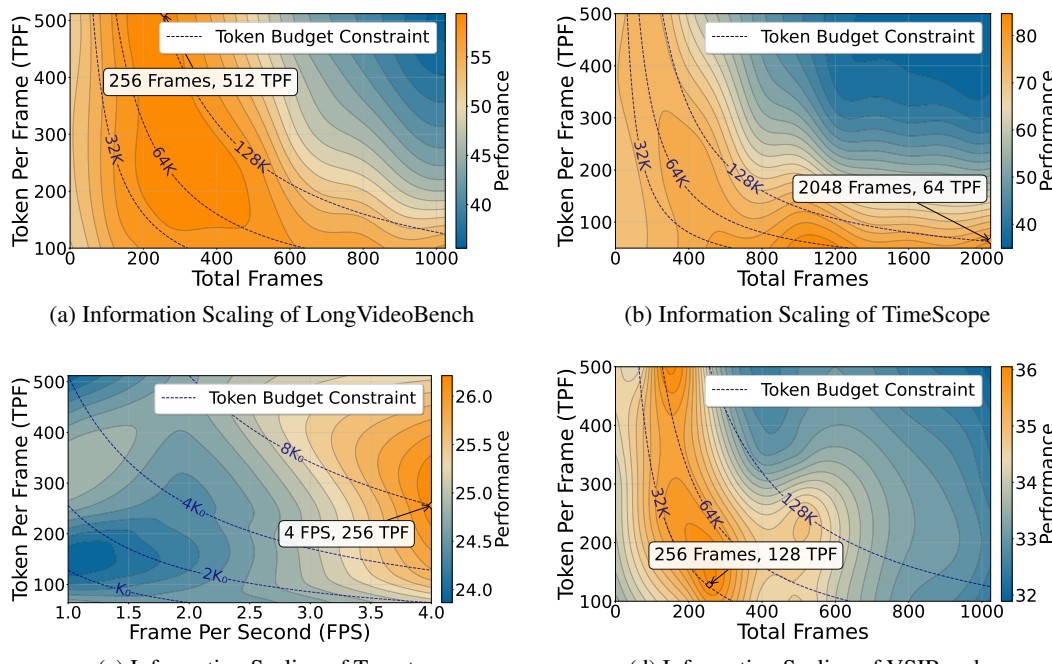

Figure 2: Scaling Performance of VideoNSA under Different Context Allocation Strategies. We highlight the Token Budget Constraint to indicate settings with equal context length, and annotate the best-performing configuration under each benchmark. Since videos in Tomato (Shangguan et al., 2024), we vary FPS instead of total frames, with FPS × TPF = 128 denoted as $K_0$.

We further examine whether the learned QKV weights of VideoNSA can imrpove performance in dense attention inference. Table 3 reports the relative performance change over the Qwen2.5-VL-7B (Qwen et al., 2025). Due to the limited quality of the training data, our fine-tuned Qwen2.5-VL-7B (Dense-SFT) exhibits slight performance drops on most benchmarks. We observe that the transferred model (Dense-NSA) allows the dense variant to recover and surpass the baseline on several benchmarks suggesting that sparse-trained weights provides inductive bias towards more effective attention distributions. However, the effect remains limited on LongVideoBench (Wu et al., 2024a). VideoNSA significantly outperforms Dense-NSA on most tasks, highlighting the importance of runtime sparsity and dynamic gating.

> **Finding 2.** How far can VideoNSA scale in context length?

The effective vision context length $L$ is jointly determined by the number of vision tokens per frame $T$ and the total number of input frames $F$. VideoNSA is trained with a maximum context length of $L = 36K$ tokens, corresponding to $T = 64$ tokens per frame. We conduct an information budget study under a fixed context length, by varying tokens per frame and frame rate. We then scale up the context length beyond the training budget, evaluating up to the maximum 128K tokens supported by the language model. As observed in Figure 2, the model consistently achieves higher performance when scaled to longer contexts beyond its training length across benchmarks. However, the ideal allocation of same token budget is highly task-dependent. LongVideoBench (Wu et al., 2024a) favors allocating more tokens per frame, while Tomato (Shangguan et al., 2024) and TimeScope (Zohar et al., 2025) benefit more from increasing the number of frames, emphasizing temporal coverage. VSIBench (Yang et al., 2025a) shows mixed preferences depending on context length, reflecting a balance between spatial and temporal sampling. Additional results on information scaling are reported in Appendix J.

> **Finding 3.** How to allocate the attention budget?

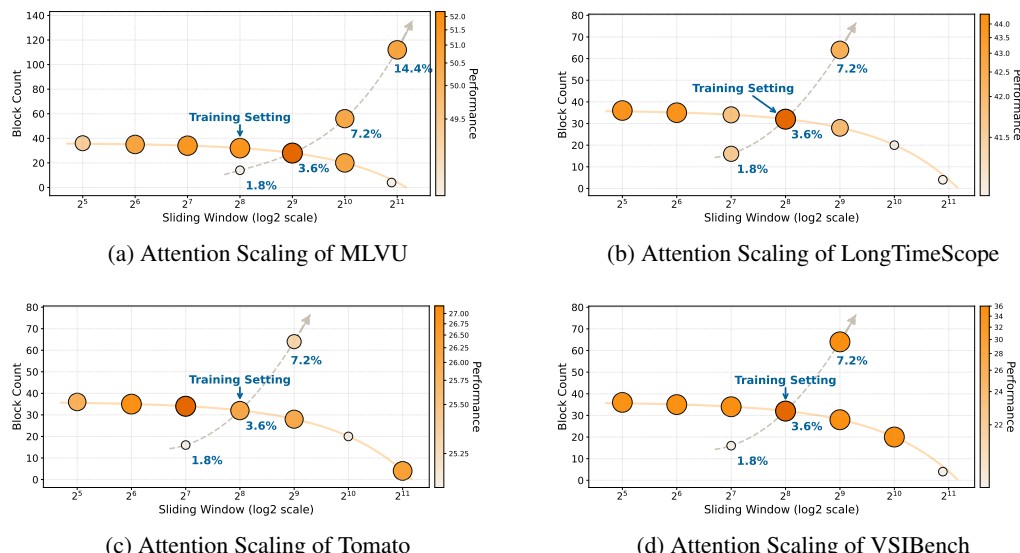

(a) Attention Scaling of MLVU

(b) Attention Scaling of LongTimeScope

(c) Attention Scaling of Tomato

(d) Attention Scaling of VSIBench

Figure 3: Scaling Performance of VideoNSA under Different Attention Allocation Strategies. Scatter points from small to large and from light to dark indicate increasing performance. We annotate the point corresponding to the same attention allocation strategy as used during training and connect configurations with equal attention budgets using solid orange lines. We further scale the best configuration using dashed lines. Percentages show attention relative to full attention.

We define the *Attention Budget* as the total number of key-value pairs visible to each query, denoted by $K_{vis}$. It is composed of a global sparse component and a local sliding-window component as: $K_{attn} = b \times s + w$, where $b$ and $s$ denote the number and size of global blocks, and $w$ is the sliding-window width. With context length $L$, compared to causal dense attention with $\frac{L(L-1)}{2}$ edges, the fraction of attention used $\gamma$ is

$$\gamma = \frac{L(cS + w)}{\frac{L(L-1)}{2}} = \frac{2(cS + w)}{L - 1},$$

To determine the optimal attention allocation, we first fix the total sequence length $L$, the attention budget $K_{\text{vis}}$, and the block size $S = 64$, while systematically varying the local attention ratio $\alpha = \frac{w}{K_{\text{attn}}}$. We then employ the optimal allocation ratio $\alpha^\star$ for attention budget scaling. As shown in Figure 3, scatter points denote different allocation strategies, with their size and color reflecting performance. We highlight the point corresponding to the training configuration, connect equal-budget settings with solid orange lines, and extend the best-performing configuration with dashed lines, where the annotated values indicate the fraction of attention used $\gamma$. Results show that model performance is highly sensitive to attention allocation. Although the optimal ratio between global and local attention varies across tasks, configurations close to the training allocation generally yield better results. Under the same budget, fine-tuning around the training setting often improves performance, whereas simply enlarging the overall budget does not consistently bring further gains. Moreover, across most benchmarks, increasing global attention (enlarging the block count) tends to outperform increasing local attention (enlarging the sliding window). Remarkably, VideoNSA achieves leading performance using only 3.6% of the full attention budget. More results are in Appendix L.

**Finding 4.** What roles do compression, selection, and sliding-window gates play in VideoNSA?

We analyze the gating distribution of VideoNSA across Tomato (Shangguan et al., 2024), VSI-Bench (Yang et al., 2025a), and LongVideoBench (Wu et al., 2024a), and aggregate the average routing gate weights over 100 examples from each. As illustrated in Figure 4, where shaded bars denote the interquartile range and horizontal lines represent mean values, each head in VideoNSA exhibits distinct and diverse preferences across branches throughout its full depth. The diversity allows different layers to specialize in distinct modes of the context-dependent information flow. The

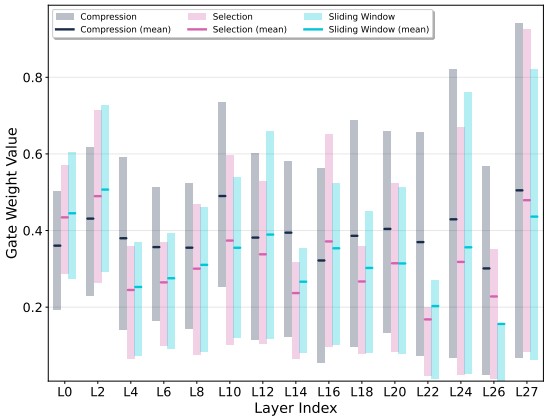

Figure 4: Gate weights across layers in VideoNSA. Compression remains dominant, while selection and sliding-window weaken in later layers.

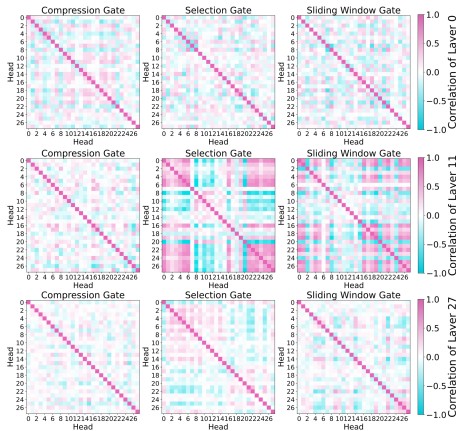

Figure 5: Inter-head similarities of gates in VideoNSA. Selection and sliding-window gates show high similarity in middle layers.

compression branch maintains relatively high average weights across most layers, underscoring its primary role in reducing redundancy while preserving salient features. The selection and sliding window gates fluctuate more strongly, occasionally surpassing the compression branch in early and middle layers. However, their contributions diminish in the final layers (e.g., L22–L26), demonstrating that the focus shifts towards aggregating high-level features. We also note strange behavior in the last layer, where all three branches are fully active despite selection and sliding window being inactive in the layers before. Full gate values distribution in Appendix N.

We further dive into the inter-head gate similarity of each layer in Figure 5. In the middle layers, both selection and sliding window gates exhibit pronounced increases in inter-head similarity. This indicates that multiple mid-layer heads converge to highly consistent gating behaviors when the model performs block selection and local temporal integration. However, the compression gate shows consistently low inter-head similarity, indicating that it operates largely in a head-independent manner. At both the initial and final layers of VideoNSA, inter-head similarity remains weak across all gates, reflecting the need to maintain diversity in early representations and to support mixing information in higher-level abstractions. More inter-head gate similarites visualization in Appendix O.

> **Finding 5.** Where does the efficiency bottleneck come from?

We measure the inference latency of each branch in VideoNSA using wall-clock time across varying context lengths from $1K$ to $128K$. The compression branch dominates runtime as the context grows, while the selection and sliding window branches contribute relatively little at longer contexts. Ideally, the compression branch grows approximately linearly with $L$, and the sliding window branch has a complexity of $O(L \cdot w)$, which results in linear scaling for a fixed window size $w$. The selection branch requires computing importance scores over all $L/b$ blocks per query, leading to a computational complexity of $O(L^2/b)$. However, wall-clock latency deviates from these estimates due to hardware parallelism, memory access patterns, and kernel

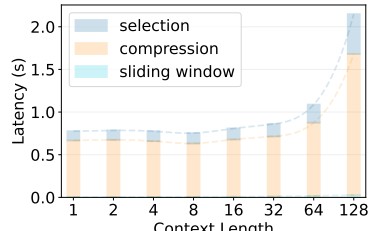

Figure 6: Inference latency of each branch in VideoNSA.

launch overheads. Overall, the compression branch emerges as the primary bottleneck, highlighting the need for further optimization of its kernel design and memory efficiency.

> **Finding 6.** Do learnable sparse mechanisms induce dynamic attention sinks?

In decoder-only transformers, a disproportionate amount of attention is often allocated to the first few tokens, which act as attention sinks and absorb excessive attention mass as a byproduct of softmax normalization. Prior studies (Gu et al., 2024; Xiao et al., 2023) show that attention sinks arise from massive activations and unusually small key and value norms, so attention directed to these tokens contributes little to the residual state. This raises an important question in learnable sparse attention: whether sparsity patterns amplify or mitigate such sinks.

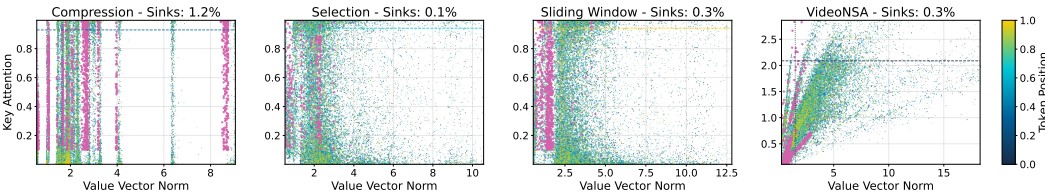

Figure 7: Attention sinks distribution of different branches. VideoNSA maintains a low overall sink ratio, with pink points indicating identified sinks.

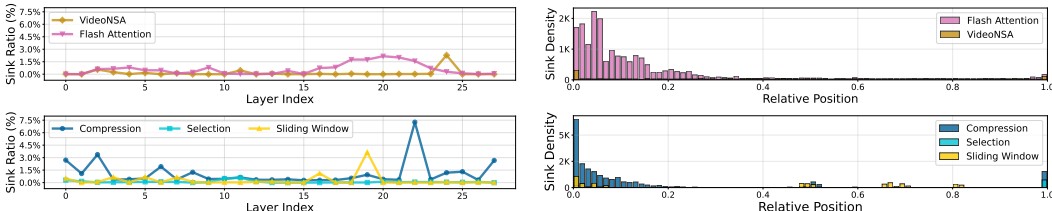

Figure 8: Layer-wise attention sink ratio distribution in different branches and Flash Attention.

Figure 9: Relative positions of attention sinks in different branches and Flash Attention.

We follow the attention sink defination in (Pai et al., 2025a):

$$\text{Attention Sink} = \mathbf{1}\Big\{\alpha > 0.1 \ \wedge \ \|v\| < \text{median}(\|v\|) - 2 \cdot \text{IQR}(\|v\|)\Big\},$$

where $\alpha$ is the average attention score received by the key, and $\|v\|$ is the value norm of the token.

Figure 7 illustrates the average distribution of attention sinks across the three branches of VideoNSA. Each frame is encoded into 256 tokens, and we adopt the same sparse attention configuration as used during training. The three branches exhibit markedly different sink behaviors. The compression branch produces the most sinks, with distinct banded concentrations along the value norm axis caused by token merging that amplifies some token norms while suppressing others. Conversely, the selection branch yields almost no sinks, as its top-$k$ block filtering mechanism enforces a smoother value norm distribution. Notably, the sliding window branch demonstrates a clearer separation between sink and non-sink tokens along the value norm axis. Critically, dynamic gating allows VideoNSA to counteract the negative effects of the compression branch, achieving a stable model with a low overall sink ratio of 0.3%.

Figure 8 indicates that VideoNSA maintains low sink ratios overall, with only minor fluctuations across layers. However, Flash Attention exhibits a gradual increase in sink ratios toward deeper layers. The compression branch maintains relatively high sink levels across most layers. The selection branch remains consistently close to zero, while the sliding window branch occasionally shows higher peaks in the middle-to-late layers, indicating that locality constraints may still introduce bias in long-sequence settings. From the perspective of positional distribution in Figure 9, Flash Attention produces sinks that are uniformly spread across the entire sequence due to its fully connected dense attention. Under dynamic gating, VideoNSA achieves smoother temporal coverage, alleviating over-reliance on early positions while avoiding the global diffusion characteristic of dense attention. In contrast, the compression branch exhibits strong accumulation at the beginning with an even steeper decay, indicating that token merging exerts its strongest impact on early-stage representations. The selection branch yields very few sinks across the sequence, while the sliding window branch produces sparse peaks at periodic boundaries of local neighborhoods. More analysis about attention sinks on various sparse attention settings can be found in Appendix S.

## 5 CONCLUSION

In this work, we present VideoNSA, a hybrid hardware-aware sparse attention model that significantly advances video understanding across various tasks. By dynamically fusing block-wise compression, salient block selection, and a sliding window, VideoNSA effectively preserves critical information while achieving near-linear scalability in efficiency and memory. Our experiments demonstrate that VideoNSA consistently outperforms existing methods on key tasks including long video understanding, temporal reasoning, and spatial understanding. While the prefill stage remains the primary bottleneck, our findings confirm that this hybrid sparse approach provides a powerful and scalable framework, paving the way for more capable video foundation models.

## 6 ACKNOWLEDGEMENT

This work is supported by NSF award IIS-2127544 and NSF award IIS-2433768. We thank Lambda, Inc. for their compute resource help on this project.

## 7 ETHICS STATEMENT

This research on video understanding utilizes publicly available datasets, ensuring that all data complies with privacy regulations. We acknowledge the potential biases that can arise in automatic answer generation, particularly concerning gender, race, or other characteristics. We have taken measures to evaluate and minimize such biases, while remaining committed to further improvements. Additionally, we recognize the potential risks of misuse, such as generating misleading answers, and have checked the training dataset with safeguards against such applications.

## 8 REPRODUCIBILITY STATEMENT

We have made several efforts to ensure the reproducibility of our work. All the key implementation details, including the architecture of our model, the training procedures, and hyperparameter settings, are described in supplementary meterial Section B. The settings of the used evaluation benchmarks are in Section C to further support reproducibility.

## 9 THE USE OF LARGE LANGUAGE MODELS

Large language models (LLMs) were used only for light editorial purposes, such as minor grammar checking and language polishing. They were not used for generating scientific content, research ideation, experiment design, or analysis. The authors take full responsibility for the entirety of the paper, and LLMs are not considered contributors or eligible for authorship.

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

# Appendix

The supplementary material is structured as follows:

- literature review about the related works in Section A.
- The training settings for VideoNSA in Section B.
- The introduction of the used evaluation benchmarks and settings in Section C.
- More results on long-form video benchmarks in Section D.
- More results on temporal reasoning benchmarks in Section E.
- More results on spatial understanding benchmarks in Section F.
- Results on additional video understanding benchmarks in Section G.
- Visualization of attention pattern in each branch in Section H.
- More results on branch combination in Section I.
- More results on information scaling study in Section J.
- Additional context-length scaling results of Qwen2.5-VL in Section K.
- More results on attention scaling study in Section L.
- Theoretical analysis of context length and attention budget scaling in Section M.
- Full gate values distribution in Section N.
- More inter-head gate similarites visualization in Section O.
- Benchmark-level gating analysis and PCA visualization in Section P.
- Additional analysis of training and inference efficiency in Section Q.
- Additional analysis on CMP latency bottleneck in Section R.
- More analysis about attention sinks on various sparse attention settings can be found in Section S.
- Comparison of sparsity patterns between text-only NSA and VideoNSA in Section T.
- Visualization of attention sinks in dense attention in Section U.

## A  RELATED WORK

### A.1  EFFICIENT VIDEO UNDERSTANDING

Video understanding systems typically convert videos into long sequences of vision tokens, which can easily exceed GPU memory and slow down inference as the video length grows. To address this, existing work mainly address this by **token compression**, **alternative sequence modeling**, and **KV-cache compression**. One important line of work emphasizes token compression. Spatial or temporal token merging methods (Wang et al., 2025c; Zhang & Fu, 2025; Li et al., 2025f; Jiang et al., 2025a; Li et al., 2025c; Shao et al., 2025; Song et al., 2024; Chai et al., 2024; Liao et al., 2025; Wu et al., 2025b; 2024b; 2025a; 2024c; Cao et al., 2024; Li et al., 2025a;e; Yang et al., 2023; Kong et al., 2025) progressively discard redundant content, while question-/task-aware strategies (Jiang et al., 2025b; Dong et al., 2025; Yao et al., 2025; Song et al., 2025b) tailor retained tokens to the query. These approaches substantially lower FLOPs but still rely on dense attention once tokens are merged. Beyond pure self-attention, Mamba-based or hybrid architectures (Jiang et al., 2025a; Ren et al., 2025; Xu et al., 2025b; Tao et al., 2025) inject state-space or recurrent modules to approach linear-time inference while preserving long-range dependencies. Also, there exists approach to design data efficient systems for further fine-tuning (Li et al., 2025d). Another direction targets the key–value cache during decoding via task-aware sparsification and streaming-friendly memory (Qin et al., 2025a; Ning et al., 2025; Kim et al., 2025; Yang et al., 2025e; Liu et al., 2025) reduce memory and improve throughput, yet prefill still scales quadratically with sequence length. In contrast to methods that mostly decide *where* to drop or compress tokens, our approach systematically probe the effectiveness of *native sparse attention* (Yuan et al., 2025b) that restructures attention itself to be learnable and sparse from the ground up. VideoNSA attains near-linear scalability up to 128K tokens and processes over 10,000 frames on a single GPU, outperforming compression-only pipelines on long-video understanding, temporal reasoning, and spatial understanding tasks.

## A.2 Sparse Attention Mechanism

Sparse attention is a central strategy for efficient long-context modeling in language and multi-modal systems. Surveys (Zhang et al., 2025e) categorize approaches into *pattern-based* vs. *dynamic/learned*. **Pattern-based sparsity.** Methods such as Longformer (Beltagy et al., 2020), StreamingLLM (Xiao et al., 2024), and TriangleMix (He et al., 2025) prescribe fixed local/strided patterns that can be applied training-free; recent multimodal works (Zhang et al., 2025d; Yang et al., 2025b) follow similar principles, while hardware-efficient kernels like Flash Sparse Attention (Yan et al., 2025) further reduce prefill latency. InfLLM-V2 (Zhao et al., 2025) uses switchable dense sparse attention to smoothly adapt models from short to long sequences while maintaining consistency and achieving efficient acceleration with high performance. ProxyAttn (Wang et al., 2025b) uses representative heads for fine-grained block importance estimation, enabling faster sparse attention with minimal performance loss. **Dynamic and trainable sparsity.** Content- or gradient-adaptive mechanisms select important connections (e.g., diagonal selection (Tyagi et al., 2025) or lag-relative strategies (Liang et al., 2025)); trainable sparse attention improves long-context reasoning (Gao et al., 2025; Vasylenko et al., 2025; Gao et al., 2024), diffusion-based video generation (Zhang et al., 2025g), and state-space models (Zhan et al., 2025). SLA (Zhang et al., 2025f) decomposes attention weights into critical, marginal, and negligible parts, combining sparse and low-rank acceleration to greatly reduce computation while preserving generation quality. Hybrid approaches such as RocketKV (Behnam et al., 2025) combine token/cache compression with learned sparsity, and MMInference (Li et al., 2025g) accelerates modality-aware sparse prefill for VLMs. Despite these advances, most techniques are optimized for text or short multimodal contexts and do not directly address the ultra-long, highly redundant spatio-temporal structure of videos. VideoNSA unifies *block-wise compression*, *salient block selection*, and a *sliding-window* branch under learnable gates that dynamically allocate computation across three native sparse branches (Yuan et al., 2025b). This end-to-end, data-driven design preserves critical global/local dependencies while scaling nearly linearly in both time and memory.

## B Detailed Training Settings

Training hyperparameters for VideoNSA are shown in Table 4. We filter a subset of LLaVA-Video-178K (Zhang et al., 2024e) as the training data. For each video, we uniformly sample at 4 frames per second and retain only those with 350–550 frames, resulting in 216K video question–answer pairs from the original 961K pairs in LLaVA-Video-178K (Zhang et al., 2024e).

Table 4: Training hyper-parameters for VideoNSA.

| Hyper-parameters | Fine-tuning |
|---|---|
| trainable parameters | ViT + MLP + LLM |
| warmup schedule | linear |
| warmup start factor | 1e-5 |
| warmup ratio | 0.1 |
| learning rate schedule | cosine |
| optimizer | AdamW (Loshchilov & Hutter, 2017) |
| optimizer hyper-parameters | $\beta_1, \beta_2 = (0.9, 0.999)$ |
| weight decay | 0.01 |
| max norm | 1 |
| epoch | 1 |
| peak learning rate | 1e-6 |
| total equivalent batch size | 32 |

## C Evaluation Benchmarks and Settings

We list all the hyper-parameters and prompt used for evaluation as shown in Table 5.

Table 5: Evaluation settings summary for each benchmarks. For all benchmarks we set temperature, top p, number of beams to 0, 0, 1 respectively. # TPF stands for the vision tokens per frame, and # F stands for the number of sampling frames.

| Benchmark | # TPF | # F | # Max New Tokens |
|---|---|---|---|
| LongVideoBench (Wu et al., 2024a) | 512 | 256 | 32 |
| LongTimeScope (Zohar et al., 2025) | 128 | 512 | 16 |
| TimeScope (Zohar et al., 2025) | 64 | 2048 | 16 |
| MLVU$_{test}$ (Zhou et al., 2024) | 128 | 512 | 16 |
| Tomato (Shangguan et al., 2024) | 4FPS | 256 | 1024 |
| VSIBench (Yang et al., 2025a) | 256 | 128 | 16 |

# D    MORE RESULTS ON LONG-FORM VIDEO BENCHMARKS

Table 6: LongTimeScope results across baselines. Metrics include overall accuracy and task-specific scores across different steps. Flash Attn stands for Qwen2.5-VL-7B (Qwen et al., 2025) accelerated by Flash Infer, and Flash Attn + SFT stands for our fine-tuning version.

| Method | Overall | 18000 | | | 28800 | | | 36000 | | |
|---|---|---|---|---|---|---|---|---|---|---|
| | | OCR | QA | Temporal | OCR | QA | Temporal | OCR | QA | Temporal |
| Flash Attn | 40.7 | 54.0 | 42.0 | 22.0 | 48.0 | 60.0 | 24.0 | 48.0 | 58.0 | 10.0 |
| Flash Attn + SFT | 40.2 | 46.0 | 30.0 | 34.0 | 46.0 | 44.0 | 36.0 | 52.0 | 44.0 | 20.0 |
| AWQ | – | – | – | – | – | – | – | – | – | – |
| XAttn | 41.1 | 52.0 | 56.0 | 30.0 | 54.0 | 52.0 | 6.0 | 52.0 | 64.0 | 4.0 |
| MInference | 44.4 | 64.0 | 56.0 | 26.0 | 58.0 | 60.0 | 8.0 | 56.0 | 66.0 | 6.0 |
| tri-shape | 28.4 | 34.0 | 36.0 | 12.0 | 48.0 | 48.0 | 0.0 | 44.0 | 32.0 | 2.0 |
| FlexPrefill | 39.1 | 52.0 | 46.0 | 24.0 | 46.0 | 56.0 | 14.0 | 46.0 | 66.0 | 2.0 |
| FastV | 35.6 | 36.0 | 50.0 | 16.0 | 44.0 | 50.0 | 4.0 | 44.0 | 64.0 | 12.0 |
| VisionZip | 31.1 | 38.0 | 32.0 | 14.0 | 56.0 | 46.0 | 0.0 | 44.0 | 46.0 | 4.0 |
| VScan | 40.4 | 48.0 | 52.0 | 24.0 | 50.0 | 52.0 | 22.0 | 46.0 | 64.0 | 6.0 |
| VideoNSA | 44.4 | 50.0 | 54.0 | 30.0 | 54.0 | 72.0 | 0.0 | 48.0 | 76.0 | 16.0 |

Table 7: LongVideoBench results across baselines. Metrics include overall accuracy and task-specific scores across different steps. Flash Attn stands for Qwen2.5-VL-7B (Qwen et al., 2025) accelerated by Flash Infer, and Flash Attn + SFT stands for our fine-tuning version.

| Method | Overall | 600 | TOS | S2E | E3E | S2A | SAA | O3O | T3O | T3E | O2E | T2O | S2O | TAA | T2E | E2O | SSS | T2A | 60 | SOS | 15 | 3600 |
|---|---|---|---|---|---|---|---|---|---|---|---|---|---|---|---|---|---|---|---|---|---|---|
| Flash Attn | 58.7 | 58.5 | 38.4 | 69.9 | 66.0 | 70.5 | 56.9 | 57.6 | 58.1 | 53.4 | 63.2 | 63.2 | 55.6 | 52.4 | 63.1 | 63.1 | 39.2 | 64.6 | 72.7 | 61.7 | 65.6 | 52.3 |
| Flash Attn + SFT | 57.8 | 55.8 | 38.4 | 65.6 | 61.7 | 73.9 | 56.9 | 65.2 | 55.4 | 57.5 | 57.5 | 56.6 | 62.5 | 51.2 | 55.4 | 66.2 | 39.2 | 63.3 | 74.4 | 58.0 | 64.6 | 52.0 |
| AWQ | 59.0 | 60.0 | 34.2 | 72.0 | 64.9 | 68.2 | 59.7 | 57.6 | 52.7 | 50.7 | 69.0 | 52.6 | 63.9 | 57.3 | 61.5 | 67.7 | 43.3 | 62.0 | 73.8 | 63.0 | 67.7 | 50.9 |
| XAttn | 59.1 | 59.4 | 36.0 | 70.0 | 66.0 | 67.2 | 57.3 | 58.1 | 55.8 | 53.8 | 64.5 | 62.2 | 64.3 | 56.3 | 65.2 | 66.7 | 41.3 | 58.5 | 75.2 | 60.7 | 68.8 | 50.6 |
| MInference | 59.2 | 60.6 | 34.6 | 74.3 | 66.0 | 68.3 | 58.7 | 56.6 | 53.1 | 52.4 | 68.0 | 56.9 | 60.1 | 58.8 | 60.5 | 65.2 | 39.2 | 63.6 | 74.6 | 66.9 | 67.3 | 50.8 |
| tri-shape | 59.5 | 60.9 | 34.6 | 73.2 | 66.0 | 69.5 | 58.7 | 58.1 | 55.8 | 52.4 | 68.0 | 55.6 | 61.5 | 60.0 | 60.5 | 66.7 | 38.2 | 63.6 | 74.6 | 66.9 | 67.8 | 51.1 |
| FlexPrefill | 58.4 | 61.7 | 31.5 | 65.6 | 62.8 | 71.6 | 59.7 | 59.1 | 58.1 | 52.1 | 65.5 | 51.3 | 62.5 | 48.8 | 61.5 | 72.3 | 42.3 | 63.3 | 71.5 | 65.4 | 58.2 | 52.1 |
| FastV | 57.3 | 57.3 | 43.8 | 64.5 | 60.6 | 70.5 | 52.8 | 56.1 | 52.7 | 48.0 | 59.8 | 67.1 | 56.9 | 48.8 | 67.7 | 66.2 | 40.2 | 58.2 | 69.8 | 61.7 | 70.9 | 48.9 |
| VisionZip | 52.4 | 53.2 | 32.9 | 63.4 | 66.0 | 58.0 | 54.2 | 50.0 | 51.4 | 42.5 | 57.5 | 47.4 | 58.3 | 45.1 | 56.9 | 61.5 | 30.9 | 51.9 | 62.2 | 61.7 | 58.2 | 46.8 |
| VScan | 58.7 | 57.0 | 29.5 | 69.0 | 65.0 | 69.6 | 56.3 | 54.1 | 56.1 | 55.5 | 61.2 | 58.5 | 61.9 | 60.2 | 58.0 | 73.4 | 41.4 | 61.3 | 74.2 | 65.9 | 73.7 | 50.3 |
| VideoNSA | 60.2 | 59.9 | 48.1 | 65.1 | 67.6 | 74.1 | 55.6 | 55.5 | 58.4 | 56.3 | 62.2 | 57.0 | 63.9 | 53.3 | 56.2 | 71.6 | 35.9 | 62.7 | 67.5 | 72.4 | 66.3 | 55.1 |

We take LongVideoBench (Wu et al., 2024a), LongTimeScope (Zohar et al., 2025), MLVU (Zhou et al., 2024), and TimeScope (Zohar et al., 2025) as representative long-video benchmarks and compare against existing token compression and sparse attention methods. As shown in Table 6, Table 7, Table 8, and Table 9, VideoNSA achieves comparable performance without specialized designs. Moreover, we observe that VideoNSA significantly outperforms the baselines on subtasks related to temporal reasoning and on videos of extended length.

Table 8: MLVU results across baselines. Metrics include overall accuracy and task-specific scores across different steps. Flash Attn stands for Qwen2.5-VL-7B (Qwen et al., 2025) accelerated by Flash Infer, and Flash Attn + SFT stands for our fine-tuning version.

| Method | Overall | PlotQA | Needle | Ego | Count | Order | Anomaly Reco | Topic Reason. | SportsQA | TutorialQA |
|---|---|---|---|---|---|---|---|---|---|---|
| Flash Attn | 51.2 | 58.0 | 68.3 | 52.8 | 31.7 | 25.7 | 46.2 | 79.1 | 38.9 | 48.8 |
| Flash Attn + SFT | 51.2 | 58.0 | 58.3 | 58.5 | 23.3 | 40.0 | 43.6 | 81.3 | 36.1 | 37.2 |
| AWQ | 46.0 | 42.7 | 53.0 | 40.9 | 27.2 | 50.2 | 57.0 | 65.0 | 38.3 | 39.2 |
| XAttn | 50.2 | 60.0 | 64.7 | 56.5 | 28.0 | 29.4 | 41.6 | 74.9 | 39.7 | 39.9 |
| MInference | 49.2 | 56.0 | 64.7 | 48.9 | 29.7 | 26.6 | 41.6 | 77.1 | 39.7 | 39.9 |
| tri-shape | 49.2 | 56.0 | 64.7 | 48.9 | 29.7 | 26.6 | 41.6 | 77.1 | 39.7 | 39.9 |
| FlexPrefill | 46.0 | 54.0 | 54.7 | 42.7 | 24.7 | 40.9 | 36.6 | 72.6 | 29.7 | 32.2 |
| FastV | 41.8 | 44.0 | 45.0 | 47.2 | 18.3 | 30.0 | 46.2 | 84.6 | 28.6 | 32.2 |
| VisionZip | 33.1 | 30.0 | 26.7 | 30.2 | 6.7 | 22.9 | 41.0 | 68.1 | 19.7 | 26.4 |
| VScan | 48.1 | 58.0 | 63.3 | 50.9 | 28.3 | 24.3 | 43.6 | 78.0 | 47.2 | 39.5 |
| VideoNSA | 51.8 | 48.0 | 69.3 | 51.3 | 27.7 | 34.6 | 44.5 | 86.2 | 47.7 | 31.6 |

Table 9: TimeScope results across baselines. Metrics include overall accuracy and task-specific scores across different steps. Flash Attn stands for Qwen2.5-VL-7B (Qwen et al., 2025) accelerated by Flash Infer, and Flash Attn + SFT stands for our fine-tuning version.

| Method | Overall | 60 | 120 | 180 | 300 | 600 | 1200 | 1800 | 3600 | 7200 | 10800 |
|---|---|---|---|---|---|---|---|---|---|---|---|
| Flash Attn | 81.0 | 96.7 | 96.0 | 96.0 | 94.7 | 94.0 | 88.0 | 82.0 | 68.7 | 52.7 | 41.3 |
| Flash Attn + SFT | 76.8 | 96.7 | 96.7 | 96.0 | 95.3 | 90.7 | 78.0 | 78.0 | 54.7 | 41.3 | 40.7 |
| AWQ | – | – | – | – | – | – | – | – | – | – | – |
| XAttn | 83.1 | 94.0 | 93.4 | 93.4 | 92.0 | 92.7 | 89.4 | 82.7 | 72.7 | 70.7 | 50.7 |
| MInference | 82.7 | 93.4 | 94.0 | 93.4 | 92.0 | 92.7 | 87.4 | 80.0 | 74.0 | 70.0 | 50.0 |
| tri-shape | 82.7 | 93.4 | 94.0 | 93.4 | 92.0 | 92.7 | 87.4 | 80.0 | 74.0 | 70.0 | 50.0 |
| FlexPrefill | 83.0 | 96.7 | 96.0 | 96.7 | 95.3 | 96.0 | 95.3 | 86.0 | 77.3 | 55.3 | 35.3 |
| FastV | 46.5 | 82.7 | 76.0 | 74.0 | 54.0 | 32.7 | 32.7 | 29.3 | 29.3 | 34.0 | 20.0 |
| VisionZip | 43.5 | 92.0 | 66.7 | 60.0 | 43.3 | 35.3 | 26.0 | 30.7 | 29.3 | 28.0 | 23.3 |
| VScan | 80.3 | 96.7 | 96.7 | 96.0 | 93.3 | 92.7 | 89.3 | 81.3 | 60.0 | 55.3 | 41.3 |
| VideoNSA | 83.7 | 96.7 | 96.0 | 97.4 | 92.0 | 85.4 | 91.6 | 89.3 | 73.3 | 63.3 | 52.0 |

# E   MORE RESULTS ON TEMPORAL REASONING BENCHMARKS

We take Tomato (Shangguan et al., 2024) as the representative temporal reasoning benchmark and compare against existing token compression and sparse attention methods. As shown in Table 10, VideoNSAachieves comparable performance without specialized designs. Moreover, we observe that VideoNSA significantly outperforms the baselines on subtasks including object counting, shape description, and human actions.

# F   MORE RESULTS ON SPATIAL UNDERSTANDING BENCHMARKS

We take VSIBench (Yang et al., 2025a) as the representative spatial understanding benchmark and compare against existing token compression and sparse attention methods. As shown in Table 11, VideoNSA achieves comparable performance without specialized designs. Moreover, we observe that VideoNSA significantly outperforms the baselines on subtasks including object relative direction, route planning, and object size estimation.

# G   RESULTS ON ADDITIONAL VIDEO UNDERSTANDING BENCHMARKS

We conduct additional experiments on LSDBench (Qu et al., 2025) and VideoEvalPro (Ma et al., 2025) to compare VideoNSA and other training-free sparse attention baselines, demonstrating the consistent advantage of VideoNSA in multiple video understanding tasks.

Table 10: Tomato results across baselines. Metrics include overall accuracy and task-specific scores across different steps. Flash Attn stands for Qwen2.5-VL-7B (Qwen et al., 2025) accelerated by Flash Infer, and Flash Attn + SFT stands for our fine-tuning version.

| Method | Overall | Direction | Count | Rotation | Shape & Trend | Vel. & Freq. | Visual Cues | Human | Simulated | Object |
|---|---|---|---|---|---|---|---|---|---|---|
| Flash Attn | 22.6 | 23.6 | 23.3 | 16.1 | 22.9 | 21.9 | 42.9 | 18.0 | 19.7 | 27.9 |
| Flash Attn + SFT | 21.7 | 19.6 | 23.3 | 18.2 | 26.0 | 18.1 | 38.6 | 18.8 | 18.0 | 25.6 |
| XAttn | 21.4 | 22.1 | 22.9 | 19.6 | 17.9 | 17.1 | 42.9 | 15.5 | 21.5 | 26.8 |
| MInference | 23.0 | 22.6 | 27.1 | 18.9 | 22.0 | 20.0 | 37.1 | 16.6 | 20.6 | 29.6 |
| FlexPrefill | 23.7 | 23.3 | 25.0 | 22.7 | 22.0 | 21.4 | 35.7 | 17.1 | 22.7 | 29.9 |
| FastV | 21.6 | 20.6 | 26.0 | 20.3 | 23.3 | 12.7 | – | 17.1 | 24.2 | 25.6 |
| VisionZip | 19.1 | 17.6 | 16.8 | 21.0 | 19.3 | 19.0 | 30.0 | 14.8 | 21.5 | 22.3 |
| VScan | 23.6 | 25.3 | 21.9 | 19.9 | 24.2 | 20.5 | 42.9 | 18.7 | 21.9 | 28.7 |
| VideoNSA | 26.5 | 21.6 | 31.5 | 22.0 | 25.6 | 23.3 | 40.0 | 21.7 | 23.6 | 29.3 |

Table 11: VSIBench results across baselines. Metrics include overall accuracy and task-specific scores across different steps. Flash Attn stands for Qwen2.5-VL-7B (Qwen et al., 2025) accelerated by Flash Infer, and Flash Attn + SFT stands for our fine-tuning version.

| Method | Overall | Obj. Order | Abs. Dist. | Counting | Rel. Dist. | Size Est. | Room Est. | Route Plan. | Rel. Dir. |
|---|---|---|---|---|---|---|---|---|---|
| Flash Attn | 29.7 | 25.7 | 16.0 | 20.5 | 34.7 | 49.5 | 22.5 | 30.4 | 38.5 |
| Flash Attn + SFT | 30.6 | 31.9 | 14.2 | 12.3 | 40.4 | 46.6 | 30.4 | 30.9 | 37.8 |
| AWQ | – | – | – | – | – | – | – | – | – |
| XAttn | 35.0 | 32.7 | 18.1 | 39.7 | 37.6 | 52.1 | 30.0 | 32.5 | 37.4 |
| MInference | 36.6 | 36.5 | 18.2 | 43.9 | 39.4 | 48.5 | 38.8 | 30.0 | 37.7 |
| tri-shape | 36.5 | 35.7 | 18.2 | 44.3 | 39.8 | 48.6 | 38.8 | 29.0 | 37.7 |
| FlexPrefill | 34.9 | 34.1 | 21.6 | 35.1 | 39.3 | 51.8 | 29.7 | 30.4 | 36.8 |
| FastV | 34.0 | 31.7 | 21.7 | 26.1 | 36.2 | 47.8 | 35.0 | 33.5 | 40.1 |
| VisionZip | 32.1 | 28.8 | 17.9 | 28.8 | 36.5 | 48.9 | 26.9 | 29.4 | 39.3 |
| VScan | 34.4 | 33.0 | 21.9 | 33.0 | 40.0 | 51.9 | 28.5 | 30.4 | 36.6 |
| VideoNSA | 36.0 | 25.5 | 19.0 | 42.5 | 35.4 | 54.0 | 30.1 | 37.5 | 43.6 |

## H VISUALIZATION OF ATTENTION PATTERN IN EACH BRANCH

We visualize the attention patterns of the last layer across the three branches in Figure 10, Figure 11, Figure 12, and Figure 13, together with the final attention output, as representative examples. The compression branch reduces redundancy to preserve salient information, the selection branch highlights task-relevant regions with sparse activations, and the sliding window branch enforces local temporal coverage by focusing on short-range dependencies. These complementary roles collectively shape the final attention output.

## I MORE RESULTS ON BRANCH COMBINATION

In this section, we report detailed results of different branch combinations across three domains, including long video understanding (Table 14, Tavke 15, Table 16, and Table 17), temporal reasoning (Table 18), and spatial understanding (Table 19). The corresponding performances are summarized in the table, which highlights how the use of individual branches or their combinations affects downstream tasks.

## J MORE RESULTS ON INFORMATION SCALING STUDY

Figure 16 shows the scaling performance of VideoNSA under different context allocation strategies on LongTimeScope and MLVU. Both benchmarks were trained with a maximum context length of 32K tokens, yet their performance consistently improves when scaled to 64K, beyond the training budget. On LongTimeScope (Zohar et al., 2025), the best results emerge around 512 frames with 128 TPF at 64K tokens, underscoring the dataset's reliance on extended temporal coverage for long-horizon reasoning. In contrast, MLVU (Zhou et al., 2024) also peaks at 64K with the same allocation, but its contours are smoother, and competitive performance extends across a broader range of frame–token trade-offs. This suggests that while LongTimeScope demands aggressive temporal scaling, MLVU benefits from a more balanced distribution of temporal and spatial information.

Table 12: Results on LSDBench (Qu et al., 2025).

| Model | Accuracy |
|---|---|
| LongVA (Zhang et al., 2024b) | 32.5 |
| LongVila (Chen et al., 2024b) | 49.8 |
| InternVL2.5 (Chen et al., 2024c) | 50.1 |
| Qwen2.5-VL-7B (Qwen et al., 2025) | 52.2 |
| Qwen2.5-VL-7B-SFT | 52.5 |
| *Sparse Attention Methods* | |
| + Tri-Shape (Li et al., 2024c) | 49.5 |
| + MInference (Jiang et al., 2024) | 49.5 |
| + FlexPrefill (Lai et al., 2025) | 52.3 |
| + XAttention (Xu et al., 2025a) | 51.3 |
| **VideoNSA** | **55.2** |

Table 13: Results on VideoEvalPro (Ma et al., 2025). HP stands for Holistic Perception, HR stands for Holistic Reasoning, LR stands for Local Reasoning, LP stands for Local Perception.

| Model | HP | HR | LR | LP | Overall |
|---|---|---|---|---|---|
| LongVA (Zhang et al., 2024b) | 20.5 | 6.8 | 19.0 | 9.5 | 16.5 |
| Video-XL (Shu et al., 2025) | 22.3 | 15.0 | 18.2 | 10.2 | 18.6 |
| InternVL2.5 (Chen et al., 2024c) | 28.8 | 19.7 | 21.5 | 16.7 | 24.6 |
| Qwen2.5-VL-7B (Qwen et al., 2025) | 33.9 | 15.6 | 24.8 | 17.8 | 27.7 |
| Qwen2.5-VL-7B-SFT | 34.5 | 15.8 | 25.3 | 18.2 | 28.3 |
| *Sparse Attention Methods* | | | | | |
| + Tri-Shape (Li et al., 2024c) | 34.1 | 16.3 | 25.1 | 20.0 | 28.4 |
| + MInference (Jiang et al., 2024) | 32.3 | **17.1** | **27.7** | 16.7 | 26.0 |
| + FlexPrefill (Lai et al., 2025) | 33.0 | 15.9 | 26.3 | 19.8 | 28.3 |
| + XAttention (Xu et al., 2025a) | 34.5 | 16.6 | 25.6 | **20.5** | 28.9 |
| **VideoNSA** | **35.4** | 16.9 | 26.3 | 19.1 | **29.4** |

Table 14: LongVideoBranch results across different branch selection strategy. Metrics include overall accuracy and task-specific scores across different steps.

| Method | Overall | 600 | TOS | S2E | E3E | S2A | SAA | O3O | T3O | T3E | O2E | T2O | S2O | TAA | T2E | E2O | SSS | T2A | 60 | SOS | 15 | 3600 |
|---|---|---|---|---|---|---|---|---|---|---|---|---|---|---|---|---|---|---|---|---|---|---|
| VideoNSA + Test SFT | 56.1 | 57.0 | 46.6 | 59.1 | 61.7 | 69.3 | 56.9 | 63.6 | 52.7 | 50.7 | 56.3 | 59.2 | 59.7 | 43.9 | 55.4 | 64.6 | 38.1 | 58.2 | 70.4 | 60.5 | 65.1 | 48.1 |
| NSA-CMP | 48.1 | 50.5 | 38.4 | 51.6 | 56.4 | 53.4 | 50.0 | 45.5 | 51.4 | 41.1 | 54.0 | 42.1 | 43.1 | 45.1 | 47.7 | 53.9 | 26.8 | 53.2 | 55.2 | 64.2 | 47.6 | 44.3 |
| NSA-SLC | 48.4 | 49.0 | 32.9 | 61.3 | 59.6 | 58.0 | 52.8 | 48.5 | 46.0 | 43.8 | 52.9 | 36.8 | 47.2 | 42.7 | 44.6 | 55.4 | 33.0 | 48.1 | 53.5 | 55.6 | 50.3 | 45.7 |
| NSA-SWA | 49.1 | 50.7 | 37.0 | 52.7 | 56.4 | 59.1 | 51.4 | 48.5 | 43.2 | 45.2 | 55.2 | 42.1 | 48.6 | 45.1 | 46.2 | 61.5 | 30.9 | 45.6 | 54.1 | 65.4 | 48.7 | 46.5 |
| NSA-CMPSLC | 49.4 | 49.5 | 34.3 | 55.9 | 61.7 | 58.0 | 56.9 | 48.5 | 47.3 | 41.1 | 56.3 | 35.5 | 52.8 | 47.6 | 46.2 | 55.4 | 34.0 | 41.8 | 54.1 | 63.0 | 48.2 | 48.2 |
| NSA-SLCSWA | 49.3 | 48.8 | 32.9 | 58.1 | 61.7 | 55.7 | 52.8 | 47.0 | 46.0 | 46.6 | 54.0 | 34.2 | 48.6 | 47.6 | 47.7 | 54.0 | 35.1 | 48.1 | 54.1 | 64.2 | 49.2 | 48.2 |
| NSA-CMPSWA | 48.8 | 49.3 | 34.3 | 53.8 | 59.6 | 54.6 | 52.8 | 50.0 | 48.7 | 42.5 | 57.5 | 40.8 | 51.4 | 42.7 | 46.2 | 55.4 | 29.9 | 45.6 | 57.6 | 64.2 | 48.7 | 45.9 |

In addition to the overall scaling trends, we further report detailed subtask-level results under different allocation settings in Table 20, Table 21, Table 22, Table 23, Table 24, and Table 25.

## K    ADDITIONAL CONTEXT-LENGTH SCALING RESULTS OF QWEN2.5-VL

We include Table 26 and Table 27 to further illustrate the long-context behavior of the base model. Since Qwen2.5-VL 7B (Qwen et al., 2025) has a maximum context window of 128k, its modeling ability tends to become less stable when approaching this upper bound. As shown in Figure K,

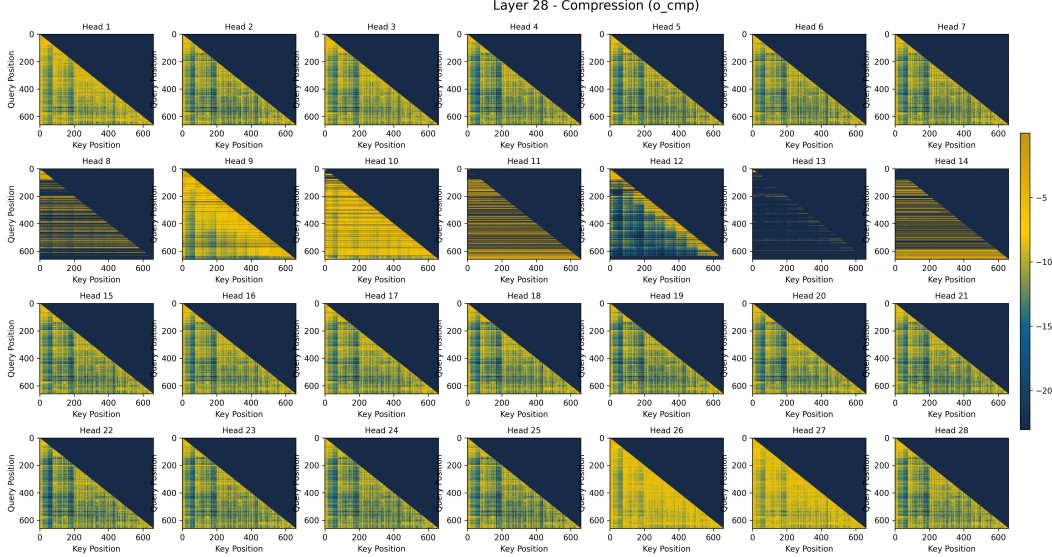

Figure 10: Attention pattern of the compression branch in the final layer of VideoNSA.

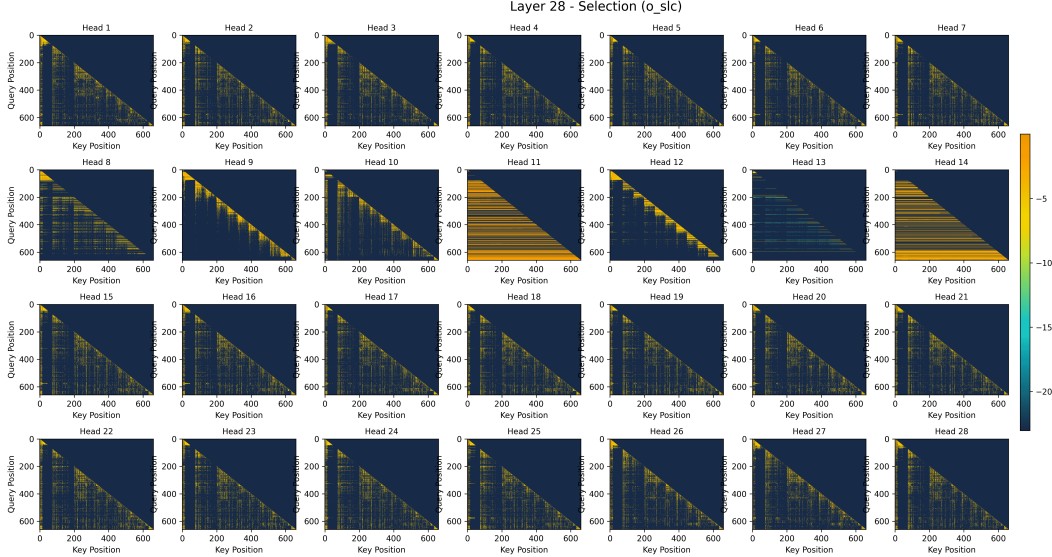

Figure 11: Attention pattern of the selection branch in the final layer of VideoNSA.

Qwen2.5-VL (Qwen et al., 2025) often peaks at 64k and slightly declines at 128k across several benchmarks. In contrast, VideoNSA maintains stable or stronger performance at 128k, demonstrating that the observed 64k > 128k phenomenon arises from backbone limitations rather than the proposed sparse architecture.

## L   MORE RESULTS ON ATTENTION SCALING STUDY

Figure 16 evaluates the scaling behavior of VideoNSA under different attention allocation strategies, where the x-axis denotes the sliding window size (log scale), the y-axis shows the block count, and the size and color of each marker reflect performance, with the dashed blue curve indicating configurations of equal attention budget and arrows marking the training setting as well as reduced-budget configurations (3.6% and 1.8%); on LongVideoBench, performance peaks near the training configuration and degrades when allocating excessive budget to local attention through larger sliding

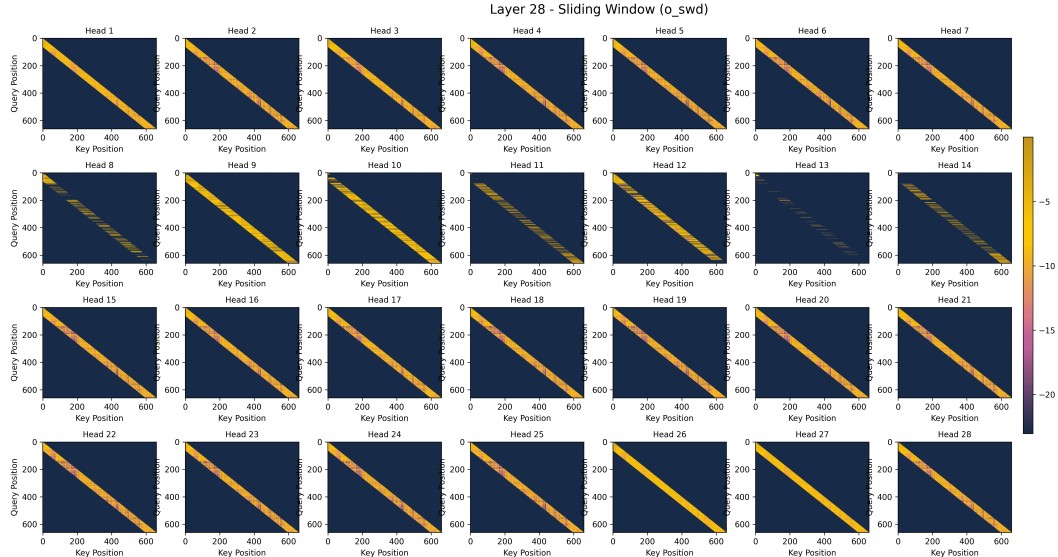

Figure 12: Attention pattern of the sliding window branch in the final layer of VideoNSA.

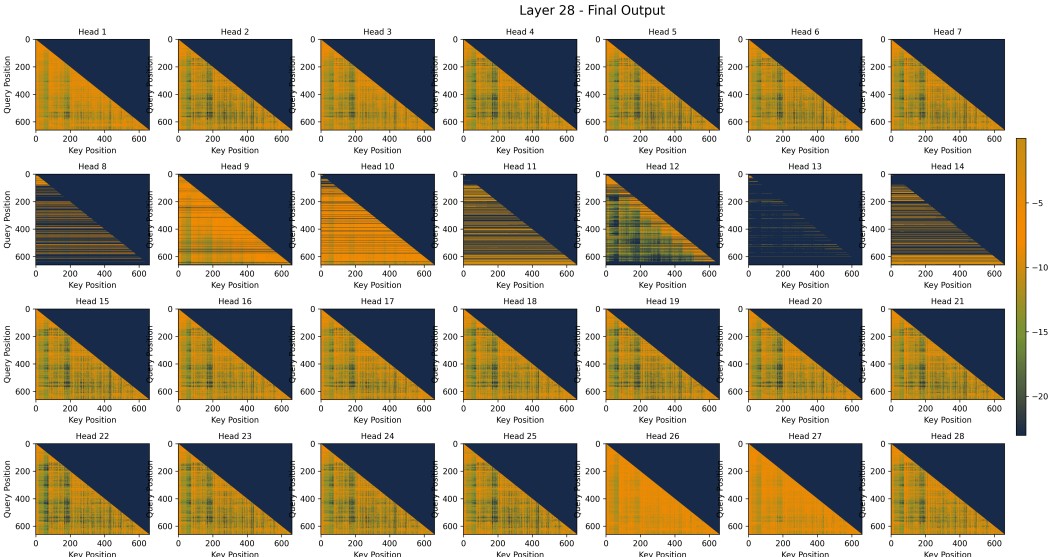

Figure 13: Attention pattern of the final vision attention output in the final layer of VideoNSA.

windows, while the best configuration achieves strong results with only 3.6% of the full budget, and on TimeScope, performance is even more sensitive, with larger sliding windows quickly reducing accuracy whereas maintaining more global blocks yields superior outcomes, and overall the results confirm that training allocations are well balanced, that prioritizing global attention is consistently more effective than enlarging local windows under equal budget, and that VideoNSA sustains leading performance with as little as 3.6% or less of the full attention cost, demonstrating both efficiency and hardware awareness.

In addition to the overall scaling trends, we further report detailed subtask-level results under different allocation settings in Table 20, Table 21, Table 22, Table 23, Table 24, and Table 25.

Table 15: LongTimeScope results across different branch selection strategy. Metrics include overall accuracy and task-specific scores across different steps.

| Method | Overall | 18000 | | | 28800 | | | 36000 | | |
|---|---|---|---|---|---|---|---|---|---|---|
| | | OCR | QA | Temporal | OCR | QA | Temporal | OCR | QA | Temporal |
| VideoNSA + Test SFT | 40.9 | 52.0 | 42.0 | 42.0 | 48.0 | 62.0 | 18.0 | 42.0 | 50.0 | 12.0 |
| NSA-CMP | 25.1 | 20.0 | 22.0 | 24.0 | 38.0 | 40.0 | 0.0 | 34.0 | 34.0 | 14.0 |
| NSA-SLC | 37.1 | 30.0 | 38.0 | 40.0 | 50.0 | 58.0 | 12.0 | 42.0 | 44.0 | 20.0 |
| NSA-SWA | 29.8 | 34.0 | 34.0 | 22.0 | 36.0 | 46.0 | 4.0 | 34.0 | 46.0 | 12.0 |
| NSA-CMPSLC | 32.4 | 36.0 | 34.0 | 24.0 | 46.0 | 54.0 | 8.0 | 42.0 | 36.0 | 12.0 |
| NSA-SLCSWA | 34.4 | 38.0 | 36.0 | 36.0 | 46.0 | 56.0 | 8.0 | 38.0 | 36.0 | 16.0 |
| NSA-CMPSWA | 31.6 | 30.0 | 38.0 | 20.0 | 40.0 | 52.0 | 16.0 | 36.0 | 36.0 | 16.0 |
| VideoNSA | 44.4 | 50.0 | 54.0 | 30.0 | 54.0 | 72.0 | 0.0 | 48.0 | 76.0 | 16.0 |

Table 16: TimeScope results across different branch selection strategy. Metrics include overall accuracy and task-specific scores across different steps.

| Method | Overall | 60 | 120 | 180 | 300 | 600 | 1200 | 1800 | 3600 | 7200 | 10800 |
|---|---|---|---|---|---|---|---|---|---|---|---|
| Full Attn | 81.0 | 96.7 | 96.0 | 96.0 | 94.7 | 94.0 | 88.0 | 82.0 | 68.7 | 52.7 | 41.3 |
| Flash Attn | 81.0 | 96.7 | 96.0 | 96.0 | 94.7 | 94.0 | 88.0 | 82.0 | 68.7 | 52.7 | 41.3 |
| Flash Attn + SFT | 76.8 | 96.7 | 96.7 | 96.0 | 95.3 | 90.7 | 78.0 | 78.0 | 54.7 | 41.3 | 40.7 |
| AWQ | – | – | – | – | – | – | – | – | – | – | – |
| XAttn | 83.1 | 94.0 | 93.4 | 93.4 | 92.0 | 92.7 | 89.4 | 82.7 | 72.7 | 70.7 | 50.7 |
| MInference | 82.7 | 93.4 | 94.0 | 93.4 | 92.0 | 92.7 | 87.4 | 80.0 | 74.0 | 70.0 | 50.0 |
| tri-shape | 82.7 | 93.4 | 94.0 | 93.4 | 92.0 | 92.7 | 87.4 | 80.0 | 74.0 | 70.0 | 50.0 |
| FlexPrefill | 83.0 | 96.7 | 96.0 | 96.7 | 95.3 | 96.0 | 95.3 | 86.0 | 77.3 | 55.3 | 35.3 |
| FastV | 46.5 | 82.7 | 76.0 | 74.0 | 54.0 | 32.7 | 32.7 | 29.3 | 29.3 | 34.0 | 20.0 |
| VisionZip | 43.5 | 92.0 | 66.7 | 60.0 | 43.3 | 35.3 | 26.0 | 30.7 | 29.3 | 28.0 | 23.3 |
| VScan | 80.3 | 96.7 | 96.7 | 96.0 | 93.3 | 92.7 | 89.3 | 81.3 | 60.0 | 55.3 | 41.3 |
| Retake | – | – | – | – | – | – | – | – | – | – | – |
| AdaRetake | – | – | – | – | – | – | – | – | – | – | – |
| SFT + Test NSA | 81.0 | 96.7 | 96.0 | 96.0 | 94.7 | 94.0 | 88.0 | 82.0 | 68.7 | 52.7 | 41.3 |
| NSA + Test SFT | 83.0 | 96.7 | 95.3 | 94.0 | 93.3 | 94.0 | 90.7 | 87.3 | 76.7 | 54.7 | 47.3 |
| NSA-CMP | 41.5 | 82.0 | 74.0 | 65.3 | 59.3 | 17.3 | 25.3 | 19.3 | 26.7 | 27.3 | 18.0 |
| NSA-SLC | 63.7 | 92.0 | 86.0 | 86.7 | 78.0 | 66.7 | 57.3 | 51.3 | 40.7 | 38.0 | 40.0 |
| NSA-SWA | 59.3 | – | – | – | – | – | – | – | – | – | – |
| NSA-CMPSLC | 57.3 | 88.7 | 80.0 | 73.3 | 73.3 | 46.7 | 44.7 | 48.7 | 42.7 | 43.3 | 32.0 |
| NSA-SLCSWA | 65.2 | 92.0 | 89.3 | 89.3 | 79.3 | 66.0 | 59.3 | 50.0 | 41.3 | 40.7 | 44.7 |
| NSA-CMPSWA | 57.3 | 88.7 | 80.0 | 73.3 | 73.3 | 46.7 | 44.7 | 48.7 | 42.7 | 43.3 | 32.0 |
| VideoNSA | 83.7 | 96.7 | 96.0 | 97.4 | 92.0 | 85.4 | 91.6 | 89.3 | 73.3 | 63.3 | 52.0 |

## M  THEORETICAL FOUNDATIONS OF SCALING BEHAVIOR

In Section 4, we perform two scaling experiments along context length and attention budget. We observe that VideoNSA exhibits strong extrapolation ability on context length: although trained with only 36K tokens, it can generalize to 128K at test time, achieving the best performance at 64K. In contrast, when scaling the attention budget, even a small reduction to 3.6% of attention computation already delivers outstanding performance, and further increasing the visible-token count does not yield additional gains. To clarify these phenomena, we provide theoretical interpretations from routing-path stability and the geometric structure of RoPE (Su et al., 2024).

**Routing-path Stability.**  Recent work (Huang et al., 2025) indicates that a model's ability to maintain performance on long sequences depends critically on the stability of its attention routing structure across positions. In the standard attention mechanism, the attention weight from the query

Table 17: MLVU results across different branch selection strategy. Metrics include overall accuracy and task-specific scores across different steps.

| Method | Overall | PlotQA | Needle | Ego | Count | Order | Anomaly Reco | Topic Reason. | SportsQA | TutorialQA |
|--------|---------|--------|--------|-----|-------|-------|--------------|---------------|----------|------------|
| NSA + Test SFT | 51.6 | 56.0 | 61.7 | 66.0 | 31.7 | 28.6 | 51.3 | 80.2 | 36.1 | 32.6 |
| NSA-CMP | 43.9 | 36.0 | 35.0 | 42.9 | – | 24.3 | 30.8 | 80.2 | 30.6 | – |
| NSA-SLC | 47.7 | 50.0 | 50.0 | 52.4 | – | 22.9 | 33.3 | 74.7 | 33.3 | – |
| NSA-SWA | 40.2 | 40.0 | 40.0 | 41.5 | 15.0 | 24.3 | 30.8 | 76.9 | 36.1 | 34.9 |
| NSA-SLCSWA | 42.4 | 42.0 | 48.3 | 45.3 | 16.7 | 25.7 | 38.5 | 75.8 | 33.3 | 34.9 |
| NSA-CMPSWA | 43.4 | 46.0 | 40.0 | 43.4 | 18.3 | 35.7 | 33.3 | 82.4 | 27.8 | 32.6 |

Table 18: Tomato results across different branch selection strategy. Metrics include overall accuracy and task-specific scores across different steps.

| Method | Overall | Direction | Count | Rotation | Shape & Trend | Vel. & Freq. | Visual Cues | Human | Simulated | Object |
|--------|---------|-----------|-------|----------|---------------|--------------|-------------|-------|-----------|--------|
| NSA + Test SFT | 23.4 | 21.3 | 29.1 | 17.5 | 25.1 | 20.0 | 40.0 | 19.3 | 19.3 | 28.5 |
| NSA-CMP | 23.3 | 22.1 | 29.5 | 17.1 | 24.7 | 20.5 | 34.3 | 19.2 | 22.7 | 27.3 |
| NSA-SLC | 24.0 | 21.3 | 32.2 | 16.4 | 26.0 | 22.9 | 32.9 | 19.8 | 21.5 | 28.7 |
| NSA-SWA | 24.0 | 21.3 | 32.2 | 16.4 | 26.0 | 22.9 | 32.9 | 19.8 | 21.5 | 28.7 |
| NSA-CMPSLC | 23.5 | 20.8 | 29.8 | 18.5 | 22.9 | 23.8 | 34.3 | 19.0 | 26.8 | 25.8 |
| NSA-SLCSWA | 23.0 | 20.6 | 27.4 | 18.5 | 22.4 | 24.8 | 32.9 | 19.5 | 21.0 | 26.8 |
| NSA-CMPSWA | 24.5 | 23.1 | 30.8 | 18.9 | 25.1 | 22.9 | 32.9 | 21.0 | 23.6 | 28.0 |

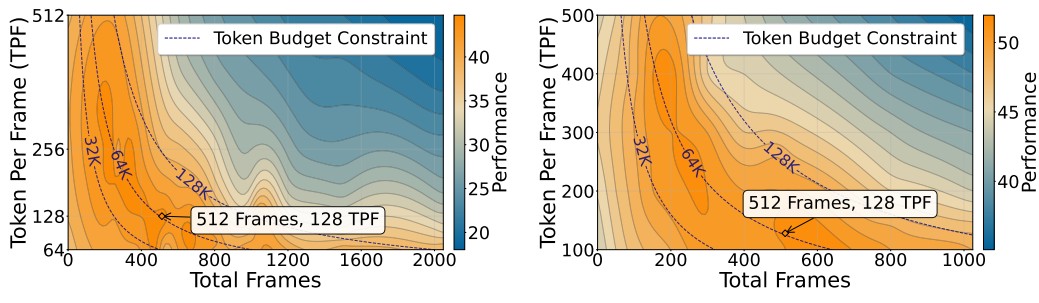

(a) Information Scaling of LongTimeScope  (b) Information Scaling of MLVU

Figure 14: Scaling Performance of VideoNSA under Different Context Allocation Strategies. We highlight the token budget constraint to indicate settings with equal context length, and annotate the best-performing configuration under each benchmark.

vector $Q_n$ at position $n$ to the key vector $Z_j$ at position $j$ is defined as

$$\text{Attn}_{n \to j} = \frac{\exp(Z_j^\top Q_n)}{\sum_k \exp(Z_k^\top Q_n)}.$$

Here, $Q_n$ and $Z_j$ denote the query and key representations at positions $n$ and $j$, respectively. If the model can consistently focus its attention on the task-relevant target token set $\mathcal{T}$ during inference, then (i) $\sum_{j \in \mathcal{T}} \text{Attn}_{n \to j}$ should dominate across different positions; and (ii) the attention assigned to the same key token $j$ should remain nearly unchanged under positional shifts, i.e.,

$$\Delta_i = \big|\text{Attn}_{n \to j} - \text{Attn}_{n+i \to j}\big| \approx 0,$$

where $\Delta_i$ measures the deviation of routing paths across positions in long sequences.

When we scale the context length using dense temporal and spatial sampling, the sparse-attention pattern and mask structure $M$ remain unchanged, which means the model continues to use the routing structure learned during training while simply facing a larger pool of candidate evidence. Since denser sampling mainly introduces redundant or finer-grained details, the model treats these tokens as auxiliary evidence, leaving the core target tokens and their relative attention weights essentially unchanged. Consequently, the overall routing-path structure is preserved, $\Delta_i$ remains small, and the model can maintain or even improve its performance at longer context lengths.

In contrast, attention-budget scaling explicitly modifies the set of visible tokens in the sparse-attention mechanism by replacing the original mask $M$ with a new mask $M'$. The effective query

Table 19: VSIBench results across different branch selection strategy. Metrics include overall accuracy and task-specific scores across different steps.

| Method | Overall | Obj. Order | Abs. Dist. | Counting | Rel. Dist. | Size Est. | Room Est. | Route Plan. | Rel. Dir. |
|---|---|---|---|---|---|---|---|---|---|
| NSA + Test SFT | 33.1 | 24.3 | 19.8 | 31.2 | 38.0 | 49.8 | 32.2 | 32.5 | 37.2 |
| NSA-CMP | 29.2 | 19.9 | 16.3 | 12.6 | 29.3 | 48.7 | 26.7 | 38.1 | 41.7 |
| NSA-SLC | 27.6 | 18.0 | 10.9 | 17.3 | 32.0 | 47.8 | 24.8 | 32.0 | 38.1 |
| NSA-SWA | 29.8 | 22.8 | 15.6 | 17.4 | 32.3 | 49.8 | 27.2 | 33.5 | 39.4 |
| NSA-CMPSLC | 29.4 | 19.9 | 16.3 | 15.1 | 31.0 | 51.1 | 25.5 | 33.5 | 42.6 |
| NSA-SLCSWA | 29.1 | 19.9 | 12.2 | 18.5 | 31.4 | 49.6 | 26.5 | 34.0 | 40.4 |
| NSA-CMPSWA | 30.3 | 22.5 | 15.6 | 15.3 | 31.1 | 52.5 | 26.7 | 35.1 | 43.3 |

Table 20: Ablation study results on information scaling of LongTimeScope (Zohar et al., 2025). Metrics include overall accuracy and task-specific scores across different steps. # TPF stands for token per frame, and # F stands for sampling frame number.

| # TPF | # F | Overall | 18000 | | | 28800 | | | 36000 | | |
|---|---|---|---|---|---|---|---|---|---|---|---|
| | | | OCR | QA | Temporal | OCR | QA | Temporal | OCR | QA | Temporal |
| 256 | 128 | 42.9 | 54.0 | 48.0 | 36.0 | 46.0 | 62.0 | 6.0 | 40.0 | 80.0 | 14.0 |
| 512 | 128 | 41.1 | 54.0 | 60.0 | 28.0 | 42.0 | 62.0 | 4.0 | 40.0 | 78.0 | 2.0 |
| 128 | 256 | 42.0 | 58.0 | 56.0 | 26.0 | 46.0 | 62.0 | 2.0 | 40.0 | 78.0 | 10.0 |
| 256 | 256 | 41.3 | 58.0 | 52.0 | 36.0 | 48.0 | 62.0 | 0.0 | 40.0 | 70.0 | 6.0 |
| 512 | 256 | 41.6 | 54.0 | 56.0 | 32.0 | 46.0 | 60.0 | 2.0 | 40.0 | 78.0 | 6.0 |
| 64 | 512 | 40.2 | 52.0 | 52.0 | 26.0 | 44.0 | 64.0 | 2.0 | 44.0 | 76.0 | 2.0 |
| 128 | 512 | 44.4 | 50.0 | 54.0 | 30.0 | 54.0 | 72.0 | 0.0 | 48.0 | 76.0 | 16.0 |
| 256 | 512 | 38.7 | 48.0 | 50.0 | 30.0 | 52.0 | 56.0 | 8.0 | 36.0 | 60.0 | 8.0 |
| 64 | 1024 | 41.6 | 54.0 | 56.0 | 22.0 | 46.0 | 66.0 | 4.0 | 36.0 | 72.0 | 18.0 |
| 128 | 1024 | 41.1 | 50.0 | 46.0 | 32.0 | 46.0 | 62.0 | 14.0 | 38.0 | 54.0 | 28.0 |
| 64 | 2048 | 38.4 | 50.0 | 62.0 | 26.0 | 40.0 | 60.0 | 2.0 | 38.0 | 42.0 | 26.0 |

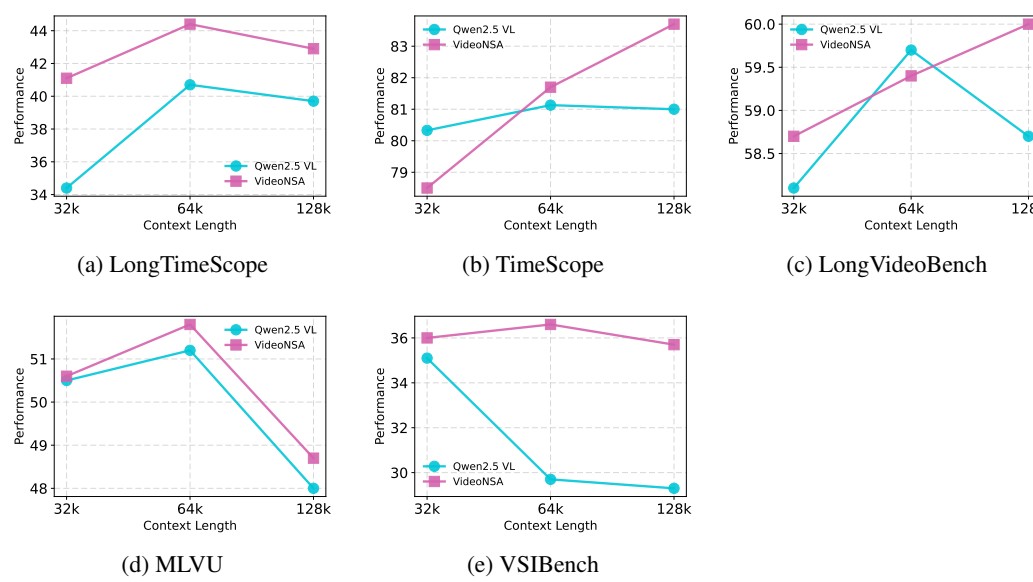

(a) LongTimeScope  (b) TimeScope  (c) LongVideoBench

(d) MLVU  (e) VSIBench

Figure 15: Performance comparison of Qwen2.5-VL and VideoNSA under different context lengths.

becomes
$$Q_{\text{eff}} = Q \odot M',$$
where $\odot$ denotes elementwise multiplication, and the corresponding new attention weight is
$$\text{Attn}'_{n \to j} \propto \exp\left(Z_j^\top (Q \odot M')\right).$$

Table 21: Ablation study results on information scaling of TimeScope (Zohar et al., 2025). Metrics include overall accuracy and task-specific scores across different steps. # TPF stands for token per frame, and # F stands for sampling frame number.

| # TPF | # F | Overall | 60 | 120 | 180 | 300 | 600 | 1200 | 1800 | 3600 | 7200 | 10800 |
|---|---|---|---|---|---|---|---|---|---|---|---|---|
| 256 | 128 | 73.1 | 96.7 | 94.7 | 93.4 | 85.4 | 72.0 | 62.6 | 57.3 | 56.6 | 54.0 | 58.6 |
| 512 | 128 | 72.5 | 95.4 | 94.0 | 92.7 | 82.0 | 72.7 | 64.6 | 57.3 | 56.3 | 53.3 | 56.6 |
| 128 | 256 | 76.5 | 98.0 | 97.4 | 96.0 | 86.7 | 78.7 | 78.0 | 63.3 | 58.6 | 54.0 | 54.0 |
| 256 | 256 | 76.1 | 96.7 | 96.7 | 91.4 | 86.7 | 76.0 | 74.6 | 64.0 | 62.0 | 56.0 | 56.6 |
| 512 | 256 | 75.8 | 95.4 | 94.7 | 90.7 | 86.7 | 76.0 | 75.3 | 66.0 | 62.6 | 53.3 | 57.3 |
| 64 | 512 | 78.5 | 96.7 | 95.4 | 94.7 | 88.0 | 80.7 | 82.6 | 71.3 | 62.0 | 58.0 | 55.3 |
| 128 | 512 | 76.5 | 98.0 | 97.4 | 96.0 | 86.7 | 78.7 | 78.0 | 63.3 | 58.6 | 54.0 | 54.0 |
| 256 | 512 | 77.3 | 96.7 | 96.7 | 90.7 | 83.4 | 76.7 | 78.0 | 72.0 | 66.6 | 59.3 | 52.6 |
| 64 | 1024 | 81.7 | 96.7 | 95.4 | 94.7 | 90.0 | 84.7 | 88.0 | 78.0 | 69.3 | 64.0 | 56.6 |
| 128 | 1024 | 81.8 | 98.0 | 97.4 | 94.0 | 85.4 | 80.7 | 92.0 | 78.0 | 72.6 | 66.0 | 54.0 |
| 64 | 2048 | 82.7 | 96.7 | 95.4 | 94.7 | 90.0 | 82.0 | 91.3 | 88.0 | 73.3 | 63.3 | 52.0 |

Table 22: Ablation study results on information scaling of LongVideoBench (Wu et al., 2024a). Metrics include overall accuracy and task-specific scores across different steps. # TPF stands for token per frame, and # F stands for sampling frame number.

| # TPF | # F | Overall | 600.0 | TOS | S2E | E3E | S2A | SAA | O3O | T3O | T3E | O2E | T2O | S2O | TAA | T2E | E2O | SSS | T2A | 60.0 | SOS | 15.0 | 3600.0 |
|---|---|---|---|---|---|---|---|---|---|---|---|---|---|---|---|---|---|---|---|---|---|---|---|
| 512 | 64 | 58.3 | 55.5 | 44.0 | 67.2 | 66.6 | 71.8 | 57.0 | 54.0 | 48.9 | 50.8 | 59.9 | 55.7 | 63.9 | 53.3 | 57.8 | 70.1 | 37.9 | 60.2 | 68.1 | 71.2 | 65.7 | 54.9 |
| 128 | 128 | 57.4 | 56.8 | 45.4 | 66.1 | 66.6 | 69.5 | 61.2 | 55.5 | 53.0 | 49.5 | 59.9 | 51.7 | 54.2 | 47.2 | 57.8 | 68.5 | 40.0 | 58.9 | 68.6 | 70.0 | 63.6 | 52.4 |
| 256 | 128 | 57.9 | 58.5 | 48.1 | 69.4 | 66.6 | 70.6 | 58.4 | 55.5 | 53.0 | 48.1 | 57.6 | 47.8 | 54.2 | 49.7 | 59.3 | 68.5 | 40.0 | 65.2 | 68.1 | 70.0 | 63.1 | 52.6 |
| 512 | 128 | 59.0 | 59.4 | 49.5 | 68.3 | 66.6 | 72.9 | 63.9 | 54.0 | 53.0 | 49.5 | 61.0 | 51.7 | 61.2 | 50.9 | 56.2 | 68.5 | 39.0 | 64.0 | 68.6 | 71.2 | 63.6 | 54.2 |
| 128 | 256 | 58.7 | 52.7 | 46.7 | 68.3 | 65.5 | 69.5 | 57.0 | 52.5 | 53.0 | 48.1 | 56.4 | 46.5 | 51.4 | 48.5 | 57.8 | 67.0 | 37.9 | 65.2 | 63.4 | 70.0 | 58.3 | 52.7 |
| 256 | 256 | 58.2 | 58.7 | 39.9 | 64.0 | 66.6 | 71.8 | 58.4 | 54.0 | 59.7 | 52.2 | 59.9 | 55.7 | 58.4 | 50.9 | 56.2 | 70.1 | 40.0 | 61.4 | 66.9 | 68.7 | 63.1 | 53.5 |
| 512 | 256 | 59.4 | 60.4 | 52.2 | 67.2 | 65.5 | 75.2 | 61.2 | 54.0 | 55.7 | 52.2 | 62.2 | 53.1 | 62.6 | 49.7 | 56.2 | 68.5 | 35.9 | 65.2 | 67.5 | 72.4 | 65.7 | 54.0 |
| 64 | 512 | 57.7 | 58.2 | 41.3 | 67.2 | 68.7 | 65.0 | 58.4 | 58.5 | 54.3 | 52.2 | 62.2 | 49.1 | 58.4 | 53.3 | 62.4 | 71.6 | 35.9 | 55.1 | 66.3 | 68.7 | 61.5 | 53.5 |
| 128 | 512 | 58.5 | 59.4 | 42.6 | 68.3 | 66.6 | 69.5 | 59.8 | 60.0 | 57.0 | 52.2 | 64.5 | 50.4 | 59.8 | 52.1 | 59.3 | 68.5 | 35.9 | 60.2 | 65.7 | 68.7 | 63.6 | 54.0 |
| 256 | 512 | 58.3 | 59.2 | 44.0 | 64.0 | 64.5 | 71.8 | 65.3 | 52.5 | 55.7 | 55.0 | 61.0 | 54.4 | 62.6 | 49.7 | 59.3 | 71.6 | 37.9 | 56.4 | 66.9 | 67.5 | 63.1 | 53.5 |
| 64 | 1024 | 58.4 | 59.4 | 42.6 | 65.1 | 68.7 | 66.1 | 62.6 | 55.5 | 58.4 | 49.5 | 64.5 | 54.4 | 58.4 | 50.9 | 59.3 | 74.7 | 36.9 | 57.6 | 66.3 | 68.7 | 61.5 | 54.2 |
| 128 | 1024 | 58.7 | 58.5 | 41.3 | 67.2 | 68.7 | 71.8 | 65.3 | 60.0 | 59.7 | 52.2 | 59.9 | 53.1 | 62.6 | 47.2 | 59.3 | 71.6 | 32.8 | 60.2 | 65.7 | 67.5 | 63.6 | 55.1 |

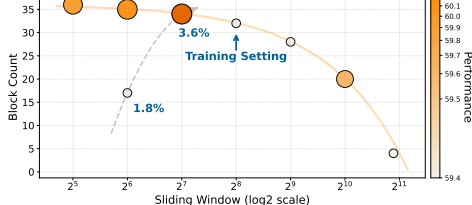
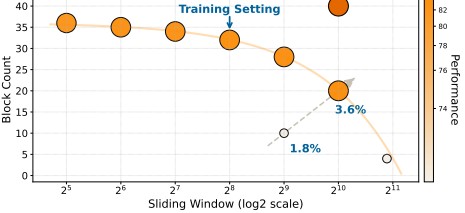

(a) Attention Scaling of LongVideoBench     (b) Attention Scaling of TimeScope

Figure 16: Scaling Performance of VideoNSA under Different Attention Allocation Strategies. We highlight the attention budget constraint to indicate settings with equal attention budget, and annotate the best-performing configuration under each benchmark.

Even if the modification from $M$ to $M'$ appears small in proportion, it substantially changes the set of candidate evidence accessible to each query and alters the relative logits. First, the newly visible tokens reduce the relative weight allocated to the original key tokens, producing a dilution effect. Second, in video tasks, the added visible tokens often lie in similar visual-semantic clusters as the original key tokens and thus have non-negligible similarity scores $Z_{j'}^{\top}Q$. These tokens directly compete with and divert attention away from the originally dominant key tokens. Under the combined influence of these effects, the stable routing-path structure learned during training is overwritten: formerly high-weight key tokens may be diluted or overshadowed by the new candidates, causing $\Delta_i$ to increase significantly. As a result, the model is more likely to follow incorrect reasoning paths, leading to degraded performance.

Table 23: Ablation study results on information scaling of MLVU (Zhou et al., 2024). Metrics include overall accuracy and task-specific scores across different steps. # TPF stands for token per frame, and # F stands for sampling frame number.

| # TPF | # F | Overall | PlotQA | Needle | Ego | Count | Order | Anomaly Reco | Topic Reason. | SportsQA | TutorialQA |
|---|---|---|---|---|---|---|---|---|---|---|---|
| 256 | 128 | 49.6 | 46.0 | 52.7 | 53.2 | 24.3 | 36.0 | 47.0 | 87.3 | 36.6 | 36.2 |
| 512 | 128 | 49.2 | 52.0 | 51.0 | 47.5 | 24.3 | 37.4 | 39.3 | 87.3 | 42.1 | 33.9 |
| 128 | 256 | 50.6 | 50.0 | 57.7 | 60.7 | 24.3 | 33.1 | 39.3 | 86.2 | 42.1 | 36.2 |
| 256 | 256 | 51.2 | 50.0 | 56.0 | 56.9 | 27.7 | 38.9 | 41.9 | 85.1 | 42.1 | 36.2 |
| 512 | 256 | 48.0 | 54.0 | 49.3 | 49.4 | 22.7 | 37.4 | 39.3 | 86.2 | 33.8 | 29.3 |
| 64 | 512 | 51.2 | 50.0 | 62.7 | 55.1 | 24.3 | 34.6 | 47.0 | 84.0 | 42.1 | 38.6 |
| 128 | 512 | 51.8 | 48.0 | 69.3 | 51.3 | 27.7 | 34.6 | 44.5 | 86.2 | 47.7 | 31.6 |
| 256 | 512 | 48.6 | 50.0 | 51.0 | 47.5 | 24.3 | 33.1 | 52.2 | 84.0 | 47.7 | 26.9 |
| 64 | 1024 | 51.8 | 56.0 | 66.0 | 53.2 | 26.0 | 36.0 | 47.0 | 84.0 | 42.1 | 31.6 |
| 128 | 1024 | 48.0 | 52.0 | 51.0 | 49.4 | 29.3 | 33.1 | 44.5 | 80.7 | 44.9 | 24.6 |

Table 24: Ablation study results on information scaling of Tomato (Shangguan et al., 2024). Metrics include overall accuracy and task-specific scores across different steps. # TPF stands for token per frame, and # F stands for sampling frame number.

| FPS | TPF | Overall | Direction | Count | Rotation | Shape&Trend | Velocity&Freq. | Visual Cues | Human | Simulated | Object |
|---|---|---|---|---|---|---|---|---|---|---|---|
| 1 | 64 | 24.7 | 22.8 | 26.7 | 20.6 | 25.1 | 27.1 | 34.3 | 21.2 | 23.2 | 28.4 |
| 1 | 128 | 23.9 | 20.6 | 29.8 | 19.9 | 24.7 | 23.8 | 32.9 | 19.8 | 24.0 | 27.6 |
| 1 | 256 | 24.7 | 22.3 | 29.5 | 19.6 | 25.1 | 25.2 | 35.7 | 20.8 | 23.3 | 28.7 |
| 1 | 512 | 23.9 | 20.6 | 29.8 | 19.9 | 24.7 | 22.9 | 34.3 | 20.3 | 21.5 | 27.9 |
| 2 | 64 | 24.5 | 21.1 | 31.8 | 19.6 | 22.4 | 25.7 | 35.7 | 20.7 | 21.9 | 28.8 |
| 2 | 128 | 24.3 | 20.6 | 30.5 | 20.3 | 24.7 | 23.3 | 38.6 | 20.7 | 22.3 | 28.4 |
| 2 | 256 | 24.4 | 21.3 | 29.5 | 18.5 | 26.5 | 24.8 | 37.1 | 20.3 | 24.0 | 28.2 |
| 2 | 512 | 24.7 | 19.4 | 32.2 | 21.3 | 25.6 | 23.8 | 37.1 | 20.0 | 24.0 | 29.1 |
| 4 | 64 | 25.1 | 22.1 | 31.5 | 19.6 | 26.5 | 23.3 | 38.6 | 21.2 | 25.0 | 29.0 |
| 4 | 128 | 25.8 | 21.8 | 33.2 | 21.3 | 25.6 | 25.7 | 37.1 | 21.5 | 25.3 | 29.9 |
| 4 | 256 | 26.2 | 23.1 | 32.5 | 20.6 | 26.9 | 26.7 | 37.1 | 21.8 | 25.3 | 30.5 |
| 4 | 512 | 26.5 | 21.6 | 31.5 | 22.0 | 25.6 | 23.3 | 40.0 | 21.7 | 23.6 | 29.3 |

**Geometric Rotational of RoPE.** RoPE (Su et al., 2024) maps the representation at position $i$ into a rotation in a two-dimensional subspace:

$$q'(i) = R(i\omega)q, \qquad k'(j) = R(j\omega)k,$$

where $R(i\omega)$ is a rotation matrix and $\omega$ denotes the frequency parameters. This yields an inner product that depends only on the relative distance between the two positions:

$$\langle q'(i), k'(j) \rangle = \langle R((i-j)\omega)q, k \rangle.$$

RoPE (Su et al., 2024) therefore establishes a structured geometric correspondence between relative distance and rotation phase. Under this geometry, when the context length is moderately increased (e.g., from 36K to 64K), the model only needs to resolve a larger phase difference $d\omega$; within this range, the growth of the phase still lies in the extrapolation regime covered by the empirical distribution seen during training. As a result, the model can naturally generalize.

LM-Infinite (Han et al., 2023) further proves that, in order to distinguish the growing clusters of relative distances $\alpha(n)$, the attention logit must increase monotonically with sequence length:

$$\sup_{q,k,d \leq n} |w(q,k,d)| \geq \left( \frac{\alpha(n)}{2} \right)^{1/(2r)} \frac{\varepsilon}{4e},$$

where $w(q,k,d)$ denotes the logit at relative distance $d$, and $\alpha(n)$ grows with $n$. This "logit growth" is controlled and beneficial at moderate lengths, expanding the dynamic range of attention and enabling the model to maintain token separability over larger distances and consistent with the strong performance we observe around 64K.

However, when the effective phase difference $d\omega$ becomes excessively large, the rotation angle may approach or exceed the periodic range of multiple frequency dimensions, giving rise to *phase aliasing*: tokens that should correspond to distinct relative distances collapse into similar or even indistinguishable phase regions. In such cases, although attention logits continue to grow with length,

Table 25: Ablation study results on information scaling of VSIBench (Yang et al., 2025a). Metrics include overall accuracy and task-specific scores across different steps. # TPF stands for token per frame, and # F stands for sampling frame number.

| TPF | # Max Frames | Overall | Obj. Order | Abs. Dist. | Counting | Rel. Dist. | Size Est. | Room Est. | Route Plan. | Rel. Dir. |
|---|---|---|---|---|---|---|---|---|---|---|
| 512 | 32 | 34.9 | 27.6 | 16.2 | 31.4 | 35.1 | 52.2 | 31.5 | 40.1 | 44.6 |
| 512 | 64 | 34.8 | 29.1 | 17.5 | 34.9 | 33.0 | 52.2 | 31.0 | 36.5 | 43.9 |
| 256 | 128 | 36.0 | 24.7 | 17.6 | 41.3 | 37.5 | 53.9 | 30.7 | 39.1 | 43.3 |
| 512 | 128 | 34.6 | 27.4 | 17.2 | 37.3 | 34.4 | 50.3 | 30.6 | 35.5 | 43.8 |
| 128 | 256 | 35.6 | 26.8 | 17.0 | 42.0 | 36.8 | 51.8 | 31.2 | 35.0 | 44.2 |
| 256 | 256 | 35.5 | 27.8 | 17.0 | 42.4 | 33.7 | 51.3 | 31.6 | 35.5 | 44.5 |
| 512 | 256 | 34.8 | 28.2 | 16.5 | 40.3 | 33.9 | 48.8 | 30.7 | 36.5 | 43.3 |
| 64 | 512 | 34.2 | 29.1 | 15.8 | 42.1 | 33.9 | 45.5 | 27.7 | 37.0 | 42.9 |
| 128 | 512 | 36.0 | 25.5 | 19.0 | 42.5 | 35.4 | 54.0 | 30.1 | 37.5 | 43.6 |
| 256 | 512 | 33.9 | 28.2 | 15.8 | 42.9 | 31.3 | 43.6 | 29.9 | 36.5 | 43.1 |
| 64 | 1024 | 35.8 | 24.4 | 18.5 | 46.4 | 34.4 | 52.4 | 29.7 | 37.5 | 43.0 |
| 128 | 1024 | 35.7 | 26.6 | 18.4 | 45.3 | 32.7 | 50.3 | 31.7 | 37.0 | 43.7 |

Table 26: Performance of Qwen2.5-VL 7B under different context lengths.

| Context | LVB | MLVU | TimeScope | LTS | VSIBench |
|---|---|---|---|---|---|
| 32k | 58.1 | 50.5 | 80.33 | 34.4 | 35.1 |
| 64k | 59.7 | 51.2 | 81.13 | 40.7 | 29.7 |
| 128k | 58.7 | 48.0 | 81.00 | 39.7 | 29.3 |

the high-frequency components of RoPE lose their discriminative resolution, reducing geometric separability among tokens, which aligns with existing analyses (Press et al., 2021; Chen et al., 2023) showing the degradation of relative positional encoding at extreme distances.

## N  FULL GATE VALUES DISTRIBUTION

## O  MORE INTER-HEAD GATE SIMILARITES VISUALIZATION

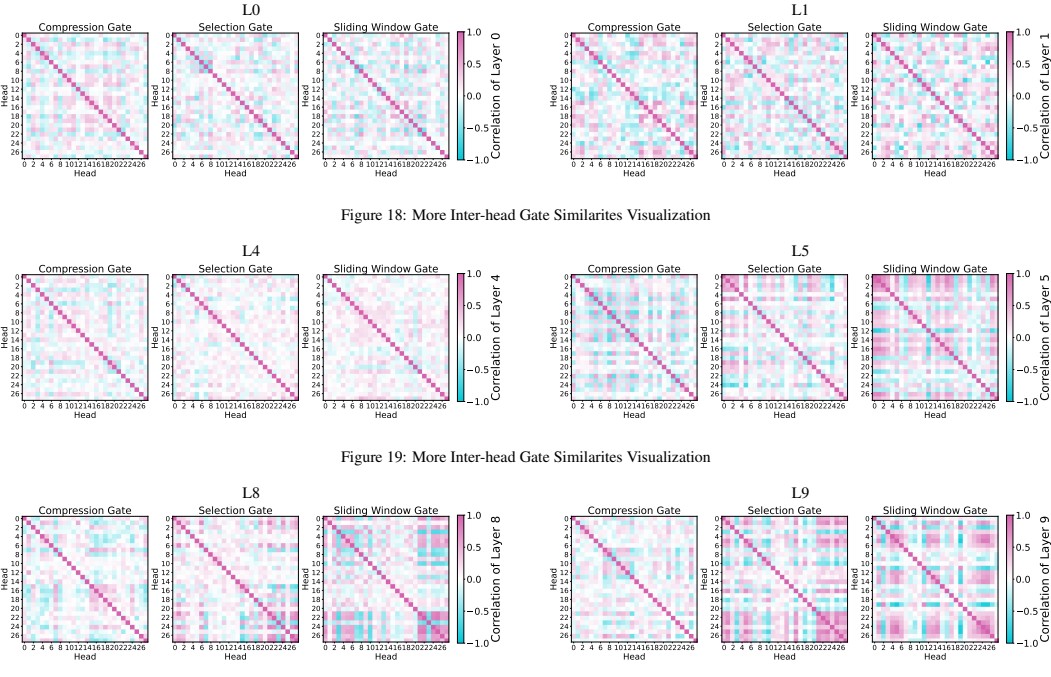

Figure 18: More Inter-head Gate Similarites Visualization

Figure 19: More Inter-head Gate Similarites Visualization

Figure 20: More Inter-head Gate Similarites Visualization

Table 27: Performance of VideoNSA under different context lengths.

| Context | LVB | MLVU | TimeScope | LTS | VSIBench |
|---------|------|------|-----------|------|----------|
| 32k | 58.7 | 50.6 | 78.5 | 41.1 | 36.0 |
| 64k | 59.4 | 51.8 | 81.7 | 44.4 | 36.6 |
| 128k | 60.0 | 48.7 | 83.7 | 42.9 | 35.7 |

Table 28: LongTimeScope (Zohar et al., 2025) results across different attention budget strategy. Metrics include overall accuracy and task-specific scores across different steps.

| Block Count | Window Size | Overall | 18000 | | | 28800 | | | 36000 | | |
|-------------|-------------|---------|------|------|----------|------|------|----------|------|------|----------|
| | | | OCR | QA | Temporal | OCR | QA | Temporal | OCR | QA | Temporal |
| 36 | 32 | 44.0 | 56.0 | 50.0 | 28.0 | 46.0 | 66.0 | 16.0 | 46.0 | 72.0 | 16.0 |
| 35 | 64 | 44.0 | 54.0 | 58.0 | 26.0 | 46.0 | 68.0 | 12.0 | 46.0 | 74.0 | 12.0 |
| 34 | 128 | 41.8 | 50.0 | 56.0 | 28.0 | 44.0 | 64.0 | 6.0 | 46.0 | 74.0 | 8.0 |
| 28 | 512 | 42.0 | 50.0 | 56.0 | 28.0 | 48.0 | 64.0 | 6.0 | 46.0 | 76.0 | 4.0 |
| 20 | 1024 | 40.9 | 52.0 | 56.0 | 28.0 | 48.0 | 64.0 | 0.0 | 44.0 | 76.0 | 0.0 |
| 4 | 1900 | 0.0 | 0.0 | 0.0 | 0.0 | 0.0 | 0.0 | 0.0 | 0.0 | 0.0 | 0.0 |
| 16 | 128 | 41.6 | 52.0 | 56.0 | 26.0 | 46.0 | 62.0 | 10.0 | 42.0 | 76.0 | 4.0 |
| 64 | 512 | 42.4 | 52.0 | 56.0 | 28.0 | 48.0 | 64.0 | 8.0 | 46.0 | 76.0 | 4.0 |

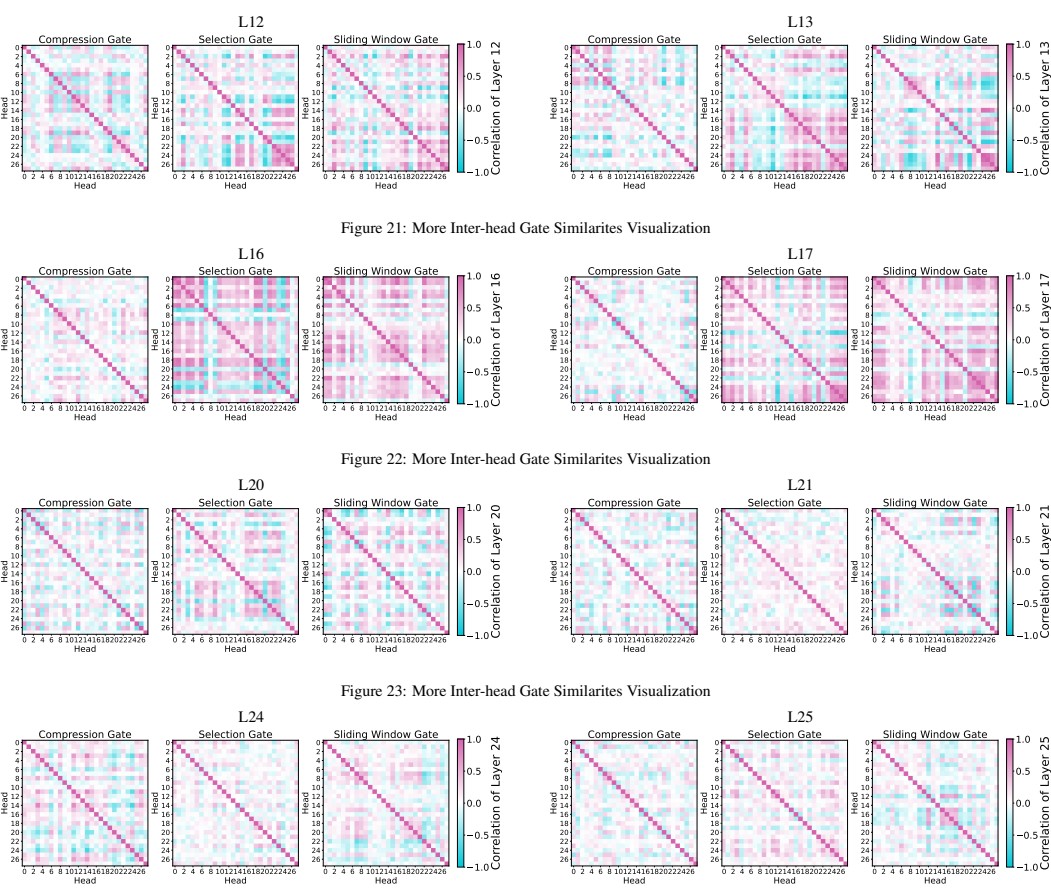

Figure 21: More Inter-head Gate Similarites Visualization

Figure 22: More Inter-head Gate Similarites Visualization

Figure 23: More Inter-head Gate Similarites Visualization

Figure 24: More Inter-head Gate Similarites Visualization

# P  BENCHMARK-LEVEL GATING ANALYSIS AND PCA VISUALIZATION

In this section, we provide additional evidence that VideoNSA's routing strategy depends on input video content rather than layer depth alone. We collect the layer–head gate vectors for representative videos from three benchmarks with distinct visual properties (LongTimeScope (Zohar et al.,

Table 29: TimeScope (Zohar et al., 2025) results across different attention budget strategy. Metrics include overall accuracy and task-specific scores across different steps.

| Block Count | Window Size | Overall | 60 | 120 | 180 | 300 | 600 | 1200 | 1800 | 3600 | 7200 | 10800 |
|---|---|---|---|---|---|---|---|---|---|---|---|---|
| 36 | 32 | 81.6 | 97.4 | 97.4 | 96.7 | 85.4 | 82.7 | 86.3 | 85.3 | 71.3 | 60.6 | 52.7 |
| 35 | 64 | 82.4 | 92.7 | 91.3 | 92.7 | 91.4 | 83.4 | 91.6 | 91.3 | 74.6 | 63.3 | 52.0 |
| 34 | 128 | 82.2 | 96.7 | 96.7 | 94.7 | 89.4 | 82.0 | 88.3 | 85.3 | 76.6 | 60.0 | 52.7 |
| 28 | 512 | 82.8 | 94.7 | 97.4 | 94.7 | 90.7 | 82.7 | 91.6 | 88.6 | 74.0 | 63.3 | 50.0 |
| 20 | 1024 | 83.2 | 96.7 | 97.4 | 97.4 | 88.7 | 85.4 | 89.0 | 84.6 | 78.6 | 58.6 | 55.3 |
| 4 | 1900 | 8.6 | 4.7 | 4.7 | 4.7 | 4.7 | 4.7 | 12.3 | 13.3 | 13.3 | 13.3 | 10.0 |
| 10 | 512 | 59.9 | 4.7 | 4.7 | 94.7 | 88.7 | 79.4 | 85.6 | 80.0 | 68.0 | 58.0 | 35.3 |
| 40 | 1024 | 83.7 | 96.7 | 96.0 | 97.4 | 92.0 | 85.4 | 91.6 | 89.3 | 73.3 | 63.3 | 52.0 |

Table 30: LongVideoBench (Wu et al., 2024a) results across different attention budget strategy. Metrics include overall accuracy and task-specific scores across different steps.

| Block Count | Window Size | Overall | 600.0 | TOS | S2E | E3E | S2A | SAA | O3O | T3O | T3E | O2E | T2O | S2O | TAA | T2E | E2O | SSS | T2A | 60.0 | SOS | 15.0 | 3600.0 |
|---|---|---|---|---|---|---|---|---|---|---|---|---|---|---|---|---|---|---|---|---|---|---|---|
| 36 | 32 | 59.9 | 57.7 | 48.1 | 64.0 | 66.6 | 72.9 | 55.6 | 52.5 | 58.4 | 57.7 | 59.9 | 54.4 | 70.9 | 52.1 | 56.2 | 70.1 | 32.8 | 66.5 | 67.5 | 72.4 | 65.2 | 56.1 |
| 35 | 64 | 60.1 | 58.7 | 48.1 | 64.0 | 65.5 | 75.2 | 57.0 | 55.5 | 59.7 | 59.1 | 58.7 | 55.7 | 66.7 | 49.7 | 56.2 | 70.1 | 34.8 | 65.2 | 66.9 | 73.7 | 65.7 | 55.9 |
| 34 | 128 | 60.2 | 59.9 | 48.1 | 65.1 | 67.6 | 74.1 | 55.6 | 55.5 | 58.4 | 56.3 | 62.2 | 57.0 | 63.9 | 53.3 | 56.2 | 71.6 | 35.9 | 62.7 | 67.5 | 72.4 | 66.3 | 55.1 |
| 28 | 512 | 59.4 | 60.4 | 46.7 | 62.9 | 66.6 | 74.1 | 58.4 | 52.5 | 54.3 | 56.3 | 64.5 | 55.7 | 59.8 | 49.7 | 56.2 | 79.3 | 34.8 | 61.4 | 67.5 | 68.7 | 64.1 | 53.3 |
| 20 | 1024 | 59.6 | 60.6 | 45.4 | 66.1 | 66.6 | 72.9 | 59.8 | 54.0 | 54.3 | 55.0 | 64.5 | 57.0 | 58.4 | 50.9 | 56.2 | 80.9 | 32.8 | 62.7 | 69.2 | 67.5 | 63.6 | 53.1 |
| 4 | 1900 | 28.3 | 27.6 | 23.4 | 24.2 | 35.7 | 25.2 | 32.0 | 29.7 | 27.3 | 24.8 | 31.1 | 30.7 | 33.4 | 35.1 | 27.0 | 28.5 | 26.6 | 19.7 | 27.4 | 28.0 | 27.1 | 29.7 |
| 17 | 64 | 59.4 | 58.0 | 46.7 | 62.9 | 65.5 | 74.1 | 55.6 | 52.5 | 58.4 | 57.7 | 58.7 | 57.0 | 66.7 | 50.9 | 56.2 | 71.6 | 30.7 | 64.0 | 68.1 | 72.4 | 65.7 | 54.2 |

2025) for multi-shot transitions, Tomato (Shangguan et al., 2024) for high-frequency motion, and VSIBench (Yang et al., 2025a) for complex spatial layouts) and project the gate vectors into a 2D space using PCA.

As shown in Figure 25, the gate patterns form three clearly separated clusters, regardless of whether we use the compression branch, the selection branch, or the sliding-window branch, which indicates that VideoNSA learns benchmark-specific routing strategies conditioned on visual content, rather than following a fixed depth pattern.

To further isolate the role of input-driven routing, we replace each layer's gate with a static value averaged from a 1K training subset, forcing the model to depend only on layer depth. As shown in Table 34, the performance drops across all six benchmarks, especially on tasks requiring long-range temporal integration, confirming that dynamic gating is essential.

## Q  ADDITIONAL ANALYSIS OF TRAINING AND INFERENCE EFFICIENCY

To complement the efficiency discussion in the main paper, we provide additional analysis of both FLOPs and wall-clock latency across different attention mechanisms and context lengths.

**Training Efficiency.**  Under identical optimization settings, training VideoNSA requires approximately $4600$ H100 GPU hours, while the dense baseline requires $5280$ H100 GPU hours. This corresponds to $0.87\times$ of the dense baseline, indicating that VideoNSA achieves slightly improved training efficiency despite using a more complex attention mechanism.

**Inference Efficiency.**  Table 35 presents the theoretical FLOPs of different attention mechanisms. In the ideal case, NSA requires only $2.05$ PFLOPs, which is $0.24\times$ that of Flash Attention, demonstrating the theoretical computational efficiency of the sparse routing structure. However, the actual FLOPs and wall-clock latency of VideoNSA are higher than this ideal value due to implementation constraints in the current NSA kernel. The Qwen2.5-VL 7B (Qwen et al., 2025) adopts an unusual head configuration of $4$ KV heads and $28$ query heads. To satisfy Triton kernel requirements, the query heads must be padded to $64$, which introduces additional computation and memory access overhead. As a result, the practical efficiency of VideoNSA deviates from its theoretical FLOPs advantage. As shown in Figure 26, VideoNSA's latency grows much more slowly than dense attention, and compared with other sparse baselines, it delivers competitive inference speed while achieving stronger model performance.

Table 31: MLVU (Zhou et al., 2024) results across different attention budget strategy. Metrics include overall accuracy and task-specific scores across different steps.

| Block Count | Window Size | Overall | Direction | Count | Rotation | Shape&Trend | Velocity&Freq. | Visual Cues | Human | Simulated | Object |
|---|---|---|---|---|---|---|---|---|---|---|---|
| 32 | 256 | 26.5 | 21.6 | 31.5 | 22.0 | 25.6 | 23.3 | 40.0 | 21.7 | 23.6 | 29.3 |
| 36 | 32 | 25.9 | 21.6 | 32.5 | 19.2 | 25.1 | 25.5 | 37.1 | 21.4 | 21.5 | 29.2 |
| 35 | 64 | 27.1 | 23.8 | 33.9 | 20.6 | 25.6 | 25.5 | 37.1 | 22.1 | 24.2 | 30.8 |
| 34 | 128 | 27.2 | 23.8 | 34.2 | 20.3 | 25.1 | 25.5 | 38.6 | 21.9 | 24.2 | 30.8 |
| 28 | 512 | 26.1 | 21.8 | 32.2 | 19.2 | 24.7 | 27.5 | 37.1 | 22.1 | 22.4 | 28.2 |
| 20 | 1024 | 25.1 | 20.6 | 30.8 | 17.5 | 23.3 | 29.4 | 34.3 | 21.4 | 23.3 | 25.6 |
| 64 | 512 | 25.3 | 21.3 | 30.5 | 19.6 | 24.2 | 27.5 | 32.9 | 21.4 | 22.9 | 27.4 |
| 4 | 2048 | 26.4 | 21.8 | 33.6 | 20.3 | 25.6 | 27.5 | 32.9 | 21.8 | 24.7 | 29.4 |
| 16 | 128 | 21.4 | 19.5 | 17.5 | 20.2 | 21.0 | 30.0 | 28.8 | 17.8 | 17.6 | 20.6 |

Table 32: Tomato (Shangguan et al., 2024) results across different attention budget strategy. Metrics include overall accuracy and task-specific scores across different steps.

| Block Count | Window Size | Overall | Direction | Count | Rotation | Shape&Trend | Velocity&Freq. | Visual Cues | Human | Simulated | Object |
|---|---|---|---|---|---|---|---|---|---|---|---|
| 36 | 32 | 25.9 | 21.6 | 32.5 | 19.2 | 25.1 | 25.5 | 37.1 | 21.4 | 21.5 | 29.2 |
| 35 | 64 | 27.1 | 23.8 | 33.9 | 20.6 | 25.6 | 25.5 | 37.1 | 22.1 | 24.2 | 30.8 |
| 34 | 128 | 27.2 | 23.8 | 34.2 | 20.3 | 25.1 | 25.5 | 38.6 | 21.9 | 24.2 | 30.8 |
| 28 | 512 | 26.1 | 21.8 | 32.2 | 19.2 | 24.7 | 27.5 | 37.1 | 22.1 | 22.4 | 28.2 |
| 20 | 1024 | 25.1 | 20.6 | 30.8 | 17.5 | 23.3 | 29.4 | 34.3 | 21.4 | 23.3 | 25.6 |
| 64 | 512 | 25.3 | 21.3 | 30.5 | 19.6 | 24.2 | 27.5 | 32.9 | 21.4 | 22.9 | 27.4 |
| 4 | 2048 | 26.4 | 21.8 | 33.6 | 20.3 | 25.6 | 27.5 | 32.9 | 21.8 | 24.7 | 29.4 |
| 16 | 128 | 21.4 | 19.5 | 17.5 | 20.2 | 21.0 | 30.0 | 28.8 | 17.8 | 17.6 | 20.6 |

# R  ADDITIONAL ANALYSIS ON CMP LATENCY BOTTLENECK

In this section, we provide additional analysis supporting the observation in findings that the CMP branch becomes the dominant source of latency as the context length increases.

Since the block size determines how many CMP operations are executed, we vary the block size and measure the resulting latency across multiple context lengths. As summarized in Table 36, although increasing the block size reduces the number of CMP executions, the overall latency improvement remains small. Smaller blocks are dominated by memory-access overhead, while larger blocks incur higher computation per block. As a result, block-size scaling affects latency only moderately within a narrow range and does not change the overall scaling trend.

We also observe that the most significant acceleration comes from more efficient NSA implementations instead of architectural hyperparameters. As shown in Table 37, the flash-nsa (mdy666, 2025) implementation runs about twice as fast as our current nsa-impl (Pai et al., 2025a) in the forward pass and up to six times faster in the backward pass. Other teams are also developing improved kernels such as optimizing NSA for TPUs (Ko, 2025). These findings show that the dominant factor affecting CMP and overall NSA latency comes from kernel efficiency, including memory access patterns and kernel design.

# S  MORE ANALYSIS ABOUT ATTENTION SINKS ON VARIOUS SPARSE ATTENTION SETTINGS

Figure 27a indicates that in the compression branch, smaller blocks produce sharper and higher sink peaks at the sequence start, while larger blocks used in training reduce the initial peak but introduce broader low-density diffusion with periodic boundary spikes. The selection sinks in 27b remain at consistently low densities under different configurations, suggesting that the top-k filtering mechanism robustly suppresses sink formation across different settings. Figure 27 shows the distribution of attention sinks under different sparse attention settings. When varying the window size, sinks are concentrated near the beginning and decay rapidly with position. Overall, larger windows yield lower sink density but broader coverage, while the training configuration ($w = 256$) strikes a middle ground and exhibits sparse periodic clusters in the mid-to-late sequence, reflecting sensitivity to local boundaries learned during training.

Table 33: VSIBench (Yang et al., 2025a) results across different attention budget strategy. Metrics include overall accuracy and task-specific scores across different steps.

| Block Count | Window Size | Overall | Appearance | Abs. Dist. | Counting | Rel. Dist. | Size Est. | Room Est. | Route Plan. | Rel. Dir. |
|---|---|---|---|---|---|---|---|---|---|---|
| 28 | 512 | 36.0 | 23.9 | 18.2 | 45.5 | 36.9 | 54.1 | 29.8 | 36.0 | 43.4 |
| 20 | 1024 | 35.9 | 24.0 | 18.5 | 46.8 | 36.7 | 53.6 | 29.0 | 36.5 | 42.0 |
| 4 | 1900 | 0.0 | 0.0 | 0.0 | 0.0 | 0.0 | 0.0 | 0.0 | 0.0 | 0.0 |
| 34 | 128 | 35.5 | 24.7 | 18.7 | 37.8 | 36.2 | 53.9 | 31.7 | 36.0 | 45.2 |
| 35 | 64 | 35.4 | 25.5 | 20.4 | 33.2 | 35.4 | 53.6 | 31.5 | 38.1 | 46.0 |
| 36 | 32 | 34.9 | 25.3 | 20.6 | 28.4 | 36.0 | 54.4 | 30.9 | 38.1 | 45.9 |
| 36 | 62 | 35.3 | 25.2 | 20.5 | 28.3 | 35.8 | 54.7 | 31.0 | 41.1 | 46.1 |
| 16 | 128 | 0.0 | 0.0 | 0.0 | 0.0 | 0.0 | 0.0 | 0.0 | 0.0 | 0.0 |
| 64 | 512 | 35.8 | 23.6 | 18.3 | 45.9 | 36.8 | 54.2 | 29.3 | 36.5 | 42.3 |

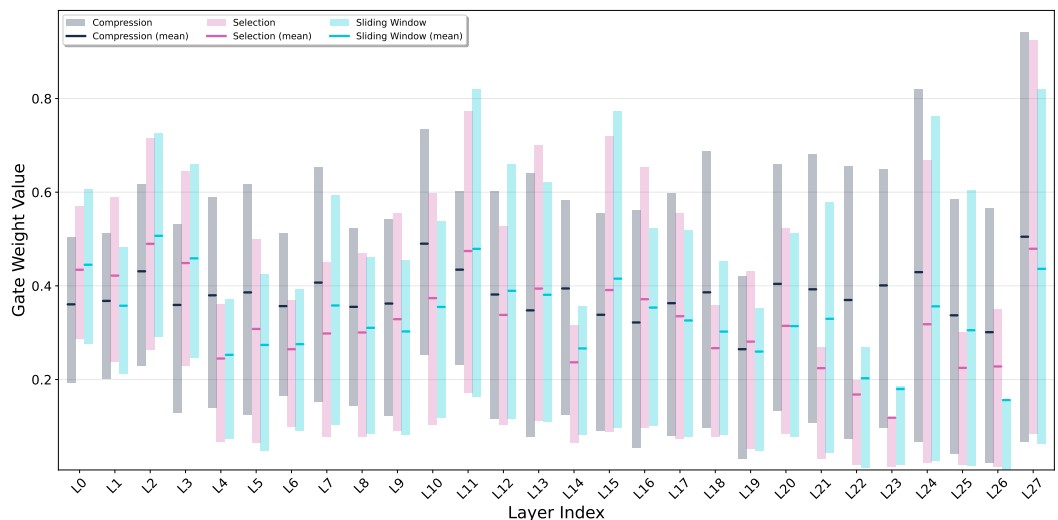

Figure 17: Gate weight distribution of each layer.

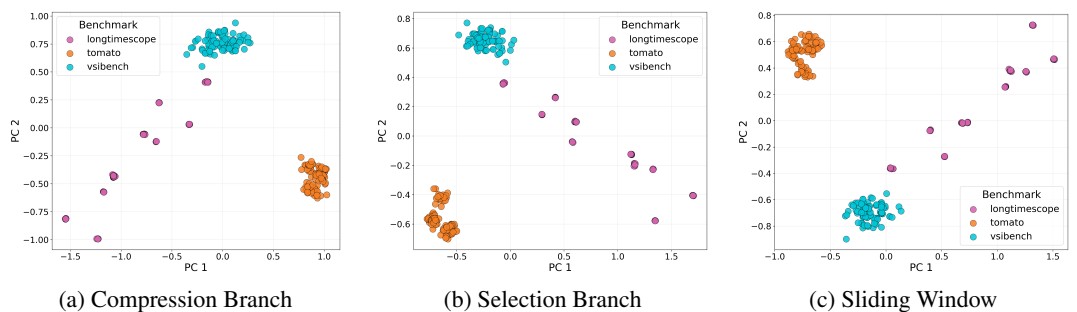

(a) Compression Branch     (b) Selection Branch     (c) Sliding Window

Figure 25: PCA visualization of benchmark-level gate patterns.

## T  DISCUSSION ON MODALITY-SPECIFIC SPARSITY

To further contextualize the modality-specific sparsity patterns exhibited by VideoNSA, we compare its behavior with the text-only NSA (Pai et al., 2025a) used in language models. As shown in Figure 28, the text-only NSA (Pai et al., 2025a) displays a distinct gating dynamic. The sliding-window branch gradually becomes dominant in deeper layers, while the compression and selection branches diminish rapidly and remain almost inactive throughout most of the network. This pattern reflects the one-dimensional and relatively uniform nature of textual sequences, where long-range interactions are sparse and stable, and the model tends to converge toward a single prevailing routing path. The text-only NSA (Pai et al., 2025a) also presents a noticeable anomaly in the final layer, where all three branches suddenly become active again despite having remained largely inactive in

Table 34: Ablation on static gates averaged over a 1K training subset.

| Model | Long Video Understanding | | | | Temporal Reasoning | Spatial Understanding |
|---|---|---|---|---|---|---|
| | LongVideoBench | MLVU$_{Test}$ | TimeScope | LongTimeScope | Tomato | VSIBench |
| Qwen2.5-VL-7B | 58.7 | 51.2 | 81.0 | 40.7 | 22.6 | 29.7 |
| VideoNSA | 59.4 (+1.1%) | 51.8 (+1.2%) | 82.7 (+2.1%) | 44.4 (+9.1%) | 26.2 (+15.9%) | 36.1 (+20.3%) |
| VideoNSA+ Static Gate | 58.4 (-0.5%) | 51.2 (0.0%) | 81.2 (+0.2%) | 41.5 (+2.0%) | 23.7 (+4.8%) | 31.8 (+7.1%) |

Table 35: Theoretical FLOPs comparison among different attention mechanisms. "VideoNSA (ideal)" denotes the theoretical FLOPs of NSA without query-head padding.

| Method | FLOPs | Relative |
|---|---|---|
| Flash Attention | 8.40 PF | 1.00× |
| Tri-shape | 7.07 PF | 0.84× |
| MInference | 4.13 PF | 0.49× |
| Flexprefill | 7.75 PF | 0.92× |
| XAttention | 1.94 PF | 0.23× |
| VideoNSA (ideal) | 2.05 PF | 0.24× |
| VideoNSA | 4.68 PF | 0.56× |

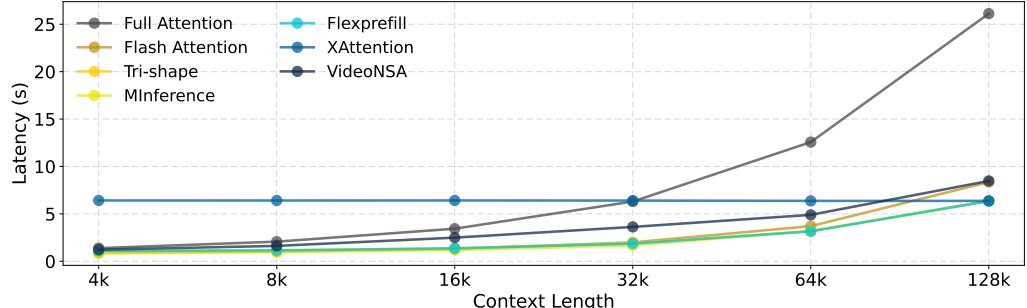

(a) Absolute prefill latency across attention mechanisms.

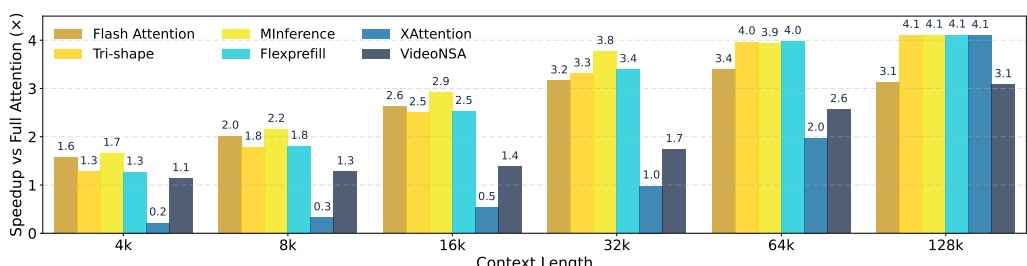

(b) Prefill-time speedup over full attention across context lengths.

Figure 26: Inference efficiency comparison across attention mechanisms.

previous layers. This behavior suggests a late-stage shift in inductive patterns that is characteristic of language modeling. In contrast, VideoNSA, as shown in Figure 17, maintains active and balanced usage of all three branches across nearly the entire depth of the network, with the compression branch playing a consistently prominent role.

The inter-head similarities further highlight the divergence between the two modalities. The text-only NSA (Pai et al., 2025a) in Figure 29 exhibits strong correlations across heads in the early layers, which indicates a set of conserved induction-like operations. Later in the model, selection and sliding window gate values become decorrelated across heads. VideoNSA, as shown in Section O, however, displays substantially weaker cross-head correlations overall, and only a few mid-layer clusters emerge in the selection and sliding-window branches. These findings imply that VideoNSA adjusts its sparse routing behavior to accommodate the rich spatiotemporal redundancy and multi-scale

Table 36: CMP latency under different block sizes and context lengths.

| Block Size | 4k | 8k | 16k | 32k | 64k | 128k |
|---|---|---|---|---|---|---|
| 32 | 0.868 | 1.022 | 1.311 | 1.982 | 3.698 | 8.343 |
| 64 | 0.882 | 1.036 | 1.322 | 1.997 | 3.687 | 8.323 |
| 128 | 0.880 | 1.027 | 1.308 | 1.993 | 3.705 | 8.353 |

Table 37: Latency comparison between different NSA implementations at 8k context length.

| Implementation | forward | backward |
|---|---|---|
| nsa-impl (Pai et al., 2025a) | 5.402 | 32.826 |
| flash-nsa (mdy666, 2025) | 2.429 | 5.537 |

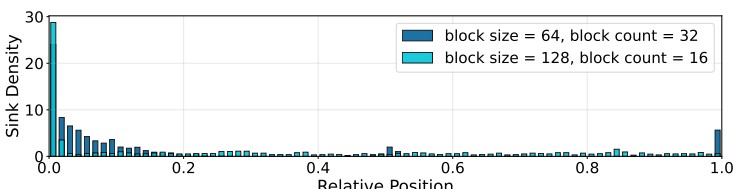

(a) Compression sinks across block size and counts.

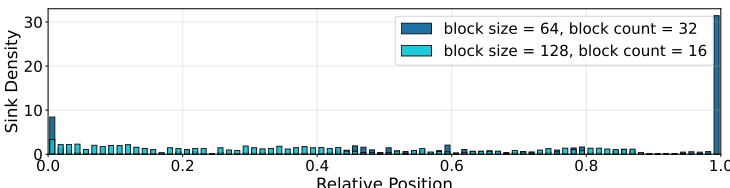

(b) Selection sinks across block size and counts.

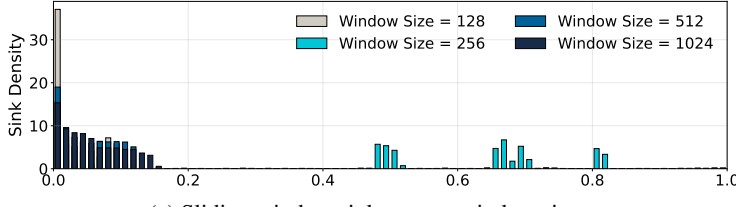

(c) Sliding window sinks across window sizes.

Figure 27: Attention sink distributions across the three branches under different sparse settings.

structure of video inputs, rather than collapsing into a single dominant pathway as observed in the text-only model.

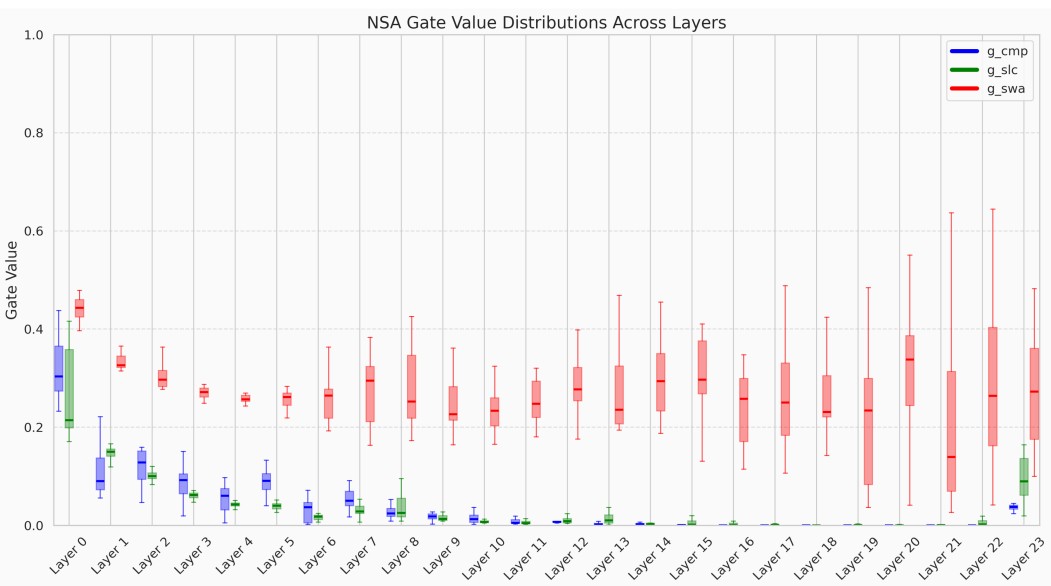

Figure 28: Layer-wise gate distributions of text-only NSA (Pai et al., 2025a).

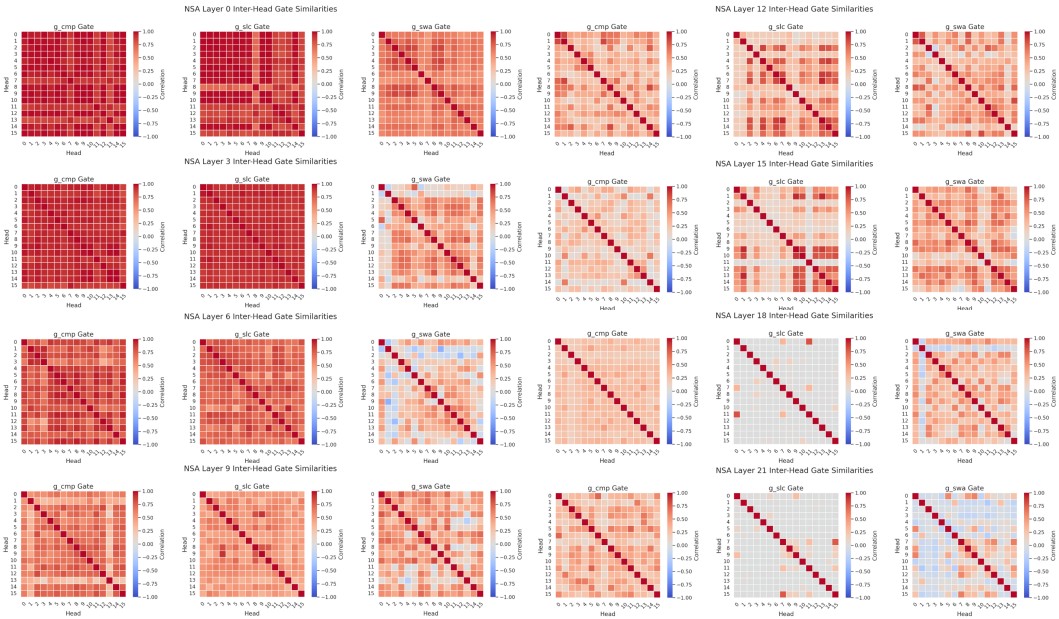

Figure 29: Inter-head gate similarities of text-only NSA (Pai et al., 2025a).

# U DENSE ATTENTION SINK VISUALIZATION

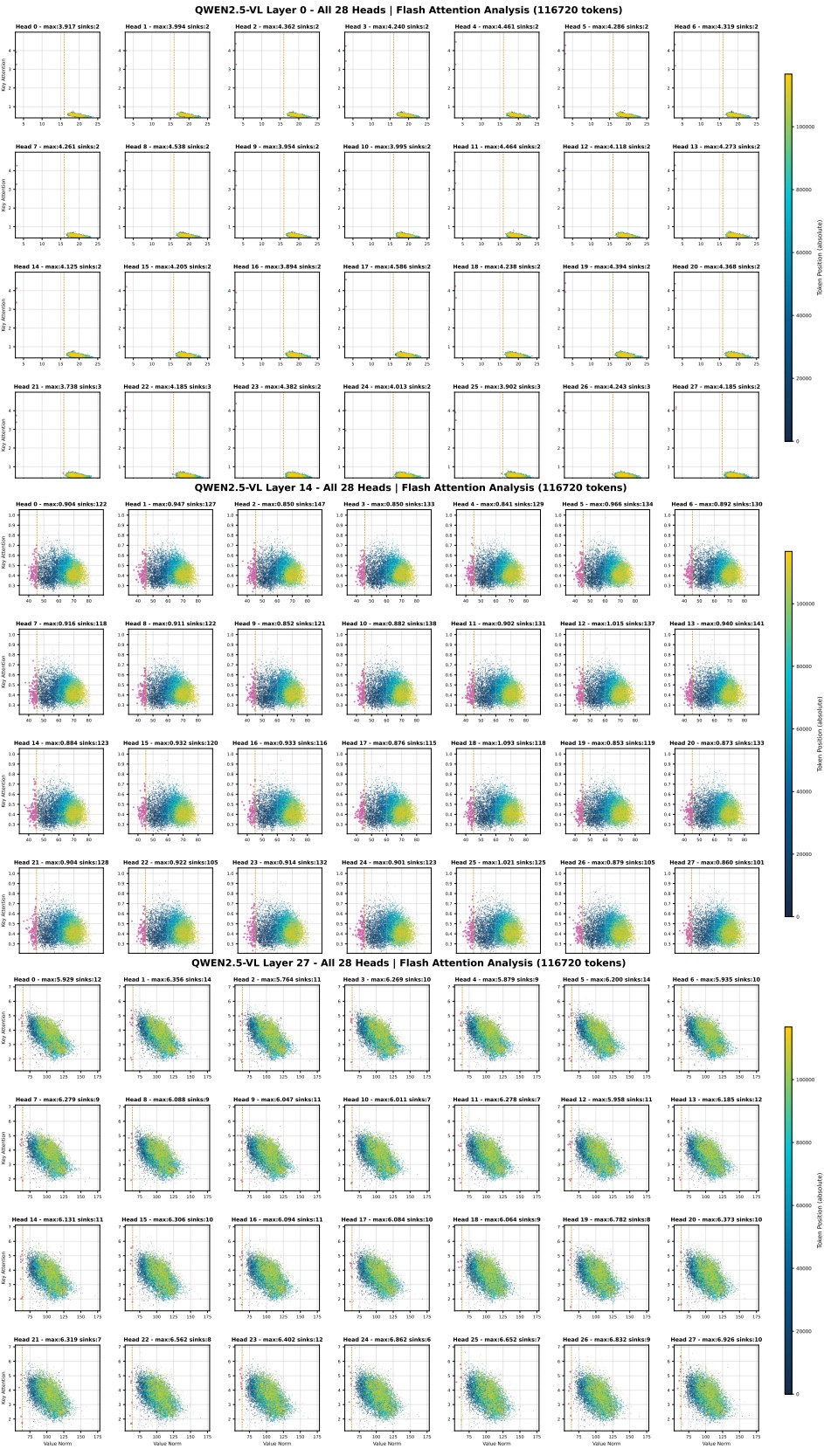

