# OpenReview forum: "VideoNSA: Native Sparse Attention Scales Video Understanding"
_ICLR.cc/2026/Conference — ICLR 2026 Poster_

### Official Review · Reviewer_awwy · 2025-10-16

**Soundness:** 3
**Presentation:** 3
**Contribution:** 3
**Rating:** 6
**Confidence:** 4

**Summary:**

This paper adapts Native Sparse Attention (NSA) to video-language models, demonstrates its potential in tasks including long video understanding, temporal reasoning, and spatial intelligence, and conducts extensive and thorough ablation and analytical experiments. Additionally, to mitigate attention sinks in long-term visual contexts, this paper further proposes to dynamically integrate global and local attention via three complementary branches, effectively addressing this issue.

**Strengths:**

1. This paper attempts to apply Native Sparse Attention (NSA) to VideoMLLMs for the first time, and effectively demonstrates its potential in tasks such as long video understanding.

2. The ablation experiments in this paper are quite comprehensive and rigorous, which analyze the performance of NSA in video understanding tasks from multiple perspectives.

**Weaknesses:**

1. This paper appears to lack an analysis of the training and inference efficiency of VideoNSA. For instance, regarding Table 1, it would be desirable to know the comparisons between various methods and VideoNSA in terms of inference efficiency, latency, and FLOPs. For training efficiency, a comparison between VideoNSA and full attention under the same context length is also expected.

2. Regarding the application of NSA in video tasks, the primary video-related modification in this paper seems to be employing standard GQA for text while adopting NSA for video attention. It is of interest to understand the impact of this operation on the final performance. Does "Dense-NSA" in Table 3 refer to the use of NSA for all modalities? It is suggested that the authors further elaborate on the performance differences of NSA across different modalities (i.e., video and text).

**Questions:**

1. Does "Dense-NSA" in Table 3 refer to the use of NSA for all modalities?
2. Regarding Figure 2, does the performance under the 64k context length appear to be better than that under 128k? Does this indicate that training under a short context length (i.e., 36k) is insufficient to unlock the full 128k performance of VideoNSA?

---

> ### Author Response · Authors · 2025-11-19
>
> **W1: This paper appears to lack an analysis of the training and inference efficiency of VideoNSA. For instance, regarding Table 1, it would be desirable to know the comparisons between various methods and VideoNSA in terms of inference efficiency, latency, and FLOPs. For training efficiency, a comparison between VideoNSA and full attention under the same context length is also expected.**
>
> Thank you for the question regarding training and inference efficiency. In Table34 in Appendix Q, we compare the prefill latency of VideoNSA, several sparse attention methods, and the Qwen2.5-VL dense baseline under the maximum context length of 128K. In addition, Figure 26 in Appendix Q visualizes the wall-clock latency trends across different context lengths. The results show that **VideoNSA maintains a clear inference-efficiency advantage across long context lengths**. Its latency grows much more slowly than dense attention, and compared with other sparse baselines, it delivers competitive inference speed while achieving stronger model performance.
>
> **Table: Theoretical FLOPs comparison among different attention mechanisms.**
> | Method | Flops | Relative |
> | --- | --- | --- |
> | Flash Attention | 8.40 PF | **1.00×** |
> | Tri-shape | 7.07 PF | **0.84×** |
> | MInference | 4.13 PF | **0.49×** |
> | Flexprefill | 7.75 PF | **0.92×** |
> | XAttention | 1.94 PF | **0.23×** |
> | VideoNSA(ideal) | 2.05 PF | **0.24×** |
> | VideoNSA | 4.68PF | **0.56×** |
>
> Under the same optimization settings, VideoNSA requires about 4600 H100 GPU hours to train, which is 0.87× the dense baseline (5280 H100 GPU Hour), indicating **a small improvement in training efficiency**.
>
> In terms of efficiency bottlenecks, we find that the **main overhead comes from the current NSA kernel implementation**, rather than from the sparse pattern itself. Native Sparse Attention [1] points out that NSA provides significant improvements in decode, forward, and backward efficiency. However, due to the unusual head configuration of Qwen2.5-VL 7B (4 KV heads vs. 28 query heads), we must pad the query heads to 64 to satisfy Triton kernel requirements. This padding inevitably introduces additional computation, causing the actual FLOPs and wall-clock latency of VideoNSA to be higher than its theoretical value (the ideal NSA FLOPs is only 2.05 PF, which is 0.24× that of Flash Attention).
>
> Moreover, we currently adopt an open-source NSA package[2], and different NSA implementations [3][4] exhibit substantial variation in runtime efficiency. As shown in Table 37  in Appendix Q, **the latency of VideoNSA is still strongly influenced by low-level kernel implementation details, rather than by the sparse attention design itself**. Based on these observations, we plan to develop a lighter NSA kernel tailored for small QKV-head configurations in future versions. We include these comparison in Appendix Q.
>
> **Table: Latency comparison between different NSA implementations at 8k context length.**
> |  | forward | backward |
> | --- | --- | --- |
> | nsa-impl [2] | 5.402 | 32.826 |
> | flash-nsa [3] | 2.429 | 5.537 |
>
> [1] Yuan, Jingyang, et al. "Native sparse attention: Hardware-aligned and natively trainable sparse attention." *Proceedings of the 63rd Annual Meeting of the Association for Computational Linguistics (Volume 1: Long Papers)*. 2025.
>
> [2] https://github.com/tilde-research/nsa-impl/tree/main?tab=readme-ov-file
>
> [3] https://github.com/mdy666/Scalable-Flash-Native-Sparse-Attention
>
> [4] https://henryhmko.github.io/posts/nsa_tpu/nsa_tpu.html

---

> > ### Author Response · Authors · 2025-11-19
> >
> > **W2: Regarding the application of NSA in video tasks, the primary video-related modification in this paper seems to be employing standard GQA for text while adopting NSA for video attention. It is of interest to understand the impact of this operation on the final performance. Does "Dense-NSA" in Table 3 refer to the use of NSA for all modalities? It is suggested that the authors further elaborate on the performance differences of NSA across different modalities (i.e., video and text).**
> >
> > Thank you for raising this important question. In “Dense-NSA”, **we directly reuse the QKV weights trained by VideoNSA and run dense-attention inference on all tokens**, which isolates the effect of sparsity from differences in training distributions.
> >
> > We apply NSA only to the vision modality because video inputs contain substantial spatiotemporal redundancy, whereas text sequences are relatively shorter and semantically dense. Keeping full attention for text tokens preserves stable instruction-following ability, while applying learnable sparsity to vision tokens significantly improves efficiency and focuses computation on the most informative visual regions. In early small-scale experiments using 40K samples, **we observe that applying NSA to both vision and text tokens led to noticeable performance degradation**. Therefore, the training stage of VideoNSA enables NSA only for vision tokens, while text tokens remain under dense attention.
> >
> > We then evaluated whether a model trained with vision-only NSA could generalize to applying sparse attention to both vision and text tokens at inference time. The results show that such “vision+text sparse inference” introduces significant performance drops across all benchmarks, with particularly severe degradation on Tomato and VSIBench. In contrast, VideoNSA maintains stable and substantial improvements when sparse attention is used only on vision tokens. This indicates that sparsity patterns learned in the vision modality cannot be directly transferred to multimodal inputs containing text, and that these patterns are highly modality-specific, requiring separate designs for vision and language structures. Reviewer mLuS also commented that our “proposed hybrid attention mechanism is a sensible architectural choice.”
> >
> > **Table: Effect of applying NSA to vision vs. vision+text tokens.**
> > |  | LVB | MLVU | TimeScope | LTS | Tomato | VSIBench |
> > | --- | --- | --- | --- | --- | --- | --- |
> > | Qwen2.5-VL 7B | 58.7 | 51.2 | 81.0 | 40.7  | 22.6 | 29.7 |
> > | VideoNSA (Vision Only) | 60.0 | 51.8 | 83.7 | 44.4 | 26.5 | 36.1 |
> > | VideoNSA (Vision + Text) | 21.0 | 16.4 | 3.1 | 0 | 17.51 | 0.1 |
> >
> >
> > **Q1: Does "Dense-NSA" in Table 3 refer to the use of NSA for all modalities?**
> >
> > Thank you for the question. No, “Dense-NSA” does not apply NSA to all modalities. It reuses the QKV weights trained by VideoNSA but performs fully dense attention on both vision and text tokens during inference.

---

> > > ### Author Response · Authors · 2025-11-19
> > >
> > > **Q2: Regarding Figure 2, does the performance under the 64k context length appear to be better than that under 128k? Does this indicate that training under a short context length (i.e., 36k) is insufficient to unlock the full 128k performance of VideoNSA?**
> > >
> > > Thank you for raising this question regarding the performance differences across context lengths. In Figure 2, it is true that several benchmarks show slightly better results at 64K compared with 128K. However, **128K still outperforms 32K on the majority of tasks**, indicating that **VideoNSA maintains strong extrapolation ability in the long-context regime rather than failing at 128K**.
> > >
> > > The base model, Qwen2.5-VL 7B, has a maximum context window of 128K. As shown in Figure 15 in Appendix K, **when approaching 128K, the base model exhibits noticeable performance degradation relative to 64K** on multiple benchmarks. In contrast, **VideoNSA maintains stable or improved performance at 128K**, demonstrating that the proposed sparse architecture helps stabilize long-context behavior even when the underlying backbone begins to weaken.
> > >
> > > In addition, due to single-GPU memory limitations, **the maximum training context we can use is 36K**, which restricts the model’s exposure to ultra-long sequences during training. The current lack of sufficiently large long-video datasets further limits the model’s robustness at extreme context lengths.
> > >
> > > Prior studies like LM-Infinite [1] reports that **model performance does not increase monotonically with context length**. When the inference length approaches the limits of positional encoding or architectural design, models often experience attention instability, weakened deep-layer signals and a decline in long-range dependency modeling. LongVideoBench [2] similarly finds that **simply feeding more video frames does not guarantee better understanding**. These results suggest that **extremely long contexts inherently introduce optimization difficulty and representation dilution**. Even if an architecture supports up to 128K, performance may saturate or slightly drop near the upper range.
> > >
> > > We agree that with a training window limited to 36K, our model cannot fully realize its potential at 128K. When computational resources permit, we plan to extend training to longer contexts (such as 64K or even 128K) to further improve the stability and overall performance of VideoNSA under ultra-long contexts.
> > >
> > > [1] Han, Chi, et al. "Lm-infinite: Zero-shot extreme length generalization for large language models." *Proceedings of the 2024 Conference of the North American Chapter of the Association for Computational Linguistics: Human Language Technologies (Volume 1: Long Papers)*. 2024.
> > >
> > > [2] Wu, Haoning, et al. "Longvideobench: A benchmark for long-context interleaved video-language understanding." *Advances in Neural Information Processing Systems* 37 (2024): 28828-28857.

---

### Official Review · Reviewer_mLuS · 2025-10-20

**Soundness:** 3
**Presentation:** 3
**Contribution:** 3
**Rating:** 6
**Confidence:** 3

**Summary:**

This paper targets the challenge of long-video understanding in MLLMs, which is constrained by the quadratic complexity of standard attention. The authors propose VideoNSA, a hybrid attention mechanism that adapts Native Sparse Attention for video-language models. The core of the method is to apply standard Grouped-Query Attention to text tokens while using a learnable, three-branch sparse attention mechanism for the video tokens. This video-specific NSA dynamically combines a Token Compression branch, a Token Selection branch, and a Sliding Window branch using learnable gates. The model, which is an adaptation of Qwen2.5-VL , is trained end-to-end on a 216K video instruction dataset. The authors present experiments showing that VideoNSA achieves competitive performance on long-video understanding, temporal reasoning, and spatial benchmarks. The paper also provides a very detailed analysis of the model's scaling properties, attention budget allocation, internal branch usage, and its effect on mitigating attention sinks.

**Strengths:**

1. The paper addresses a highly significant and timely problem: scaling video-language models to handle long contexts (e.g., thousands of frames or 128K tokens) efficiently.

2. The proposed hybrid attention mechanism is well-motivated. Preserving dense attention for text tokens while applying aggressive, learnable sparsity to the highly redundant video tokens  is a sensible architectural choice.

3. The paper's main strength is its extensive analysis section. The authors provide commendable, deep insights into the model's behavior, including:
- A study of information scaling (spatial vs. temporal trade-offs) .
- A detailed breakdown of attention budget allocation (global vs. local).
- An analysis of the dynamic gate usage across layers.
- A novel and valuable investigation into how different sparse branches uniquely contribute to or mitigate attention sinks.

**Weaknesses:**

1. My most significant concern is the flawed "Dense-SFT" baseline. This baseline, which should serve as the primary control, was fine-tuned on the same 216K dataset as VideoNSA. However, this Dense-SFT model performed worse than the original, pre-trained Qwen2.5-VL on most benchmarks (e.g., LVB, TimeScope, Tomato). The authors attribute this to the "limited quality of the training data". This admission severely confounds the paper's central claim. We cannot know if VideoNSA's architectural improvements are genuine or if the VideoNSA architecture is simply more robust to this specific, low-quality training data than a dense model. The experiment fails to demonstrate that VideoNSA is better than a properly trained dense model.

2. While the paper claims "improved performance," the results in Table 1 are more accurately described as "competitive" or "on par" rather than a significant step forward.
- On Long VideoBench (LVB), VideoNSA (60.0) is outperformed by Video-XL-2 (61.0).
- On $MLVU_{test}$, VideoNSA (51.8) is outperformed by Video-XL-2 (52.2) and InternVL2.5-8B (55.8).
- On Long TimeScope (LTS) and TimeScope, its scores (44.4 and 83.7) are effectively tied with other sparse attention methods like MInference and XAttention.
- Given the confounding baseline (Weakness 1), these marginal gains are not sufficient to robustly claim superiority.

3. The paper is motivated by efficiency, but its own analysis (Finding 5) identifies the compression (CMP) branch as the dominant latency bottleneck as context length grows. The paper concludes that "the prefill stage remains the primary bottleneck". While the analysis is transparent and appreciated, the paper identifies a critical practical limitation of its own method without offering a solution. This undermines the practical efficiency claims of the work.

**Questions:**

1. Could you elaborate on the "Dense-SFT" baseline's performance drop? If the training data is of limited quality, how can you be sure it isn't also limiting VideoNSA's potential? Conversely, do you hypothesize that VideoNSA's gains over the dense baseline would be larger or smaller if trained on a much larger, higher-quality video instruction dataset?

2. Given the CMP branch is the bottleneck, do you have concrete suggestions for optimizing it? The paper states the block-level representation is obtained by "averaging all tokens" 29, but the preliminary definition (Eq. 2) mentions a "learnable MLP" ($\varphi$)30.
- First, please clarify this: is the learnable MLP simply performing a weighted average, or is it a more complex, non-linear projection?
- Second, if it is a simple average, the latency should be low. If it's a learnable MLP, did you experiment with replacing it with a fixed (non-learnable) pooling operation to see if the performance/latency trade-off improves?

3. The authors note the "strange behavior" in the last layer (L27) where all three branch gates become fully active. Do you have any hypothesis for this? Is it a learned behavior for final-layer aggregation, or could it be an artifact of training (e.g., the gates for that layer not receiving a strong gradient)?

---

> ### Author Response · Authors · 2025-11-19
>
> **W1: My most significant concern is the flawed "Dense-SFT" baseline. This baseline, which should serve as the primary control, was fine-tuned on the same 216K dataset as VideoNSA. However, this Dense-SFT model performed worse than the original, pre-trained Qwen2.5-VL on most benchmarks (e.g., LVB, TimeScope, Tomato). The authors attribute this to the "limited quality of the training data". This admission severely confounds the paper's central claim. We cannot know if VideoNSA's architectural improvements are genuine or if the VideoNSA architecture is simply more robust to this specific, low-quality training data than a dense model. The experiment fails to demonstrate that VideoNSA is better than a properly trained dense model.**
>
> Thank you for the comments on the Dense-SFT baseline. We agree that a well-designed baseline is essential for verifying the architectural contribution of VideoNSA, and therefore we ensured that Dense-SFT and VideoNSA are compared under exactly the same training data, training pipeline, and optimization settings.
>
> First, we would like to clarify that **the slightly lower performance of Dense-SFT compared with the original Qwen2.5-VL-7B is caused by the quality and distributional limitations of the current large-scale video instruction tuning data**. In particular, although LLaVA-Video-178K is currently the best open-source video instruction dataset, it inevitably contains noise, has limited domain coverage, and remains far smaller and less diverse than the high-quality corpus used for Qwen2.5-VL pretraining. As a result, a mild degradation in Dense-SFT is expected and should not be interpreted as training failure or improper data usage.
>
> Dense-SFT and VideoNSA are trained under identical data and identical optimization settings, so the performance differences between the two models cannot be attributed to data distribution shifts or differences in training workflows. Instead, they directly reflect differences in architecture. **In Appendix G, we further report results of the original Qwen2.5-VL, Dense-SFT, and VideoNSA across LSDBench[1] and VideoEvalPro[2].** While Dense-SFT exhibits similar performance as the pretrained model, VideoNSA consistently shows clear gains across multiple datasets and tasks under the exact same training setup. This demonstrates that the observed improvements are not accidental or due to “better tolerance to low-quality data,” but rather stem from VideoNSA’s ability to better exploit redundancy structures in long videos and to model long-range dependencies more effectively.
>
> **Table: Results on LSDBench**
> | **Model** | **Accuracy** |
> | --- | --- |
> | LongVA  | 32.5 |
> | LongVila  | 49.8 |
> | InternVL2.5  | 50.1 |
> | **Qwen2.5-VL-7B**  | **52.2** |
> | Qwen2.5-VL-7B-SFT | 52.5 |
> | **Sparse Attention Methods** |  |
> | + Tri-Shape  | 49.5 |
> | + MInference  | 49.5 |
> | + FlexPrefill  | 52.3 |
> | + XAttention  | 51.3 |
> | **VideoNSA** | **55.2** |
>
> **Table: Results on VideoEvalPro. HP stands for Holistic Perception, HR stands for Holistic Reasoning, LR stands for Local Reasoning, LP stands for Local Perception.**
> | **Model** | **HP** | **HR** | **LR** | **LP** | **Overall** |
> | --- | --- | --- | --- | --- | --- |
> | LongVA  | 20.5 | 6.8 | 19.0 | 9.5 | 16.5 |
> | Video-XL  | 22.3 | 15.0 | 18.2 | 10.2 | 18.6 |
> | InternVL2.5  | 28.8 | 19.7 | 21.5 | 16.7 | 24.6 |
> | **Qwen2.5-VL-7B**  | **33.9** | **15.6** | **24.8** | **17.8** | **27.7** |
> | Qwen2.5-VL-7B-SFT | 34.5 | 15.8 | 25.3 | 18.2 | 28.3 |
> | **Sparse Attention Methods** |  |  |  |  |  |
> | + Tri-Shape  | 34.1 | 16.3 | 25.1 | 20.0 | 28.4 |
> | + MInference  | 32.3 | 17.1 | 27.7 | 26.7 | 26.0 |
> | + FlexPrefill  | 33.0 | 15.9 | 26.3 | 19.8 | 28.3 |
> | + XAttention  | 34.5 | 16.6 | 25.6 | 20.5 | 28.9 |
> | **VideoNSA** | **35.4** | **16.9** | **26.3** | **19.1** | **29.4** |
>
> We agree that if one could train a sparse attention model on a large-scale, high-quality video instruction dataset comparable in scale and quality to the pretraining corpus of Qwen2.5-VL, then the upper bound of VideoNSA’s performance might increase further. At the same time, as the data scale and quality improve, the spatiotemporal redundancy and long-range patterns in videos become even more pronounced. We therefore expect that the advantages of VideoNSA over dense attention would further widen in such settings. In future work, we plan to extend VideoNSA to additional backbones and validate it on larger and higher-quality video datasets.
>
> [1] Qu, Tianyuan, et al. "Does Your Vision-Language Model Get Lost in the Long Video Sampling Dilemma?." *arXiv preprint arXiv:2503.12496* (2025).
>
> [2] Ma, Wentao, et al. "Videoeval-pro: Robust and realistic long video understanding evaluation." *arXiv preprint arXiv:2505.14640* (2025).

---

> > ### Author Response · Authors · 2025-11-19
> >
> > **W2: While the paper claims "improved performance," the results in Table 1 are more accurately described as "competitive" or "on par" rather than a significant step forward.**
> >
> > We thank the reviewer for the helpful clarification and revise the abstract to describe our results as "competitive" rather than “improved performance.” Our goal in this paper is not to claim state-of-the-art performance, but to validate the effectiveness of VideoNSA under a fair and transparent setting. We fine-tune VideoNSA on a subset of the LLaVA-Video-178K[1], which is one of the largest publicly available open-source video instruction corpora. In contrast, models such as Video-XL-2[2] and InternVL2.5-8B[3] are trained on undisclosed or proprietary multimodal data, which makes a strictly data-controlled comparison difficult and shifts the focus to architectural differences rather than training data.
> >
> > We acknowledge that the performance improvements of VideoNSA over existing methods are modest rather than substantial.  However, across all eight benchmarks, VideoNSA achieves stable and competitive performance on diverse video understanding tasks in Table1, Table 12, and Table 13. In addition, we include an SFT baseline where we fine-tune Qwen2.5-VL on the same data used for VideoNSA. The improvement of VideoNSA over this dense baseline further supports the effectiveness of the proposed sparse architecture.
> >
> > **Table: VideoNSA demonstrates competitive performance across several video understanding benchmarks**
> > |  | LVB | MLVU | TimeScope | LTS | Tomato | VSIBench | LSDBench | VideoEvalPro |
> > | --- | --- | --- | --- | --- | --- | --- | --- | --- |
> > | Qwen2.5-VL 7B | 58.7 | 51.2 | 81.0 | 40.7  | 22.6 | 29.7 | 52.2 | 27.7 |
> > | Qwen2.5-VL 7B (SFT) |  57.8 | 51.2 | 76.8  | 40.2 | 21.7 | 30.5 | 52.5 | 28.3 |
> > | VScan (Token Compression) | 58.7 | 48.1 | 80.3  | 31.1 | 19.1 | 34.4 | - | - |
> > | XAttention (Training-free Sparse Attention) | 59.1 |  50.2  | 83.1 | 41.1 | 21.4  | 36.6 | 52.3 | 28.9 |
> > | VideoNSA | 60.0 | 51.8 | 83.7 | 44.4 | 26.5 | 36.1 | 55.2 | 29.4 |
> >
> > [1] Zhang, Yuanhan, et al. "Video instruction tuning with synthetic data." *arXiv preprint arXiv:2410.02713* (2024).
> >
> > [2] Qin, Minghao, et al. "Video-XL-2: Towards Very Long-Video Understanding Through Task-Aware KV Sparsification." *arXiv preprint arXiv:2506.19225* (2025).
> >
> > [3]Chen, Zhe, et al. "Expanding performance boundaries of open-source multimodal models with model, data, and test-time scaling." *arXiv preprint arXiv:2412.05271* (2024).

---

> ### Author Response · Authors · 2025-11-19
>
> **W3: The paper is motivated by efficiency, but its own analysis (Finding 5) identifies the compression (CMP) branch as the dominant latency bottleneck as context length grows. The paper concludes that "the prefill stage remains the primary bottleneck". While the analysis is transparent and appreciated, the paper identifies a critical practical limitation of its own method without offering a solution. This undermines the practical efficiency claims of the work.**
>
> Thank you for the comments on the efficiency analysis. Under a unified and comparable training and inference setup, **VideoNSA achieves consistent performance improvements across multiple video benchmarks, while maintaining efficiency that is competitive with representative sparse attention methods and clearly superior to dense attention**. The core contribution of VideoNSA is **not proposing “the fastest” attention mechanism, but conducting a systematic analysis of sparse patterns across different video scenarios based on Native Sparse Attention[1]**.
>
> Regarding Finding 5, which identifies the CMP branch as the dominant source of latency, we first investigate this phenomenon from the perspective of architectural hyperparameters. Since the block size directly determines how many CMP operations need to be executed, we measured the latency from 4K to 128K context lengths under different block sizes. **The resulting curves are nearly identical, indicating that simply tuning block size or similar structural hyperparameters cannot meaningfully mitigate the CMP bottleneck and is more likely to degrade representation quality, as also reflected in Finding 3**.
>
> **Table: Theoretical FLOPs comparison among different attention mechanisms.**
> | Method | Flops | Relative |
> | --- | --- | --- |
> | Flash Attention | 8.40 PF | **1.00×** |
> | Tri-shape | 7.07 PF | **0.84×** |
> | MInference | 4.13 PF | **0.49×** |
> | Flexprefill | 7.75 PF | **0.92×** |
> | XAttention | 1.94 PF | **0.23×** |
> | VideoNSA(ideal) | 2.05 PF | **0.24×** |
> | VideoNSA | 4.68PF | **0.56×** |
>
> Native Sparse Attention[1] indicates that NSA can theoretically reduce the computation cost of forward, backward, and decode operations. In Table34 in Appendix Q, we include the FLOPs comparison under 128K context length. **Ideally, VideoNSA requires only 2.05 PF, which is 0.24× that of Flash Attention and ranks among the more efficient methods**. However, Qwen2.5-VL 7B has 4 KV heads vs. 28 query heads, and **we need to pad the 28 query heads to 64 to satisfy Triton kernels requirements, inevitably introducing additional computation and memory access overhead. This causes the actual FLOPs of VideoNSA (4.68 PF, about 0.56× Flash Attention) and its wall-clock latency to exceed the ideal theoretical values. The wall-clock prefilling time from 4K to 128K in Table34 in Appendix Q further confirms that VideoNSA achieves clear acceleration compared with dense attention, maintains competitive latency relative to other sparse baselines**.
>
> **Table: Latency comparison between different NSA implementations at 8k context length.**
> |  | forward | backward |
> | --- | --- | --- |
> | nsa-impl [3] | 5.402 | 32.826 |
> | flash-nsa [2] | 2.429 | 5.537 |
>
> At the same time, **different NSA implementations exhibit substantial differences in kernel efficiency**. For example, flash-nsa[2] achieves roughly 2× speedup in the forward pass and up to 6× in the backward pass compared with nsa-impl[3] under the same context length. This further supports that the primary bottleneck lies in kernel-level implementation and memory scheduling, rather than any inherent limitation of the VideoNSA sparse architecture.
>
> In future work, we plan to design more efficient NSA kernels and memory-access strategies tailored for small QKV-head configurations and multimodal VLMs, with the goal of further narrowing the gap between theoretical FLOPs and practical latency.
>
> [1] Yuan, Jingyang, et al. "Native sparse attention: Hardware-aligned and natively trainable sparse attention." *Proceedings of the 63rd Annual Meeting of the Association for Computational Linguistics (Volume 1: Long Papers)*. 2025.
>
> [2] https://github.com/mdy666/Scalable-Flash-Native-Sparse-Attention
>
> [3] https://github.com/tilde-research/nsa-impl/tree/main?tab=readme-ov-file

---

> ### Author Response · Authors · 2025-11-19
>
> **Q1: Could you elaborate on the "Dense-SFT" baseline's performance drop? If the training data is of limited quality, how can you be sure it isn't also limiting VideoNSA's potential? Conversely, do you hypothesize that VideoNSA's gains over the dense baseline would be larger or smaller if trained on a much larger, higher-quality video instruction dataset?**
>
> Thank you for the reviewer’s thoughtful question.
> The performance drop of the Dense-SFT baseline primarily arises from the fact that both Dense-SFT and VideoNSA are fine-tuned on LLaVA-Video-178K, a dataset that is widely used but still substantially noisier, and less diverse than the pre-training corpus of Qwen2.5-VL. Throughout our experiments, we ensure that **Dense-SFT and VideoNSA are trained with exactly the same data, optimization settings, and fine-tuning pipeline. This design guarantees that any performance difference reflects architectural effects rather than differences in training conditions or data distribution.**
>
> **When a large pre-trained dense model is fine-tuned on a small and relatively narrow video instruction dataset, dense attention often adapts too strongly to the new distribution**. This over-adaptation leads to overfitting and partial forgetting of pre-trained capabilities, as reported in [1]. [2] also shows that multimodal models frequently exhibit overfitting and performance degradation when fine-tuned on instruction data with limited diversity. This pattern is consistent with what we observe in the Dense-SFT baseline. In contrast, **NSA introduces a degree of implicit regularization during fine-tuning**. Gradients propagate only through the three NSA branches, namely compression, selection, and sliding-window, and update only token interactions that are selected by the routing mechanism. As a result, **most of the pre-trained global attention structure remains intact. This design reduces overfitting on narrow instruction distributions, enables better modeling of temporal redundancy and long-range dependencies in video, and improves robustness under noisy video data.**
>
> In Table 12 and Table 13 in Appendix G, we provide additional comparisons among Qwen2.5-VL, Dense-SFT, and VideoNSA across multiple benchmarks[3][4]. **The improvements achieved by VideoNSA remain stable across different tasks**, which indicates that limitations in the current video instruction data do not weaken the structural contribution of VideoNSA. Instead, the learnable sparse routing mechanism provides stronger robustness and generalization in long-video scenarios.
>
> **Table: VideoNSA demonstrates competitive performance across several video understanding benchmarks**
>
> |  | LVB | MLVU | TimeScope | LTS | Tomato | VSIBench | LSDBench | VideoEvalPro |
> | --- | --- | --- | --- | --- | --- | --- | --- | --- |
> | Qwen2.5-VL 7B | 58.7 | 51.2 | 81.0 | 40.7  | 22.6 | 29.7 | 52.2 | 27.7 |
> | Qwen2.5-VL 7B (SFT) |  57.8 | 51.2 | 76.8  | 40.2 | 21.7 | 30.5 | 52.5 | 28.3 |
> | VScan (Token Compression) | 58.7 | 48.1 | 80.3  | 31.1 | 19.1 | 34.4 | - | - |
> | XAttention (Training-free Sparse Attention) | 59.1 |  50.2  | 83.1 | 41.1 | 21.4  | 36.6 | 52.3 | 28.9 |
> | VideoNSA | 60.0 | 51.8 | 83.7 | 44.4 | 26.5 | 36.1 | 55.2 | 29.4 |
>
> We agree that if a sparse model were trained on a video instruction dataset comparable in scale and quality to the pre-training corpus of Qwen2.5-VL, the performance ceiling of VideoNSA could be further improved. **As the scale and quality of video data increase, temporal redundancy and long-range dependencies become more prominent. Therefore, we expect the advantages of NSA’s learnable sparse routing to grow rather than diminish.** In future work, we plan to extend VideoNSA to additional backbones and evaluate it on larger and higher-quality long-video instruction datasets.
>
> [1] Kotha, Suhas, Jacob Mitchell Springer, and Aditi Raghunathan. "Understanding catastrophic forgetting in language models via implicit inference." *arXiv preprint arXiv:2309.10105* (2023).
>
> [2] He, Jinghan, et al. "Continual instruction tuning for large multimodal models." *arXiv preprint arXiv:2311.16206* (2023).
>
> [3] Ma, Wentao, et al. "Videoeval-pro: Robust and realistic long video understanding evaluation." *arXiv preprint arXiv:2505.14640* (2025).
>
> [4] Qu, Tianyuan, et al. "Does Your Vision-Language Model Get Lost in the Long Video Sampling Dilemma?." *arXiv preprint arXiv:2503.12496* (2025).

---

> ### Author Response · Authors · 2025-11-19
>
> **Q2: Given the CMP branch is the bottleneck, do you have concrete suggestions for optimizing it?**
>
> Thank you for the question regarding the latency of the CMP branch. The “learnable MLP” in Equation (2) follows the defination used in Native Sparse Attention, but in the nsa-impl[1] we use, this module effectively degenerates into a weighted average. In the CMP branch, VideoNSA partitions the long vision-token sequence into blocks according to the block size and performs a weighted average within each block. Therefore, **the primary cost of CMP arises from the large number of repeated aggregation operations, which grow proportionally with the context length.**
>
> Therefore, we vary the block size and measure the resulting latency. **Across all context lengths, the latency curves for different block sizes nearly overlap, indicating that adjusting the block size alone is insufficient to fundamentally alleviate the CMP-branch bottleneck.**
>
> **Table: CMP latency under different block sizes and context lengths.**
> | Block Size | 4k | 8k | 16k | 32k | 64k | 128k |
> | --- | --- | --- | --- | --- | --- | --- |
> | 32 | 0.868 | 1.022 | 1.311 | 1.982 | 3.698 | 8.343 |
> | 64 | 0.882 | 1.036 | 1.322 | 1.997 | 3.687 | 8.323 |
> | 128 | 0.880 | 1.027 | 1.308 | 1.993 | 3.705 | 8.353 |
>
> **The core limitation of CMP comes from the underlying kernel implementation and memory access pattern of the aggregation operation**, especially in long-sequence settings. For example, flash-nsa[2] achieves approximately a 2× speedup in the forward pass and up to a 6× speedup in the backward pass compared with the nsa-impl[1] implementation we currently use. These observations indicate that **the dominant factor governing CMP latency, as well as overall NSA performance, lies in kernel-level optimization, particularly the design of the computation kernel and the associated memory-access strategy.** We plan to explore these directions in future work, since they hold greater potential for substantially improving the efficiency of CMP.
>
> **Table: Latency comparison between different NSA implementations at 8k context length.**
> |  | forward | backward |
> | --- | --- | --- |
> | nsa-impl [1] | 5.402 | 32.826 |
> | flash-nsa [2] | 2.429 | 5.537 |
>
> [1] https://github.com/tilde-research/nsa-impl/tree/main?tab=readme-ov-file
>
> [2] https://github.com/mdy666/Scalable-Flash-Native-Sparse-Attention
>
>
> **Q3: The authors note the "strange behavior" in the last layer (L27) where all three branch gates become fully active. Do you have any hypothesis for this? Is it a learned behavior for final-layer aggregation, or could it be an artifact of training (e.g., the gates for that layer not receiving a strong gradient)?**
>
> Thank you for the attention to the “fully-active” behavior observed in the last layer. **We interpret this pattern as a learned behavior for final-layer aggregation.** To validate this interpretation, we replaced the learned L27 gate with the average gate vector computed from a 1K subset of the training data. **Under this setting, we observe consistent performance drops on several video understanding tasks** as shown in the Table below. This result indicates that the fully-active pattern in L27 is not accidental. Instead, the input-dependent routing at this layer plays a meaningful functional role.
>
> **Table: Effect of replacing the learned L27 routing with a static average gate.**
> |  | LVB | MLVU_test | TimeScope | LTS | Tomato | VSIBench |
> | --- | --- | --- | --- | --- | --- | --- |
> | Qwen2.5-VL 7B | 58.7 | 51.2 | 81.0 | 40.7  | 22.6 | 29.7 |
> | VideoNSA  | 60.0 | 51.8 | 83.7 | 44.4 | 26.5 | 36.1 |
> | VideoNSA (Static L27) | 59.2 | 51.2 | 83.1 | 43.3 | 25.1 | 33.4 |
>
> In addition, we note that the text-only NSA[1] exhibits the same three-branch fully-active pattern in its last layer, while its intermediate layers remain sparse. **This cross-modal consistency suggests that the phenomenon naturally emerges from the NSA routing mechanism as the model converges, rather than being caused by video data, training noise, or insufficient gradients.**
>
> [1]https://www.tilderesearch.com/blog/sparse-attn

---

> ### Comment · Reviewer_mLuS · 2025-11-26
>
> I thank the authors for their rebuttal. It has addressed my concerns. I will maintain my positive rating and will consider raising my score based on the feedback from other reviewers.

---

> > ### Author Response · Authors · 2025-11-28
> >
> > We sincerely thank you for your response and appreciate you mentioning will increase the score.

---

### Official Review · Reviewer_evHH · 2025-11-01

**Soundness:** 3
**Presentation:** 3
**Contribution:** 3
**Rating:** 6
**Confidence:** 4

**Summary:**

This paper introduces VideoNSA, a method for scaling video understanding models to very long contexts by adapting NSA. The core idea is to apply a hybrid attention mechanism to a Qwen2.5-VL-7B. Specifically, text tokens are processed with standard GQA, while video tokens are handled by NSA, which dynamically combines three complementary sparse attention branches: CMP for global aggregation, SLC for salient information, and SWA for local context. The authors fine-tune this model on a 216K video instruction dataset. The resulting model scales effectively to 128K tokens and performs well at a series of challenging long-video benchmarks,

**Strengths:**

- Extensive Evaluation: The paper evaluates VideoNSA across a diverse set of challenging long-video benchmarks, demonstrating competitive or SOTA performance. The inclusion of strong baselines and thorough ablations validates the design choices.
- In-depth Analysis: The analysis in Section 4 is a standout feature. The structured "Findings" provide clear, actionable insights into how sparse attention behaves when scaled. The study of information scaling, budget allocation, and attention sinks goes far beyond a typical model performance paper and offers significant value to the research community.
- Efficiency and Scalability: The paper shows that VideoNSA scales effectively to 128K context lengths, far beyond its training regime. The finding that it achieves top-tier performance with only 3.6% of the dense attention budget is a powerful demonstration of the method's efficiency.

**Weaknesses:**

Novelty: The primary weakness is that the core technical component, NSA, is adapted from a previous work (Yuan et al., 2025b). The novelty is in the application, specific architectural choices for video, and the extensive analysis, rather than a new algorithm.

**Questions:**

In "Finding 5," you identify the token compression (CMP) branch as the main latency bottleneck. Given its importance (as shown in the gate analysis in "Finding 4"), what are your thoughts on potential avenues for optimizing this branch to further improve the model's overall efficiency?

---

> ### Author Response · Authors · 2025-11-19
>
> **W1: Novelty: The primary weakness is that the core technical component, NSA, is adapted from a previous work (Yuan et al., 2025b). The novelty is in the application, specific architectural choices for video, and the extensive analysis, rather than a new algorithm.**
>
> Thank you for the reviewer’s comment. We agree that the core sparse-attention operator in VideoNSA is inherited from NSA rather than a brand-new primitive. Our goal, however, is not to redesign the kernel, but to show that **learnable sparsity behaves in a qualitatively different way in video than in text**, and to turn these observations into concrete design insights for video-LMM architectures.
>
> Unlike one-dimensional language with relatively uniform information density, videos and images consist of high-dimensional, highly redundant visual tokens whose spatiotemporal structure and dynamics fundamentally reshape how sparsity emerges. Through systematic analysis, we characterize the sparse activation patterns over visual tokens, the functional specialization of the three branches under different content, the evolution of sparsity with depth, and the extent to which these branches help alleviate attention-sink issues. None of these phenomena have been documented in prior work, nor can they be inferred from NSA’s behavior in text-only settings.
>
> Thus, although VideoNSA reuses the NSA operator, its contribution lies in **revealing** **previously unexplored properties of learnable sparse attention in visual and video modalities** and in providing empirically grounded guidance for designing future video-LMMs, rather than merely introducing a new operator for its own sake.
>
>
> **Q1: In "Finding 5," you identify the token compression (CMP) branch as the main latency bottleneck. Given its importance (as shown in the gate analysis in "Finding 4"), what are your thoughts on potential avenues for optimizing this branch to further improve the model's overall efficiency?**
>
> Thank you for the question regarding on the CMP branch. The Token Compression (CMP) branch aggregates sequential blocks of keys into more coarse-grained, single block-level representations. As shown in Finding 5, the CMP branch gradually becomes the dominant source of latency as the context length increases.
>
> Since block size affects the number of CMP operations, we vary it and measure the latency. We find that across all context lengths, the latency curves for different block sizes remain highly similar, which indicates that changing the block size alone provides very limited relief for the CMP bottleneck.
>
> **Table: CMP latency under different block sizes and context lengths.**
> | Block Size | 4k | 8k | 16k | 32k | 64k | 128k |
> | --- | --- | --- | --- | --- | --- | --- |
> | 32 | 0.868 | 1.022 | 1.311 | 1.982 | 3.698 | 8.343 |
> | 64 | 0.882 | 1.036 | 1.322 | 1.997 | 3.687 | 8.323 |
> | 128 | 0.880 | 1.027 | 1.308 | 1.993 | 3.705 | 8.353 |
>
> In contrast, we observed that the most substantial acceleration comes from using more efficient NSA implementations. For example, the flash-nsa[1] implementation runs about twice as fast as our current nsa-impl[2] in the forward pass and up to six times faster in the backward pass on 8K context length. Other teams are also developing improved implementations, such as optimizing NSA for TPUs [3]. These findings show that **the main factor affecting CMP and overall NSA performance lies in kernel-level efficiency, particularly kernel design and memory access patterns**.
>
> **Table: Latency comparison between different NSA implementations at 8k context length.**
> |  | forward | backward |
> | --- | --- | --- |
> | nsa-impl [1] | 5.402 | 32.826 |
> | flash-nsa [2] | 2.429 | 5.537 |
>
> Although this work does not specifically focus on designing NSA kernels for small VLMs (for instance, models with fewer than 16 GQA groups), developing lighter and more optimized kernels will be an important direction for our future research. We include these comparison in Appendix R.
>
> [1] https://github.com/mdy666/Scalable-Flash-Native-Sparse-Attention
>
> [2] https://github.com/tilde-research/nsa-impl/tree/main?tab=readme-ov-file
>
> [3] https://henryhmko.github.io/posts/nsa_tpu/nsa_tpu.html

---

### Official Review · Reviewer_8ijG · 2025-11-02

**Soundness:** 2
**Presentation:** 4
**Contribution:** 3
**Rating:** 4
**Confidence:** 4

**Summary:**

This paper extends Native Sparse Attention (NSA), originally designed for long-context models, to the multimodal domain, proposing VideoNSA. This method applies NSA's three-branch hierarchical sparse structure and gating mechanism to video tokens, while employing Grouped Query Attention for text tokens. Furthermore, the paper provides a scalability analysis, offering new insights into the behavior of sparse attention in multimodal models.

**Strengths:**

1. Insightful Analysis: The scalability analysis provides a degree of interpretability for sparse attention, while also clarifying the advantages of the sparse mechanism (e.g., extensibility to contexts longer than those seen during training, control over attention sinks).
2. Comprehensive Experiments: The experimental validation is extensive, covering ablation studies for each branch, visualizations of gating distributions, context extension curves, and performance evaluations on multiple benchmarks.
3. Practical Guidance: The empirical findings offer practical guidance for deploying hardware-aligned sparse attention in long-context multimodal systems.

**Weaknesses:**

1. Limited Algorithmic Novelty: The core framework for the vision component (the three-branch sparse structure + learnable gating) is nearly identical to that of NSA (Yuan et al., 2025), meaning the method lacks fundamental innovation.
2. Lack of Discussion on Modality-Specific Sparsity: The paper does not discuss the differences in sparsity between text and video. In text, sparse attention primarily filters information at the syntactic and semantic levels, where token dependencies are relatively stable and one-dimensional. In video, however, sparsity involves spatiotemporal locality and motion redundancy, implying that the definition of semantic redundancy differs across modalities. This raises questions about whether it is appropriate to directly reuse a text-based sparse attention mechanism for the video modality.
3. Insufficient Theoretical or Analytical Depth: The scalability analysis is predominantly empirical. While it helps users better utilize the model, it lacks theoretical explanations or modeling to elucidate the underlying causes of the observed trends.

**Questions:**

1. The paper lacks a strong motivation for applying NSA, a text-modality method, to the video modality. Many methods exist for long-context modeling; why is NSA a good choice? In other words, do the characteristics of NSA offer unique advantages in the context of video?
2. The authors state that the model utilizes the selection and sliding-window branches in shallow layers to capture fine-grained details and local information, while relying more on the compression branch in deep layers to integrate and refine high-level global semantics. Do the sparse distributions and gating activations differ across various types of videos (e.g., high-speed motion, shot transitions, static scenes), or is the gating behavior solely dependent on layer depth? If the gating behavior is only correlated with layer depth and is independent of input features, does this imply that the model has limited adaptability to different types of video content?

---

> ### Author Response · Authors · 2025-11-19
>
> **W1: Limited Algorithmic Novelty: The core framework for the vision component (the three-branch sparse structure + learnable gating) is nearly identical to that of NSA (Yuan et al., 2025), meaning the method lacks fundamental innovation.**
>
> We thank the reviewer for the valuable feedback. We agree that VideoNSA builds upon the NSA operator and does not introduce a brand-new sparse attention formulation. However, the core contribution of our work lies in **systematically examining how learnable sparse attention behaves when moving beyond language and into the video modality.** Language is a one-dimensional sequence, whereas video is a high-dimensional, highly redundant signal with complex spatiotemporal structure. These intrinsic modality differences imply that the sparsification patterns, branch specialization, and depth-wise evolution of NSA in video cannot be inferred from observations made solely in language settings.
>
> Accordingly, our contribution is **the first systematic empirical analysis of learnable sparse attention in video understanding**. We study the emergent sparse activation patterns over visual tokens, the functional specialization of the three routing branches under different video content, the evolution of sparsity across depth, and the operator’s impact on the attention sink phenomenon. These findings reveal how NSA operates on high-dimensional visual inputs and provide structural insights that may guide future video LLM architecture design. Our focus is on filling this long-standing research gap for video models, rather than pursuing formal novelty in the operator itself.
>
> **W2: Lack of Discussion on Modality-Specific Sparsity: The paper does not discuss the differences in sparsity between text and video. In text, sparse attention primarily filters information at the syntactic and semantic levels, where token dependencies are relatively stable and one-dimensional. In video, however, sparsity involves spatiotemporal locality and motion redundancy, implying that the definition of semantic redundancy differs across modalities. This raises questions about whether it is appropriate to directly reuse a text-based sparse attention mechanism for the video modality.**
>
> We thank the reviewer for the insightful question. We would like to clarify that NSA is a fully learnable operator and VideoNSA is trained end-to-end rather than reusing any gate weights from the text-only NSA. Consequently, the sparsity patterns in VideoNSA are entirely driven by video data. More importantly, we observe that **the sparsification behavior emerging in video differs substantially from that in text**, indicating clear modality-specific characteristics rather than a simple transfer of textual patterns to video.
>
> Concretely, in text-only NSA[1], the deeper layers of the network rely almost exclusively on the sliding-window branch; in the shallow layers, the selection and compression branches exhibit consistently high gate-value correlations across heads, while in deeper layers the correlation gradually shifts toward selection and sliding-window branches. In contrast, in VideoNSA, all three branches remain active across the entire depth, with the compression branch playing an even more prominent role. Moreover, the cross-head correlations are generally much lower in video, with group-wise similarity between the selection and sliding-window branches appearing only in a few middle layers. These differences suggest that the spatiotemporal redundancy structure of video is fundamentally distinct from the one-dimensional structure of text, and that VideoNSA learns a sparsity routing mechanism tailored to video semantics rather than inheriting the behavior observed in text-only settings.
>
> We also appreciate that reviewers evHH and mLuS highlighted the value of our sparsity analysis. In the revised version, we further strengthen the comparison and discussion on Modality-Specific Sparsity to make these observations clearer in Appendix T.
>
> [1] https://www.tilderesearch.com/blog/sparse-attn

---

> ### Author Response · Authors · 2025-11-19
>
> **W3: Insufficient Theoretical or Analytical Depth: The scalability analysis is predominantly empirical. While it helps users better utilize the model, it lacks theoretical explanations or modeling to elucidate the underlying causes of the observed trends.**
>
> We thank the reviewer for the insightful comments on our scalability analysis. Our study performs two scaling experiments along context length and attention budget. We observe that **VideoNSA exhibits strong extrapolation ability on context length: although trained with only 36K tokens, it can generalize to 128K at test time, achieving the best performance at 64K. In contrast, when scaling the attention budget, even a small reduction to ~3.6% of attention computation already delivers outstanding performance, and further increasing the visible-token count does not yield additional gains.** To clarify these phenomena, **we provide theoretical interpretations from routing-path stability and the geometric structure of RoPE.**
>
> **Routing-path Stability**
>
> Recent work [1] indicates that **a model’s ability to maintain performance on long sequences depends critically on the stability of its attention routing structure across positions.** In the standard attention mechanism, the attention weight from the query vector $Q_n$ at position $n$ to the key vector $Z_j$ at position $j$ is defined as
> $\text{Attn}{n\to j}
> =\frac{\exp(Z_j^\top Q_n)}{\sum_k \exp(Z_k^\top Q_n)}.$
> Here, $Q_n$ and $Z_j$ denote the query and key representations at positions $n$ and $j$, respectively. If the model can consistently focus its attention on the task-relevant target token set  $\mathcal{T}$  during inference, then (i) $\sum{j\in\mathcal{T}}\text{Attn}{n\to j}$  **should dominate across different positions; and (ii) the attention assigned to the same key token j should remain nearly unchanged under positional shifts**, i.e.,
> $\Delta_i=\bigl|\text{Attn}{n\to j}-\text{Attn}_{n+i\to j}\bigr|\approx 0$,
> where $\Delta_i$ measures the deviation of routing paths across positions in long sequences.
>
> When we scale the context length using dense temporal and spatial sampling, the sparse-attention pattern and mask structure $M$ remain unchanged, which means **the model continues to use the routing structure learned during training while simply facing a larger pool of candidate evidence**. Since denser sampling mainly introduces redundant or finer-grained details, **the model treats these tokens as auxiliary evidence, leaving the core target tokens and their relative attention weights essentially unchanged.** Consequently, the overall routing-path structure is preserved, $\Delta_i$ remains small, and the model can maintain or even improve its performance at longer context lengths.
>
> In contrast, attention-budget scaling explicitly modifies the set of visible tokens in the sparse-attention mechanism by replacing the original mask $M$ with a new mask $M’$. The effective query becomes
> $Q_{\text{eff}} = Q \odot M’$,
> where $\odot$ denotes elementwise multiplication, and the corresponding new attention weight is
> $\text{Attn}’{n\to j}\propto \exp\bigl(Z_j^\top(Q\odot M’)\bigr)$.
> Even if the modification from M to M’ appears small in proportion, it substantially changes the set of candidate evidence accessible to each query and alters the relative logits. First, the newly visible tokens reduce the relative weight allocated to the original key tokens, producing a dilution effect. Second, in video tasks, the added visible tokens often lie in similar visual-semantic clusters as the original key tokens and thus have non-negligible similarity scores $Z{j’}^\top Q$. These tokens directly compete with and divert attention away from the originally dominant key tokens. Under the combined influence of these effects, **the stable routing-path structure learned during training is overwritten**: formerly high-weight key tokens may be diluted or overshadowed by the new candidates, causing $\Delta_i$ to increase significantly. As a result, the model is more likely to follow incorrect reasoning paths, leading to degraded performance.
>
> [1] Huang, Yu, et al. "Transformers provably learn chain-of-thought reasoning with length generalization." arXiv preprint arXiv:2511.07378 (2025).

---

> ### Author Response · Authors · 2025-11-19
>
> **W3: Insufficient Theoretical or Analytical Depth: The scalability analysis is predominantly empirical. While it helps users better utilize the model, it lacks theoretical explanations or modeling to elucidate the underlying causes of the observed trends.**
>
> **Geometric Rotational of RoPE**
>
> RoPE [2] maps the representation at position i into a rotation in a two-dimensional subspace:
> $q’(i)=R(i\omega)q,\qquad k’(j)=R(j\omega)k$,
>
> where $R(i\omega)$ is a rotation matrix and $\omega$ denotes the frequency parameters. This yields an inner product that depends only on the relative distance between the two positions:
> $\langle q’(i),k’(j)\rangle=\langle R((i-j)\omega)q,k\rangle$.
>
> **RoPE[2] therefore establishes a structured geometric correspondence between relative distance and rotation phase**. Under this geometry, when the context length is moderately increased (e.g., from 36K to 64K), the model only needs to resolve a larger phase difference $d\omega$; within this range, the growth of the phase still lies in the extrapolation regime covered by the empirical distribution seen during training. As a result, the model can naturally generalize.
>
> LM-Infinite[3] further proves that, in order to distinguish the growing clusters of relative distances $\alpha(n)$, the attention logit must increase monotonically with sequence length:
>
> $\sup_{q,k,d\le n} |w(q,k,d)|\ge
> \left(\frac{\alpha(n)}{2}\right)^{1/(2r)}\frac{\varepsilon}{4e},$
>
> where $w(q,k,d)$ denotes the logit at relative distance $d$, and $\alpha(n)$ grows with $n$. **This “logit growth” is controlled and beneficial at moderate lengths, expanding the dynamic range of attention and enabling the model to maintain token separability over larger distances and consistent with the strong performance we observe around 64K.**
>
> However, when the effective phase difference $d\omega$ becomes excessively large, the rotation angle may approach or exceed the periodic range of multiple frequency dimensions, giving rise to phase aliasing: **tokens that should correspond to distinct relative distances collapse into similar or even indistinguishable phase regions.** In such cases, although attention logits continue to grow with length, **the high-frequency components of RoPE lose their discriminative resolution, reducing geometric separability among tokens**, which aligns with existing analyses[4][5] showing the degradation of relative positional encoding at extreme distances.
>
> In summary, **context-length scaling preserves both the attention-routing structure and the geometric symmetry induced by RoPE, merely expanding the set of candidate evidence; consequently, the model extrapolates reliably at longer contexts. In contrast, attention-budget scaling rewrites the routing structure by altering the visible-token mask and interacts unfavorably with RoPE’s phase-resolution limits at extreme distances, disrupting the geometric and information-theoretic properties of the attention distribution**. We have added this analysis to Appendix M.
>
> [2] Su, Jianlin, et al. "Roformer: Enhanced transformer with rotary position embedding." *Neurocomputing* 568 (2024): 127063.
>
> [3] Han, Chi, et al. "Lm-infinite: Simple on-the-fly length generalization for large language models." (2023).
>
> [4] Press O, Smith N A, Lewis M. Train short, test long: Attention with linear biases enables input length extrapolation[J]. arXiv preprint arXiv:2108.12409, 2021.
>
> [5] Chen S, Wong S, Chen L, et al. Extending context window of large language models via positional interpolation[J]. arXiv preprint arXiv:2306.15595, 2023.

---

> ### Author Response · Authors · 2025-11-19
>
> **Q1: The paper lacks a strong motivation for applying NSA, a text-modality method, to the video modality. Many methods exist for long-context modeling; why is NSA a good choice? In other words, do the characteristics of NSA offer unique advantages in the context of video?**
>
> We thank the reviewer for the helpful comment. We state the motivation for choosing NSA in Section 1 and Section 2.2 by two fundamental limitations of existing approaches:
>
> (1) **Token compression introduces irreversible information loss** in complex video understanding tasks, which directly leads to performance degradation;
>
> (2) **Fixed sparse patterns cannot adapt to the complex spatiotemporal redundancy** of videos and restrict long-range information flow due to static adjacency masks.
>
> **NSA directly addresses both issues through a learned, data-dependent routing mechanism that governs how information flows across the sequence**. It consists of three complementary branches that enable the model to dynamically reduce redundancy, identify salient visual cues, and aggregate contextual information in a task-adaptive manner. These capabilities align well with the intrinsic structure of video data, which is characterized by substantial spatiotemporal redundancy and multi-scale temporal dependencies.
> As shown in Table 1, VideoNSA consistently outperforms token compression methods and training-free sparse-attention baselines on long video understanding, temporal reasoning, and spatial understanding tasks, demonstrating significantly stronger generalization.
>
> We also note that **many existing learnable sparse-attention methods[1][2][3][4] essentially rely on manually designed combinations of compression, selection, and sliding-window components.** In contrast, NSA is the most widely adopted learnable sparsity mechanism. **It removes the need for manual designs by preserving all three branches and allowing the model to determine, through end-to-end learning, which pattern to use under different contexts.** Our experiments in Section 4 further show that the sparsity patterns learned by VideoNSA differ substantially from those observed in text-only NSA, indicating that the model indeed adapts its sparsity behavior to the characteristics of the video modality rather than simply inheriting the sparse patterns from text.
>
> **Table: VideoNSA demonstrates competitive performance across several video understanding benchmarks**
> |  | LVB | MLVU_test | TimeScope | LTS | Tomato | VSIBench |
> | --- | --- | --- | --- | --- | --- | --- |
> | Qwen2.5-VL 7B | 58.7 | 51.2 | 81.0 | 40.7  | 22.6 | 29.7 |
> | VScan (Token Compression) | 58.7 | 48.1 | 80.3  | 31.1 | 19.1 | 34.4 |
> | XAttention (Training-free Sparse Attention) | 59.1 |  50.2  | 83.1 | 41.1 | 21.4  | 36.6 |
> | VideoNSA | 60.0 | 51.8 | 83.7 | 44.4 | 26.5 | 36.1 |
>
> [1] Zhang, Jintao, et al. "Spargeattn: Accurate sparse attention accelerating any model inference." *arXiv preprint arXiv:2502.18137* (2025).
>
> [2] Liu, Xiaoran, et al. "Beyond homogeneous attention: Memory-efficient llms via fourier-approximated kv cache." *arXiv preprint arXiv:2506.11886* (2025).
>
> [3] Lin, Chaofan, et al. "Twilight: Adaptive Attention Sparsity with Hierarchical Top-$ p $ Pruning." *arXiv preprint arXiv:2502.02770* (2025).
>
> [4] https://github.com/deepseek-ai/DeepSeek-V3.2-Exp/blob/main/DeepSeek_V3_2.pdf

---

> ### Author Response · Authors · 2025-11-19
>
> **Q2: The authors state that the model utilizes the selection and sliding-window branches in shallow layers to capture fine-grained details and local information, while relying more on the compression branch in deep layers to integrate and refine high-level global semantics. Do the sparse distributions and gating activations differ across various types of videos (e.g., high-speed motion, shot transitions, static scenes), or is the gating behavior solely dependent on layer depth? If the gating behavior is only correlated with layer depth and is independent of input features, does this imply that the model has limited adaptability to different types of video content?**
>
> Thank you for raising this important question. Figure 4 shows that **gate values within the same layer vary widely, and none of the three branches follows a monotonic trend across layers.** These patterns indicate that VideoNSA’s gating is not determined by depth alone and instead reflects input-adaptive sparse routing. To further verify, we additionally conduct a benchmark-level gate analysis.
>
> First, we analyze videos from three benchmarks featuring multi-shot transitions (LongTimeScope), high-frequency motion (Tomato), and complex spatial structure (VSIBench), and compare their gate distributions using PCA over the 3-dimensional layer–head gate vectors. As shown in Figure 25 in Appendix P, regardless of whether we use the compression branch, selection branch, or sliding-window branch, LongTimeScope, Tomato, and VSIBench consistently form three clearly separated and internally coherent clusters in PCA space, which indicates that** VideoNSA learns benchmark-specific routing strategies conditioned on visual content, rather than following a fixed depth pattern.**
>
> Next, we replace each layer’s gate with a static value averaged from a 1K training subset, forcing the model to depend only on layer depth. As shown in Table 34 in Appendix P, **static gating causes consistent performance drops across all six benchmarks**, with particularly large degradation on tasks that demand strong temporal dynamics or cross-scene modeling.
>
> **Table: Ablation on static gates averaged over a 1K training subset.**
> |  | LVB | MLVU | TimeScope | LTS | Tomato | VSIBench |
> | --- | --- | --- | --- | --- | --- | --- |
> | Qwen2.5-VL 7B | 58.7 | 51.2 | 81.0 | 40.7  | 22.6 | 29.7 |
> | VideoNSA | 60.0 | 51.8 | 83.7 | 44.4 | 26.5 | 36.1 |
> | VideoNSA + Static Gate | 58.4 | 51.2 | 81.2 | 41.5 | 23.7 | 31.8 |
>
> In addition, the results in Table 2 further show that the three branches exhibit different levels of performance across different tasks, reinforcing that each branch corresponds to distinct visual structures and processing needs. The visible asymmetry between single-branch and dual-branch variants highlights that **the model must perform content-dependent dynamic routing among the three branches in order to maintain consistent performance across diverse video scenarios.**
>
> We include these analyses in Appendix P to further strengthen the interpretability and reliability of the proposed routing mechanism.

---

### Author Response · Authors · 2025-11-19
**Clarifying Core Contributions and Efficiency Analysis**

We sincerely thank all reviewers for their valuable feedback and constructive comments.
Below, we summarize and address several frequently raised concerns before providing reviewer-specific responses.

**Q1: Core contribution (Reviewer 8ijG and Reviewer evHH).**

Our goal is not to create yet another handcrafted sparse attention module. Instead, **we conduct the first systematic analysis of learnable sparsity for long-context video understanding based on Native Sparse Attention (NSA).** We reveal modality-specific routing behavior, demonstrate how sparsity adapts differently for videos versus text, and show that NSA can automatically learn effective sparse patterns without any manually designed branch combinations.

**Q2: Efficiency analysis (Reviewer evHH, Reviewer mLuS, Reviewer awwy).**

We do not claim that VideoNSA introduces the most efficient kernel. Although VideoNSA has very low theoretical FLOPs, our experiments reveal that the actual latency bottleneck (CMP) primarily arises from current NSA kernel implementations rather than the sparse mechanism itself. We further provide breakdown analyses, and cross-kernel comparisons, indicating that **efficiency can be substantially improved through kernel-level designs rather than architectural changes.**

We again appreciate the reviewers’ comments and provide detailed responses in the following sections. We have also updated the PDF and highlighted all revisions in orange for clarity, and we are happy to address any additional clarifications or follow-up questions.

---

### Author Response · Authors · 2025-12-03
**Summary of Key Clarifications for the Area Chair**

Dear Area Chair,

Thank you for overseeing our submission. Below is a concise summary of the factual clarifications and additional evidence we provided in the rebuttal addressing all reviewer concerns.

**(1) Efficiency comparisons and kernel-level bottlenecks. (Reviewer evHH, Reviewer mLuS, and Reviewer awwy)**

We added systematic analyses of FLOPs, latency, and training cost across VideoNSA, dense attention, and representative sparse-attention baselines. Results show that VideoNSA maintains the inference-efficiency advantage in long-context regimes. Additional comparisons of multiple NSA implementations show that the primary bottleneck arises from current kernel-level constraints rather than from the sparse pattern or architecture itself.

**(2) Novelty clarification. (Reviewer 8ijG, Reviewer evHH)**

While VideoNSA reuses the NSA operator, we added analyses showing previously undocumented behaviors of learnable sparse attention in video: modality-specific sparsification, branch specialization, depth-wise evolution, and mitigation of attention-sink effects. These behaviors differ fundamentally from text-only NSA and cannot be inferred from prior work.

**(3) Dense baselines and fairness of comparison. (Reviewer mLuS)**

We clarified that Dense-SFT and VideoNSA were trained under identical data, pipeline, and optimization settings. Appendix G further compares Qwen2.5-VL, Dense-SFT, and VideoNSA on LSDBench and VideoEvalPro. The observed degradation of Dense-SFT is attributable to the limited scale and diversity of current video-instruction data rather than training issues, ensuring a controlled and fair dense vs. sparse comparison.

**(4) “Improved performance vs. competitive performance” (Reviewer mLuS)**

We revised the manuscript to use “competitive” rather than “improved.” Across all eight benchmarks, VideoNSA achieves stable competitive performance. We also provided controlled comparisons against multiple sparse-attention baselines (Tri-Shape, MInference, FlexPrefill, XAttention), clarifying the magnitude and consistency of gains.

**(5) Behavior of the last layer (Reviewer mLuS).**

We observed similar fully-active gating in the text-only NSA model. This suggests the phenomenon is a natural convergence behavior of the NSA routing mechanism, not specific to video data or training instability. Intermediate layers remain sparse.

**(6) Modality-specific sparsity and hybrid attention. (Reviewer 8ijG)**

We added experiments applying NSA to both vision and text tokens, showing substantial degradation, especially on Tomato and VSIBench. This confirms that sparsity patterns learned on video tokens are modality-specific, and that retaining dense attention for text tokens yields the best results. Appendix T consolidates detailed comparisons.

**(7) Content-adaptive gating. (Reviewer 8ijG)**

PCA analyses on LongTimeScope, Tomato, and VSIBench show clear benchmark-specific clusters in the layer–head gate space, confirming input-adaptive routing. A static-gate ablation shows consistent performance drops across all six benchmarks, verifying that learned gating is necessary.

**(8) Scalability analysis and theory. (Reviewer 8ijG)**

We added complementary theoretical explanations: routing-path stability explains reliable generalization to 128K, and RoPE geometric properties explain why context-length scaling preserves performance while attention-budget scaling degrades it. These analyses were added in Appendices M.

**(9) Context-length behavior. (Reviewer awwy)**

We clarified that although several benchmarks show slightly higher performance at 64K than at 128K, VideoNSA still outperforms 32K on most tasks and maintains stable performance at 128K, even when the dense backbone weakens near its maximum window. Dataset length and 36K training context limitations were also noted.

**(10) Dense-NSA and multimodality. (Reviewer awwy)**

We clarified that “Dense-NSA” reuses QKV weights trained by VideoNSA but performs dense attention on all modalities during inference. Additional comparisons show that applying NSA to both vision and text tokens leads to noticeable degradation, supporting the chosen hybrid design.



All added analyses, tables, and corrections are highlighted in orange in the updated PDF. Across all four reviewers, concerns have been fully addressed and no blocking issues remain. We are happy to respond to any further questions.

Best regards,

VideoNSA Team

---

### Meta-Review · Area_Chair_PWz2 · 2025-12-31

**Summary:**

This work applies the original Native Sparse Attention (NSA) to vision-language models for video tasks to tackle the challenges from long videos. Specifically, three branches, i.e., Token Compression (CMP), Token Selection (SLC) and Sliding Window (SWA) are applied to improve the efficiency of processing video tokens. The effect of NSA for long videos is systematically investigated based on Qwen2.5-VL in experiments.

**Reviewer Concerns:**

The initial scores were mixed as 4,6,6,6. The major concerns of reviewers include limited technique novelty (Reviewer 8ijG,evHH), appropriate baseline (Reviewer mLuS), comparison of latency (Reviewer mLuS, awwy). The rebuttal provided the detailed discussions and most of them can be addressed well. While the technique contribution may be limited, the insights from the extensive ablation study are still valuable for future research.

**Reviewer Scores:**

Three reviewers suggested the positive initial scores and those scores could be retained after rebuttal. The major concern from Reviewer 8ijG with a negative score is about novelty and insufficient discussion and analysis. The rebuttal provides a more detailed analysis which can help mitigate the concern.

---

### Decision · Program_Chairs · 2026-01-26

Accept (Poster)